# Validation of GRASP algorithm product from POLDER/PARASOL data and assessment of multi-angular polarimetry potential for aerosol monitoring

Cheng Chen[1, 2], Oleg Dubovik[1], David Fuertes[2], Pavel Litvinov[2], Tatyana Lapyonok[1], Anton Lopatin[2], Fabrice Ducos[1], Yevgeny Derimian[1], Maurice Herman[1], Didier Tanré[1], Lorraine A. Remer[3], Alexei Lyapustin[4], Andrew M. Sayer[4, 5], Robert C. Levy[4], N. Christina Hsu[4], Jacques Descloitres[6], Lei Li[1, 7], Benjamin Torres[1], Yana Karol[2], Milagros Herrera[1], Marcos Herreras[1], Michael Aspetsberger[8], Moritz Wanzenboeck[8], Lukas Bindreiter[8], Daniel Marth[8], Andreas Hangler[8] and Christian Federspiel[8]

[1] Univ. Lille, CNRS, UMR 8518 - LOA - Laboratoire d'Optique Atmosphérique, F-59000 Lille, France
[2] GRASP-SAS, Villeneuve d'Ascq, France
[3] Joint Center for Earth Systems Technology, University of Maryland, Baltimore, MD, USA
[4] Universities Space Research Association, Columbia, MD, USA
[5] NASA Goddard Space Flight Center, Greenbelt, MD, USA
[6] Univ. Lille, CNRS, CNES, UMS 2877 - AERIS/ICARE Data and Services Center, F-59000 Lille, France
[7] State Key Laboratory of Severe Weather (LASW) and Key Laboratory of Atmospheric Chemistry (LAC), Chinese Academy of Meteorological Sciences, CMA, Beijing, 100081, China
[8] Cloudflight Austria GmbH, High Performance Computing, Linz, Austria

*Correspondence to*: Oleg Dubovik (oleg.dubovik@univ-lille.fr)

**Abstract.** Proven by multiple theoretical and practical studies, multi-angular spectral polarimetry is ideal for comprehensive retrieval of properties of aerosols. Furthermore, a large number of advanced space polarimeters have been launched recently or planned to be deployed in the coming few years (Dubovik et al., 2019). Nevertheless, at present, practical utilization of aerosol products from polarimetry is rather limited, due to the relatively small amount of polarimetric observations compared to photometric observations, as well as challenges in making full use of the extensive information content available in these complex observations. Indeed, while in recent years several new algorithms have been developed to provide enhanced aerosol retrievals from satellite polarimetry, the practical value of available aerosol products from polarimeters yet remains to be proven. In this regard, this paper presents the analysis of aerosol products obtained by the Generalized Retrieval of Atmosphere and Surface Properties (GRASP) algorithm from POLDER/PARASOL observations. After about a decade of development, GRASP has been adapted for operational processing of polarimetric satellite observations and several aerosol products from



POLDER/PARASOL observations have been released. These updated PARASOL/GRASP products are

publicly available (e.g., http://www.icare.univ-lille.fr, www.grasp-open.com/products/), the dataset used in

the current study is registered under: http://doi.org/10.5281/zenodo.3887265 (Chen et al., 2020).

The objective of this study is to comprehensively evaluate the GRASP aerosol products obtained

from POLDER/PARASOL observations. First, the validation of the entire 2005 - 2013 archive was

conducted by comparing to ground-based Aerosol Robotic Network (AERONET) data. The subjects of the

validation are spectral aerosol optical depth (AOD), aerosol absorption optical depth (AAOD) and single

scattering albedo (SSA) at 6 wavelengths, as well as Ångström exponent (AE), fine mode AOD (AODF)

and coarse mode AOD (AODC) interpolated to the reference wavelength 550 nm. Second, an inter-

comparison of PARASOL/GRASP products with the PARASOL/Operational, MODIS Dark Target (DT),

Deep Blue (DB) and Multi Angle Implementation of Atmospheric Correction (MAIAC) aerosol products

for the year 2008 was performed. Over land both satellite data validations and inter-comparisons were

conducted separately for different surface types, discriminated by bins of Normalized Difference

Vegetation Index (NDVI):  <0.2, 0.2≤ and <0.4, 0.4≤ and <0.6, and ≥0.6. Three PARASOL/GRASP

products were analyzed: GRASP/HP ("High Precision"), Optimized, and Models. These different products

are consistent but were obtained using different assumptions in aerosol modeling with different accuracies

of atmospheric radiative transfer (RT) calculations. Specifically, when using GRASP/HP or Optimized

there is direct retrieval of the aerosol size distribution and spectral complex index of refraction. When using

GRASP/Models, the aerosol is approximated by a mixture of several aerosol components, each with their

own fixed size distribution and optical properties, and only the concentrations of those components are

retrieved. GRASP/HP employs the most accurate RT calculations, while GRASP/Optimized and

GRASP/Models are optimized to achieve the best trade-off between accuracy and speed. In all these

options, the underlying surface reflectance is retrieved simultaneously with the aerosol properties and the

radiative transfer calculations are performed "on line" during the retrieval.

All validation results obtained for the full archive of PARASOL/GRASP products show solid quality of retrieved aerosol characteristics. The GRASP/Models retrievals, however, provided the most

solid AOD products, e.g. AOD (550 nm) is unbiased, has the highest correlation (R~0.92) and the highest fraction of retrievals (~55.3%) satisfying the accuracy requirements of the Global Climate Observing System (GCOS) when compared to AERONET observations. GRASP/HP and GRASP/Optimized AOD products show a non-negligible positive bias (~0.07) when AOD is low (<0.2). On the other hand, the detailed aerosol microphysical characteristics (AE, AODF, AODC and SSA, etc.) provided by GRASP/HP

and GRASP/Optimized correlate generally better with AERONET than do the results of GRASP/Models. Overall, GRASP/HP processing demonstrates the high quality of microphysical characteristics retrieval versus AERONET. Evidently, GRASP/Models approach is more adapted for retrieval of total AOD, while the detailed aerosol microphysical properties are limited when a mixture of aerosol models with fixed optical properties are used.

The results of a comparative analysis of PARASOL/GRASP and MODIS products showed that, based on validation against AERONET, the PARASOL/GRASP AOD (550 nm) product is of similar and sometimes of higher quality compared to the MODIS products. All AOD retrievals are more accurate and in good agreement over ocean. Over land, especially over bright surfaces, the retrieval quality degrades and the differences in total AOD products increase. The detailed aerosol characteristics, such as AE, AODF and

AODC from PARASOL/GRASP are generally more reliable, especially over land. The global inter-comparisons of PARASOL/GRASP versus MODIS showed rather robust agreement, though some patterns and tendencies were observed. Over ocean, PARASOL/Models and MODIS/DT AOD agree well with the correlation coefficient of 0.92. Over land, the correlation between PARASOL/Models and the different MODIS products is lower, ranging from 0.76 to 0.85. There is no significant global offset; though over

bright surfaces MODIS products tend to show higher values compared to PARASOL/Models when AOD is low, and smaller values for moderate and high AODs. Seasonal means suggest that PARASOL/GRASP products show more biomass burning aerosol loading in central Africa and dust over the Taklamakan

Desert, but less AOD in the northern Sahara. It is noticeable also that the correlation for the data over
AERONET sites is somewhat higher, suggesting that the retrieval assumptions generally work better over
AERONET sites than over the rest of the globe. One of the potential reasons may be that MODIS
retrievals, in general, rely more on AERONET climatology than GRASP retrievals.

Overall, the analysis shows that the quality of AOD retrieval from multi-angular polarimetric
observations like POLDER is at least comparable to those of single-viewing MODIS-like imagers. At the
same time, the multi-angular polarimetric observations provide more information on other aerosol
properties (e.g. spectral AODF, AODC, AE), as well as additional parameters such as AAOD and SSA.

## 1 Introduction

Over the past few decades, satellite remote sensing has provided essential advances in
understanding the global distribution of atmospheric aerosols (Kaufman et al., 2002; Remer et al., 2008)
and constraining aerosol climate effects (Bellouin et al., 2005; Myhre, 2009; Yu et al., 2006). Nevertheless,
aerosol effects remain the largest contributor to forcing uncertainty according to the Intergovernmental
Panel on Climate Change (IPCC) assessments (Boucher et al., 2013). Over the past few decades, satellite
remote sensing techniques have developed rapidly and extensively, and various (primarily photometric)
instruments have been developed and deployed to monitor atmospheric aerosols from space (Bréon et al.,
2011; Dubovik et al., 2019; King et al., 1999; Kokhanovsky et al., 2015; Li et al., 2009; Tanré et al., 2011).
While the design and capabilites of the photometric observations are constantly evolving, the greatest
improvement has been in the form of Multi-Angular multi-spectral Polarimetry (MAP) measurements
(Hansen et al., 1995; Knobelspiesse et al., 2012; Mishchenko et al., 2004; Waquet et al., 2009; Tanré et al.,
2011). MAP measurements have enough inherent information content to greatly improve our understanding
about aerosol properties. Several space-borne polarimeters have already been deployed and more advanced
versions will be deployed soon (Dubovik et al., 2019). In addition, there are many airborne versions of
orbital polarimeters that have operated during field campaigns, which can be used to verify and improve the



retrieval concepts (e.g. Knobelspiesse et al., 2020). Although the overall volume of polarimetric observations remains small compared to photometric observations, the potential for rapid advancement is large.

Several factors contribute to the current limited visibility of MAP observations and algorithms including: (i) limited amount of polarimetric observations in comparison to photometric ones, (ii) general complexity of polarimetric observations, and (iii) consequent challenges in developing capable retrieval algorithms. As a result, at present, there is a lack of extensive aerosol products from satellite MAPs that attract the aerosol science community. This tendency is especially evident by the contrast with the increase

of constantly improved aerosol products from mono- and bi- viewing photometric images. For example, the archive of most popular Moderate Resolution Imaging Spectroradiometer (MODIS) observations has been processed using many different algorithms, and NASA distributes three complementary MODIS aerosol products: Dark Target (DT) by Remer et al. (2005) and Levy et al., (2013), Deep Blue by Hsu et al. (2004, 2006, 2013) and Multi Angle Implementation of Atmospheric Correction (MAIAC) by Lyapustin et al.

(2018). Similarly, significant effort has been directed to improve aerosol products from European ENVISAT satellite platform observations in frame of Climate Change Initiative (CCI) projects of European Space Agency (e.g. see de Leeuw et al., 2015; Holzer-Popp et al., 2013; Popp et al., 2016). As a result, the product archives of MEdium Resolution Imaging Spectrometer (MERIS) and especially Advanced Along-Track Scanning Radiometer (AATSR) missions are constantly updated and improved.

To date, only one space-borne MAP has a long and wide enough coverage to advance aerosol science. The Polarization and Directionality of the Earth's Reflectances (POLDER) instrument was designed and developed by the French space agency Centre National d'Études Spatiales (CNES) to measure the spectral directional polarized solar radiation reflected by the Earth-atmosphere system (Deschamps et al., 1994). POLDER-1 and -2 flew on board of the Japanese Advanced Earth Observing

Satellites (ADEOS) platforms ADEOS-I and –II from November 1996 till June 1997 and from April 2003 till October 2003 correspondingly. Unfortunately, due to the failures of the platforms' solar panels, the



POLDER-1 and -2 have rather a limited time series of observations. POLDER-3 was launched in December 2004 on PARASOL (Polarization & Anisotropy of Reflectances for Atmospheric Sciences coupled with Observations from a Lidar) platform developed by CNES. POLDER-3/PARASOL (hereafter

PARASOL), was operational from March 2005 till October 2013 within the A-Train constellation, which is making nearly contiguous observations of many facets of the Earth system through a series of low-orbiting satellites (e.g. MODIS/AQUA, CALIOP/CALIPSO, OMI/AURA) (Parkinson, 2003; Schoeberl et al., 2006; Tanré et al., 2011; Winker et al., 2010). The PARASOL imager has 3 gaseous absorption channels (763, 765 and 710 nm), in addition to 6 channels (443, 490, 565, 670, 865 and 1020 nm) measuring the total

radiance, and 3 channels (490, 670 and 865 nm) measuring the polarization. The number of viewing angles is similar for all spectral channels varying from 14 to 16 depending on solar zenith angle and geographical location. PARASOL provided global coverage about every 2 days with a nadir spatial resolution ~6 km (Tanré et al., 2011).

Several POLDER-1, 2 and PARASOL aerosol products were developed by the science team at

LOA (Laboratoire d'Optique Atmosphérique, Lille, France). Hereafter, we refer to this aerosol product as POLDER/Operational or Operational. The initial POLDER/Operational aerosol retrieval over ocean by Deuzé et al. (1999) provided total Aerosol Optical Depth (AOD) from the measured total and polarized radiances at 670 and 865 nm with expected accuracy of ±0.05±0.05AOD (Goloub et al., 1999). The updated algorithm by Herman et al. (2005) provided AOD of fine and coarse modes and, when geometrical

conditions are optimal (scattering angle ranging between 90°-160°), the spherical/non-spherical separation of coarse mode particles (Herman et al., 2005). Over land, the algorithm by Deuzé et al. (2001) retrieves only fine ("accumulation") mode AOD (AODF) using only polarized light at two wavelengths (670 and 865 nm) to capitalize on the small and fairly neutral polarized reflectance typical of land surfaces (Deuzé et al., 2001; Herman et al., 1997). These algorithms were designed to utilize the benefits of MAP information

within the framework of a conventional MODIS like Look-up-Table (LUT) approach (Tanré et al., 1997; Kaufman et al. 1997) and did not intend to extend substantially the set of retrieved parameters. Moreover,

over land the POLDER/Operational retrieval provided only AODF while MODIS algorithms derives the total AOD.

The Generalized Retrieval of Atmosphere and Surface Properties (GRASP) algorithm considered

here was developed to further exploit the aerosol information content of POLDER spectral multi-angular polarization measurements (Dubovik et al., 2011, 2014). The algorithm allows for a large number of unknown parameters and retrieves a set of parameters affecting measurements at all wavelengths, all angles, and all states of polarization using the multi-term least square method (Dubovik, 2004). As will be later described in detail in section 2.1, GRASP does not utilize pre-calculated LUTs but instead searches in a

continuous space for the solutions and optimizes the statistical properties of the obtained retrieval. The GRASP algorithm derives an extended set of aerosol parameters from POLDER data, including spectral AOD, spectral Aerosol Absorption Optical Depth (AAOD), spectral AODF, spectral AODC, particle size distribution, Single Scattering Albedo (SSA), complex refractive index, fraction of spherical particles, etc. (see Table 1 and discussion in the next Section). The full archives of POLDER-1, 2 and PARASOL were

processed with GRASP and the resulting datasets are available for public at the official GRASP algorithm website (www.grasp-open.com) and the AERIS/ICARE Data and Services Center (http://www.icare.univ-lille.fr).

This paper presents and discusses new publicly-available aerosol products obtained from POLDER observations, which represent the longest to date satellite MAP record (Tanré et al., 2011; Dubovik et al.,

2019). Specifically, the discussion focuses on a new extended aerosol product generated by the recently developed GRASP algorithm (Dubovik et al., 2011, 2014). Here we perform quantitative analysis of PARASOL/GRASP aerosol products (the longest POLDER data set) through validation with AERONET reference data, as well as by comparisons with the POLDER/Operational product and the widely used MODIS DT, DB and MAIAC aerosol products. The analysis pursues two objectives. The first is to

understand the accuracy and value of each PARASOL/GRASP aerosol product. The second objective is to clarify the specifics, advantages, and shortcomings of MAP aerosol products compared to those from

photometric mono-viewing imagers. Thus, the analysis provides useful information for the aerosol community to meet the future challenge of accurate aerosol monitoring in the coming era of polarimetric missions. Over the next few years, we expect deployment of a number of new and existing satellite and

airborne MAPs including 3MI (Multi-View Multi-Channel Multi-Polarization Imaging), DPC (Directional Polarimetric Camera), Aerosol-UA (Ukraine), PACE (Plankton, Aerosol, Cloud, ocean Ecosystem), AirHARP (Airborne Hyper-Angular Rainbow Polarimeter), AirMSPI (Airborne Multi-angle SpectroPolarimeter Imager), SPEX (Spectro-Polarimetric Experiment), RSP (Research Scanning Polarimeter), etc. (Dubovik et al., 2019; Fougnie et al., 2018; Fu et al., 2020; Gao et al., 2020; Hasekamp et

al., 2019; Knobelspiesse et al., 2020; Li et al., 2018; Milinevsky et al., 2019; Puthukkudy et al., 2020; Remer et al., 2019). By providing a comprehensive analysis of PARASOL/GRASP products, we prove that the aerosol community can utilize the new era of MAP measurements.

## 2 Data description and validation approach

The analysis compares several satellite data products. From POLDER, we have both the products of

the Operational algorithm and the GRASP retrieval. From MODIS, we utilize products generated by three different algorithms (DT, DB, and MAIAC). For all satellite products, validation is based on AERONET observations and retrievals.

### 2.1 POLDER/GRASP aerosol products

GRASP is a new-generation algorithm developed for deriving extensive aerosol properties from all

varieties of remote sensing instruments. The overall concept of the algorithm is described by Dubovik et al. (2014), while specific technical aspects are detailed in Dubovik et al. (2011). GRASP is based on highly advanced statistically optimized fitting implemented as multi-term least square minimization (Dubovik, 2004) which had earlier been successfully implemented for aerosol retrievals from ground-based AERONET radiometers (Dubovik and King, 2000; Dubovik et al., 2000, 2002a, 2002b, 2006). GRASP

inherits many methodological aspects in numerical inversion and aerosol modeling from the AERONET

retrieval developments. In fact, all retrieval set-ups including modeling of aerosol microphysical and optical properties, surface reflectance, numerical inversion, utilization of multiple a priori constraints, etc. can be realized using GRASP. At the same time, the GRASP concept and algorithm are highly flexible and versatile. GRASP includes several additional original features, and enables the implementation of advanced

retrieval scenarios. A unique aspect of GRASP is that it can perform radiative transfer (RT) computations fully accounting for multiple interactions of the scattered solar light in the atmosphere, and that it can perform it online without the use of traditional LUTs.

The GRASP retrieval can utilize whatever information content is available. If there is sufficient information content of the observations, GRASP will find the aerosol solutions. In the case of any currently

operational observations, GRASP can make optimal assumptions to constrain the solution. For example, GRASP can retrieve both aerosol and underlying surface properties simultaneously from satellite observations using additional a priori constraints on the spectral variability of land Bidirectional Reflection Distribution Function (BRDF). Or (probably the most essential methodological novelty) it can operate by relying on the multi-pixel concept wherein the statistically optimized retrieval is performed simultaneously

for a large group of pixels (Dubovik et al., 2011). This feature brings additional possibilities for improving the accuracy of satellite retrievals by using known constraints on the inter-pixel variability of retrieved aerosol and surface reflectance parameters. As a result, using this methodology GRASP provides reliable retrievals of detailed aerosol properties that traditionally have been difficult to obtain from satellites, for example, spectral AOD and AAOD over land including very bright deserts. The GRASP algorithm source

code and detailed documentation are available from https://www.grasp-open.com.

It should be noted that GRASP is a flexible inversion algorithm that can be applied to a wide variety of satellite, ground-based and laboratory observations. It has already been applied to ground-based AERONET photometers and LiDARs (Benavent-Oltra et al., 2017, 2019; Hu et al., 2019; Lopatin et al., 2013; Titos et al., 2019; Tsekeri et al., 2017), sky cameras (Román et al., 2017), polar-nephelometer data

(Espinosa et al., 2017, 2019; Schuster et al., 2019), and surface measurements of AOD (Torres et al., 2017).
In addition, GRASP is being used for several satellite instruments; aerosol products were generated for POLDER observations (discussed here) and for MERIS/Envisat, and there are ongoing developments for producing GRASP aerosol products from Sentinel-3 and -5P observations and operational aerosol retrievals for future Sentinel-4 and 3MI/Metop missions. GRASP is constantly being updated to produce many user-

oriented products such as estimates of covariance matrices (Herrera et al., in preparation, 2020), direct radiative forcing (Derimian et al., 2016), and so on.

For POLDER, GRASP utilizes radiance and polarization observations from all available spectral channels with minor gaseous absorption, i.e. for total radiance 5 channels for POLDER-1 and -2, and 6 for PARASOL, and for polarized radiances (3 spectral channels for all instruments). The retrieval uses a

unique global set of constraints (no location-specific assumptions) and a single initial guess globally. GRASP performs radiative transfer computations fully accounting for multiple interactions of the scattered solar light in the atmosphere on-line without using a traditional LUT. Since these RT computations are complex and time consuming, significant effort has been put into optimization and acceleration of the code for operational processing of voluminous datasets. At present, the speed of GRASP retrieval is appropriate

for processing the full archive of POLDER observations at native resolution (POLDER-1 and -2 at ~7 km and PARASOL at ~6 km) using rather moderate computing resources, e.g. 3-4 sec/pixel for GRASP/HP, 0.3-0.5 sec/pixel for GRASP/Optimized and 0.1-0.2 sec/pixel for GRASP/Models, in a single core processor (the description of GRASP/HP, GRASP/Optimized and GRASP/Models will be detailed further in this section).

Since GRASP has been designed for use with different observations, it allows a variety of different possibilities on modeling aerosol scattering, surface reflectance and generally on implementing atmospheric radiative transfer calculations. As a result, different configurations of the atmospheric forward model can be used even for interpretation of the same data (as is the case here with POLDER). Currently, the full POLDER/PARASOL data archive is processed by GRASP using the three following retrieval

configurations:





1) PARASOL/GRASP «optimized» (in the sense that radiative transfer calculations were optimized to find the best trade-off between speed of processing and accuracy of results);

2) PARASOL/GRASP «high-precision» (radiative transfer calculations with high precision were used).

3) PARASOL/GRASP «models» (the simplest, fastest processing; aerosol is assumed to be external mixture of several aerosol models).

The «optimized» and «high-precision» are different only by the precision of the RT calculations, while conceptually they are the same: aerosol size distribution, spectral values of complex index of refraction, fraction of spherical particles and the Aerosol Layer Height (ALH), are retrieved simultaneously with the

surface BRDF and Bidirectional Polarization Distribution Function (BPDF) parameters. The retrievals were performed using one aerosol component model with 5 bins of the size distribution and spectrally dependent complex refractive index. The aerosol vertical distribution was modeled using an exponential profile and scale height was retrieved. The details of implementation are discussed by Dubovik et al. (2011). The «models» approach uses different assumption for modeling aerosol properties (surface treatment is the

same as above): aerosol is assumed to be an external mixture of several aerosol components and only their concentrations are retrieved together with ALH and spectral BRDF/BPDF parameters. The size distribution, complex refractive index and non-sphericity parameter for each aerosol component are derived from the results of AERONET aerosol climatology for the main distinct aerosol types (Dubovik et al., 2002b) and improved in a series of sensitivity tests with satellite data. For retrievals over land, GRASP

retrieves the parameters of the Ross-Li BRDF (Li and Strahler, 1992; Ross, 1981) and BPDF (Maignan et al., 2009) models under assumption that the retrieved parameters are spectrally smooth (the strength of smoothness is different for each parameter) (Litvinov et al., 2011a, 2011b). For retrievals over ocean, the wind speed and a spectrally dependent Lambertian albedo are included in the state vector. It should be noted that "models" approach firstly was intended to be used for mono viewing satellite observations such

as those from MERIS/Envisat. However, once the approach was tested with PARASOL data, the obtained

results were quite appealing especially in conditions of low aerosol loading, motivating the generation of PARASOL/GRASP «models» archive that is included in the consideration here.

The three archives (Optimized, HP and Models) are released publicly and can be found at the AERIS/ICARE Data and Services Center (http://www.icare.univ-lille.fr) and at GRASP-OPEN website

(https://www.grasp-open.com/products/) in slightly different formats. The AERIS/ICARE is the official distributor of POLDER Level-1 and -2 data and allows the user to dive into the data using a web tool, which plots the results online. The AERIS/ICARE provides detailed visualization of the data, while GRASP-OPEN site is faster in releasing new products but with no visualization. The original PARASOL/GRASP retrievals are stored at Level-1, Level-2 and -3 products and are publicly available in

the form of daily, monthly, seasonal, yearly and climatological datasets. The Level-2 data contain full resolution data filtered following established quality criteria. Level 3 data is aggregated into a 0.1° and 1° grid box using the sinusoidal projection from Level-2 data. The list of retrieved aerosol parameters, as well as derived aerosol characteristics can be found in the Table 1. In this study, we adopt the current latest version of Optimized, HP (v1.2) and Models (v2.1) products.

**[Table 1]**

In addition to the PARASOL/GRASP products, all observations of POLDER-1 and -2 were also processed (using a single GRASP/Models approach). These data records are much shorter than PARASOL and therefore not included in the following analysis. However, based on limited comparisons (not presented here), the quality of the POLDER-1 and -2/GRASP retrievals is expected to be similar to those of

PARASOL/GRASP retrievals. Also, recently a new "GRASP/Component" approach has been developed (Li et al., 2019, 2020a, 2020b). This approach retrieved the size resolved fractions of aerosol components representing the different composition species, like black carbon, brown carbon, fine/coarse mode non-absorbing soluble and insoluble, coarse mode absorbing and aerosol water. The retrieved fractions drive the aerosol spectral index of refraction in modeling atmospheric radiances. This provides a fourth retrieval



archive; however, the results have not yet been fully analyzed and are not released in a user friendly format,

so the GRASP/Component data set will not be considered in this study.

PARASOL/GRASP aerosol products have already appeared in many studies, i.e. validation (Tan et al., 2019; Wei et al., 2019, 2020), data assimilation (Chen et al., 2018, 2019), AOD products merging (Li et al., 2020; Sogacheva et al., 2020). Despite these preliminary applications of the products, no systematic

evaluation of the global PARASOL/GRASP aerosol products has been published. Moreover, most early studies are based on the GRASP/Optimized products, which were released first. The evaluation of PARASOL/GRASP surface properties, as well as aerosol microphysical parameters (size distribution, complex refractive indices, fraction of spherical particles), and aerosol layer height, will be the subject of separate studies.

**2.2 MODIS Dark Target, Deep Blue and MAIAC aerosol products**

The MODIS sensors on board TERRA since 2000 (overpass ~10:30 local) and AQUA since 2002 (overpass ~13:30 local) provide near-global coverage twice per day. In this study, we will employ products from MODIS/AQUA only, which is on the same A-Train afternoon constellation orbit as PARASOL. MODIS has a wider swath of 2330 km compared to ~1600 km of PARASOL, 36 spectral channels ranging

from 410 to 15000 nm and higher spatial resolution for cloud mask. There are 3 mature aerosol products produced operationally and distributed by NASA: *Dark Target, Deep Blue and MAIAC.*

*MODIS Dark Target*

The Dark Target (DT) algorithm over land is based on an empirical surface reflectance relationship

between blue and red channels with the shortwave infrared (2113 nm) radiance. The AOD is retrieved by matching LUT values to observations at 466 nm, and then varying the weighting between two fixed aerosol models until the residual between LUT and observations are minimized at 645 nm. The main product is AOD at 553 nm with AOD reported at 466 nm, 645 nm and 2113 nm, consistent with the selected weighted aerosol model (Kaufman et al., 1997; Levy et al., 2007a, 2007b). Over ocean, the simplicity of the dark

ocean surface permits the retrieval of AOD and aerosol particle size (Tanré et al., 1997). In this situation

the algorithm chooses one fine mode out of four and one coarse mode out of five, along with the relative

weight between fine and coarse mode by minimizing the summed difference between LUT and

observations in six wavelengths (550, 660, 870, 1240, 1610, and 2130 nm) (Tanré et al., 1997; Remer et al.,

2005; Levy et al., 2013). The MODIS DT aerosol products are periodically updated to improve overall

performance (Levy et al., 2003, 2007a, 2007b, 2013; Remer et al., 2005; Gupta et al. 2016). The widely

recognized limitation of the DT algorithm is the complex spectral structure of bright land surfaces (e.g.

desert, bare soil, snow) that violates the assumptions of the empirical relationships between wavelengths

and increases uncertainty in the aerosol retrievals to unacceptable levels. Therefore, DT does not provide

coverage over these cases.


*MODIS Deep Blue*

The Deep Blue (DB) algorithm retrieves over both bright (except snow) and vegetated land

surfaces. It is able to retrieve over brighter surfaces than DT because it makes use of the much darker

surface reflectance in the deep blue (412 nm) channel (Hsu et al., 2004, 2006, 2013). Depending on the

processing path, determined by observed reflectance and vegetation indices, the algorithm will invoke

empirical spectral relationships of surface reflectance similar to DT (vegetation), rely on a pre-calculated

data base of surface reflectance (arid/deserts) or apply a hybrid method (urban surfaces). The MODIS DB

aerosol products have also gone through several version updates (Hsu et al., 2013; Sayer et al., 2015).

Within the MODIS official products, the DB algorithm is applied for only land aerosol retrieval. Over

vegetated surfaces DT tends to provide more retrievals in the tropics, and DB more retrievals at mid-

latitudes, due to different pixel selection and cloud screening criteria (Sayer et al., 2014).

*MODIS MAIAC*

The Multi Angle Implementation of Atmospheric Correction (MAIAC) algorithm has been

developed and applied to MODIS (Lyapustin et al., 2011a, 2011b, 2012, 2018), and is running

operationally in the NASA system. The MAIAC algorithm uses the minimum reflectance method to

dynamically characterize spectral ratios of the surface reflectance (which are prescribed in the DT) and

separate aerosol and surface contributions to the measurements. The accumulation of up to 16 days of the

last observations in the operational memory allows MAIAC to derive spectral surface BRDF. The MAIAC

aerosol product is available at higher spatial resolution of 1 km, in comparison to DT and DB that provide

aerosol products at 3 km and 10 km. As a more recent addition to the MODIS family of aerosol products

than DT and DB, MAIAC has shown itself to produce an AOD product as accurate or better than the older

algorithms over all types of land surfaces (Jethva et al., 2019), and thus offers a complementary/alternative

product to those from the original DT and DB algorithms.


All three MODIS algorithms (DT, DB and MAIAC) are developed based on LUT approaches with

a fixed certain number of aerosol models. Over ocean, DT assumes 9 aerosol models (4 fine models plus 5

coarse models); any retrieval corresponds to one of total 20 combinations of one fine mode and one coarse

mode (Levy et al., 2003; Remer et al., 2005; Tanré et al., 1997). Over land, DT algorithm adopts aerosol

models from AERONET retrievals, clustering down to three possible spherical fine-mode dominant models

(non-absorbing, moderately-absorbing and absorbing) and 1 spheroid coarse-mode dominant model (Levy

et al., 2007a). In addition, the fine- and coarse-mode dominant aerosol models over land are defined as a

function of season and location (Levy et al., 2013). The DB algorithm makes use of prescribed dust, and

smoke/sulfate aerosol models in the LUT (Hsu et al., 2013). For example, over vegetated surfaces, AE is

limited to some extent ($0.0 \leq AE \leq 1.8$), and fixed at 1.5 for low AOD conditions. Over bright arid/desert

surfaces the AE is limited to a maximum of 1.0 (Hsu et al., 2013; Sayer et al., 2013). A geographic

distribution of aerosol models is also adopted in the MAIAC algorithm, where the aerosol model

parameters are regionally, and may be parameterized as a function of AOD (dynamic models) for regions

with high humidity variations. The detailed description of the MAIAC regional aerosol models can be

found in Lyapustin et al., (2018). Hence, the MODIS aerosol products do not have the ability to retrieve

aerosol particle properties with known uncertainties, with the exceptions of size parameter (over ocean in

DT), SSA for dust (in DB), and AE (with known caveats).

In this study, MODIS Collection 6 aerosol products (MYD04_L2) from DT and DB algorithms

were acquired from the AERIS/ICARE Data and Services Center (http://www.icare.univ-lille.fr, last

access: 30 August 2019), where the unchanged NASA MODIS data are redistributed with enhanced

visualization. Note that the latest versions of DB and DT are Collection 6.1, although the differences

between the two versions are small on a large scale (Sayer et al., 2019) and do not significantly affect the

conclusions presented here. The latest MAIAC Collection 6 aerosol data (MAC19A2) is obtained from

NASA LAADS (Level-1 and Atmosphere Archive and Distribution System) DAAC (Distributed Active

Archive Center) (https://ladsweb.modaps.eosdis.nasa.gov, last access: 8 January 2020).

### 2.3 AERONET Dataset

The Aerosol Robotic Network (AERONET) is a global distributed network of well-calibrated sun-

sky photometers (Holben et al., 1998). By measuring direct Sun radiance, AERONET provides high

temporal (every 3 or 15 minutes in daytime depending on the operation mode of the instruments) multi-

wavelength AOD products with high reliable accuracy (~±0.01 to ±0.02) (Eck et al., 1999). Strict protocols

for the calibration and maintenance assure homogeneity among all its instruments. Due to its high data

quality, the AERONET AOD products are widely used as "ground truth" to evaluate satellite remote

sensing aerosol products (Bréon et al., 2011; Chu et al., 2002; Kahn et al., 2005; Liu et al., 2004; Remer et

al., 2005, 2002; Sayer et al., 2013).

In addition to direct Sun observations, AERONET radiometers conduct routine measurements of the

sky-scanning diffuse radiation. These observations are used to derive aerosol microphysical properties, e.g.

single scattering albedo, complex refractive index, size distribution and sphericity via Dubovik and King

(2000). The accuracy of the AERONET inversion products has been analyzed in many studies (Dubovik et

al., 2000; Sinyuk et al., 2020) and resulting recommendations were adopted for providing aerosol products

of highest quality (e.g. increase of quality of retrieval products with aerosol loading and range of observed

scattering angles). The microphysical properties provided by AERONET contribute to aerosol and climatic

applications. For example, the AERONET-derived aerosol particle property climatology (Dubovik et al.,

2002b), are used in some form in nearly in all satellite retrieval algorithms (including MODIS, see Levy et

al., 2007b; Lyapustin et al., 2018) and feed the climate models used to characterize aerosol climate effects

(Kinne et al., 2003; Sato et al., 2003).

In this study, the up-to-date AERONET Version 3 Level 2.0 dataset (http://www.aeronet.gsfc.nasa.gov, last access: 3 September 2019) (Giles et al., 2019) with standard cloud screening and quality control were used (Smirnov et al., 2000). We make use of all AERONET sites with data during the POLDER/PARASOL archive (2005-2013). The AERONET direct-sun AOD, Ångström

Exponent, fine and coarse mode AOD from spectral deconvolution algorithm (SDA) (O'Neill et al., 2003), AAOD and SSA products are chosen as references for satellite products validation.

## 2.4 Data quality assurance and matchup methodology

One of the main issues in satellite data validation is how to match the temporally-varying AERONET point measurements with the spatially-varying satellite remote sensing aerosol products at over

pass time (Ichoku et al. 2002). This issue is compounded when multiple satellite products are involved that vary from ~1 km to ~100 km pixel spatial resolution. There are some insightful studies (Kinne et al., 2013; Schutgens et al., 2017) that quantify the AERONET sites spatial representativeness at the scales from ~50 km to ~300 km, which can be used for evaluation of chemical transport model simulations. However, the spatial resolutions (~50 km to ~300 km) considered in those studies are seemingly too coarse for validation

of satellite products of 1 km for MAIAC, 10 km for DB and DT and ~6 km data from PARASOL/GRASP.

This study considers aerosol products at 10 km spatial resolution; that is the native resolution of MODIS DB and DT products and seems to be the best compromise for comparing PARASOL/GRASP, MODIS DT, DB and MAIAC results. Also, 10 km is utilized by the aerosol community and other datasets

(e.g. ESA CCI products mentioned earlier). That also was a reason for the generation of

PARASOL/GRASP Level 3 products. Thus, we adopted PARASOL/GRASP Level 3 daily 0.1° gridded

aerosol products, MODIS/AQUA Level 2 daily DT and DB 10 km products and the 1 km MODIS MAIAC

aggregated to 0.1° (MAIAC_0.1) and 0.01° (MAIAC_0.01) resolution for the inter-comparisons.

MAIAC_0.01 essentially represents the single 1 km pixel retrieval. The PARASOL/Operational L2 daily

aerosol product is directly used for validation, which is at 18.5 km x 18.5 km spatial resolution.

The strategies to select PARASOL/GRASP retrieval products with highest quality are presented in

Table 2. The land pixel is defined only if 100% of the 0.1° by 0.1° grid box has been identified as land, so

an ocean pixel must contain 0% land. Also, to guarantee proper coast elimination, the first pixel bordering

ocean and land is removed (see Fig. 1). We selected the more reliable retrievals using "Residual Relative"

(mean-root-square of relative error in fitting the measurements by the algorithm) for PARASOL/GRASP

products. We adopted the same threshold for GRASP/Optimized and GRASP/HP (0.05 over land and 0.1

over ocean). These thresholds are suggested for general users. For GRASP/Models approach, we did not

use any filter, since L3 products were generated from L2 using filtering. GRASP data filtering and quality

assurance schemes are likely to be improved in the future. Nonetheless, in this study we tried to avoid

additional filtering of PARASOL/GRASP L3 products, since most of users would utilize the data with no

screening or with a very straightforward filtering. For MODIS DT, DB and MAIAC products, we select the

data only with the highest Quality Assurance (QA) flag. The highest "Quality Index" was selected for

PARASOL/Operational products (Bréon et al., 2011). Any pixel with fitting residual higher than the

threshold for PARASOL or QA lower than the highest flag for MODIS will be set to "no data".

**[Table 2]**

For validation with AERONET over land, we averaged all land satellite retrievals in a 3x3 window

for the gridded satellite data centered over the AERONET station. For ocean sites, in order to select pure

ocean pixels and keep reasonably high number of validation points, we decided to use a 9x9 window over

the AERONET site, using only pure ocean pixels. Any ocean pixels adjacent to land or land-ocean mixed



pixels were omitted as represented in Figure 1. The minimal number of accepted satellite data pixels within

the window is 1 over land and 41 over ocean; otherwise, the data were excluded from comparison. The

PARASOL/Operational product is treated a bit differently over ocean due to its relatively coarse resolution

(~18.5 km), with a similar land-like 3x3 window centered over the AERONET station.

The AERONET direct-sun AOD, AE, AODF and AODC data were averaged within ±30 minutes of

the MODIS/AQUA and PARASOL overpass time, while AERONET SSA and AAOD (which have a lower

sampling frequency) are averaged within ±180 minutes. In addition, AERONET station elevations greater

than 3600 m above mean sea level, satellite 3x3 or 9x9 data sets with AOD standard deviation greater than

0.05 between window pixels were excluded.

**[Figure 1]**

**2.5 Considered metrics for comparison statistics**

For quantifying the validation results, we used standard statistical parameters, including Pearson's

linear correlation coefficient (R), root mean square error (RMSE), slope and offset of linear regression and

bias.

$$\text{R} = \frac{\sum (O_{i,\text{satellite}} - \overline{O_{\text{satellite}}})(O_{i,\text{AERONET}} - \overline{O_{\text{AERONET}}})}{\sqrt{\sum (O_{i,\text{satellite}} - \overline{O_{\text{satellite}}})^2 \sum (O_{i,\text{AERONET}} - \overline{O_{\text{AERONET}}})^2}} \tag{1}$$

$$\text{RMSE} = \sqrt{\frac{\sum_{i=1}^{N}(O_{i,\text{satellite}} - O_{i,\text{AERONET}})^2}{N}} \tag{2}$$

$$\text{BIAS} = \frac{1}{N}\sum_{i=1}^{N}\left(O_{i,\text{satellite}} - O_{i,\text{AERONET}}\right) \tag{3}$$

where $N$ is the number of matched data points $i$; $O_{\text{satellite}}$ represents the observations from satellite, and

$O_{\text{AERONET}}$ represents the referenced observations from AERONET; $\overline{O_{\text{satellite}}}$ and $\overline{O_{\text{AERONET}}}$ are the mean

value for satellite and AERONET observations.

For MODIS validation, a commonly-used metric is the fraction agreeing within and Expected Error

(EE) envelope such as ±0.05±0.15AOD (Remer et al., 2005) or ±0.05±0.1AOD (Lyapustin et al., 2018). In





this study, we adopted stricter requirements proposed by the Global Climate Observing System (GCOS) (the greater of 0.03 or 10%), which have been adopted in the Aerosol_cci study (Popp et al., 2016) as well as the latest DB validation (Sayer et al., 2019). Following the Aerosol_cci study by Popp et al. (2016), the uncertainty of 0.01 for AERONET AOD has been taken into account and GCOS is defined as:

$\text{GCOS} = \max(\pm 0.04, \pm 0.1\text{AOD})$          (4)

Hence, the GCOS fraction (%) is the percentage of satellite retrieved AOD satisfying the GCOS requirement.

### 3 Validation of satellite observation by comparison with AERONET data: results and discussion

In order to characterize the quality of the retrieved aerosol parameters from PARASOL, the set of
main aerosol parameters including AOD, AE, AODF, AODC, SSA and AAOD were evaluated for the entire PARASOL ~9 years (2005-2013) data archive. This list includes all main aerosol parameters expected to be retrieved from MAP instruments in general (Dubovik et al., 2019). In addition, the validation results of AOD, AE, AODF and AODC were compared with the results of validation of these (where available) from the standard MODIS products for the year 2008.

PARASOL/GRASP retrievals are available and validated at six wavelengths (443, 490, 565, 670, 865 and 1020 nm). The MODIS retrievals and even PARASOL/Operational have different spectral coverage and, therefore, the comparisons of the GRASP product focused on the aerosol properties at midvisible (550 nm) that is commonly used in the satellite data comparisons and analysis (e.g. Sayer et al., 2018; Sogacheva et al., 2020). Therefore, for PARASOL/GRASP and PARASOL/Operational data the
aerosol products were generated at 550 nm by interpolations in log-log space from the closest channels available from the products. Similarly, AERONET aerosol products were also interpolated to 550 nm since the ground-based radiometers do not have a 550 nm channel.

### 3.1 Global validation of POLDER/GRASP aerosol products

*Aerosol Optical Depth*

Fig. 2 shows scatter plots of co-located PARASOL/GRASP AOD against AERONET AOD at 550

nm for the entire POLDER/PARASOL archive; Fig. 2a for GRASP/Optimized; Fig. 2b for GRASP/HP;

and Fig. 2c for GRASP/Models. Validation metrics for total spectral AOD (443, 490, 550, 565, 670, 865

and 1020 nm), as well as AOD separated for land and ocean, are presented in Table 3. As can be seen from

Fig. 2 and Table 3, all retrievals present good agreement with AERONET spectrally. Overall, based on

these metrics the quality of the comparison with AERONET is best for GRASP/Models. For example, for

AOD (550 nm) GRASP/Models shows better performance than GRASP/Optimized and GRASP/HP:

R=0.923 as compared to 0.877 and 0.899 and RSME=0.119 for GRASP/Models as compared to 0.160 and

0.161 for GRASP/Optimized and GRASP/HP respectively at 550 nm (see in Fig. 2). GRASP/Optimized

and GRASP/HP show a positive overall bias of 0.06-0.07 for all AOD conditions, that remains for low

AOD conditions (AOD<0.2) and even increases to 0.08 (GRASP/Optimized). In comparison,

GRASP/Models has small overall bias (of 0.01 for AOD at 550 nm) that slightly increases to 0.03 for high

AOD conditions (AOD>0.7). Because of the bias in GRASP/HP and GRASP/Optimized AOD, GCOS

fraction for them is much lower than for GRASP/Models AOD: e.g. 55.3% (AOD at 550 nm) for land +

ocean vs. 28.2% and 34.4% respectively. Over ocean, all three archives show good correlation with

coefficients R > 0.93 at 550 nm. Nevertheless, GRASP/Models over ocean has the highest R= 0.950 and

offers the best performance for the other statistical metrics.

       It is very important to note the robust performance of PARASOL/GRASP AOD retrieval in all

spectral channels. For example, GRASP/Models product shows only minor spectrally independent bias of

0.01 over land, and over ocean the bias is about 0.02 at 440 nm and decreases to zero at longer wavelengths,

and the GCOS fraction for all wavelengths is at least ~50% over land and ~60% over ocean.

**[Figure 2]**

**[Table 3]**

*Ångström Exponent*





AE was determined from AOD at two different wavelengths (AE $= \frac{\ln[\tau(\lambda_1)/\tau(\lambda_2)]}{\ln(\lambda_2/\lambda_1)}$). The accuracy of

AE decreases for low AOD because even a small spectral bias the AOD affects AE strongly (e.g., Wagner

and Silva, 2008). Therefore, the threshold of PARASOL AOD (550 nm) > 0.2 was used in AE validation.

For calculating the PARASOL AE (440/870), the AOD retrieved at 443 nm and 865 nm are interpolated to

nominal 440 nm and 870 nm wavelengths. Fig. 3 shows the scatter plots of PARASOL/GRASP AE against

AERONET AE (440/870) for the whole archive (Fig. 3a: for GRASP/Optimized, Fig. 3b: for GRASP/HP

and Fig. 3c: for GRASP/Models). GRASP/HP has a higher correlation R (0.845) than GRASP/Optimized

(0.800) and GRASP/Models (0.692). In addition, GRASP/HP shows a lower RMSE (0.334) than

GRASP/Optimized (0.356) and GRASP/Models (0.415). The statistics of separated land and ocean AE

validation are presented in Table 4. Over ocean, the correlation coefficients are significantly higher

(R>0.93) than over land for all three datasets. Overall, the AE correlation statistical metrics is the best for

GRASP/HP both over land and ocean. GRASP/Models product has the smallest BIAS over land, which is

counterpoised by overestimation of low and underestimation of high AE values due to assumed size

distributions in the aerosol model-based approach. Both GRASP/Optimized and GRASP/HP capture AE

well when large particles are dominant (AE<1.0), while the products tend to slightly underestimate AE

when small particles are dominant (AE>1.0).

**[Figure 3]**

                                             **[Table 4]**

*Fine- and Coarse- mode Aerosol Optical Depth*

        Fig. 4 shows the validation of PARASOL/GRASP AODF against SDA AODF provided by

AERONET. AERONET SDA products (O'Neill et al, 2003) reported only at 500 nm, therefore here were

interpolated to AODF at 550 nm based on AE using a quadratic fit in log-log space (Eck et al., 1999). Over

land + ocean, GRASP/HP AODF shows the best validation statistics with correlation R=0.925, BIAS=0.01

and Slope=0.892 compared to R=0.922, BIAS=0.02 and Slope=0.840 for GRASP/Optimized and R=0.867,

BIAS=-0.02, and Slope=0.662 for GRASP/Models. GRASP/Models AODF product has a slightly smaller

RMSE (0.092) than GRASP/HP (0.097) and GRASP/Optimized (0.099). Even though, the GCOS

requirement is initially defined for total AOD, here we also applied GCOS fraction on AODF validation

based on max ($\pm0.04, \pm0.1$AODF). The GCOS fraction for all AODF products is at least ~55% over land +

ocean. The GCOS fraction is highest for GRASP/Models (65.2%), which is dominant for low aerosol

loading cases (AODF<0.2). For moderate and high aerosol loadings (AODF>=0.2), GRASP/Optimized and

GRASP/HP show better performance than GRASP/Models, in terms of GCOS fraction and biases. The

linear regression slope for GRASP/Models is weakest 0.662 compared to 0.892 and 0.840 for GRASP/HP

and GRASP/Optimized respectively. These facts suggest a possible underestimation of fine-mode aerosol

in high AOD conditions for GRASP/Models. Caution is required in the interpretation of the regression

slope as these data may not meet the assumptions behind the technique; however, the results are useful in a

comparative sense. The statistics for separated land and ocean are presented in Table 5. As can be seen,

overall, PARASOL/GRASP AODF products show very good agreement with AERONET SDA products.

GRASP/HP AODF demonstrates best performance in terms of the highest correlation and smallest bias.

**[Figure 4]**

**[Table 5]**

The coarse-mode AOD (AODC) traditionally a difficult parameter to derive from satellite

observations especially over bright land surfaces, since nadir -looking satellite measurements are not very

sensitive to large particles. The validation of all archived PARASOL/GRASP AODC with AERONET

SDA AODC is presented in Fig. 5. Generally, the global (land + ocean) statistical metrics for AODC are

less convincing than that for AODF but still reasonable: GRASP/HP has higher correlation R (0.745) and

Slope (0.936) than GRASP/Optimized (R=0.689, Slope=0.748) and GRASP/Models (R=0.579,

Slope=0.657). GRASP/Models retrievals show a smaller bias (0.02) and RMSE (0.109) than

GRASP/Optimized (BIAS=0.04, RMSE=0.116) and GRASP/HP (BIAS=0.05, RMSE=0.123). The GCOS

fraction of AODC max ($\pm0.04, \pm0.1$AODC) for GRASP/Models (65.4%) is higher than GRASP/Optimized

(44.3%) and GRASP/HP (48.4%). In line with AODF, GRASP/Models has better performance for low

aerosol loading cases, which account for ~90% of the number of points. The statistics of separated land and
ocean AODC validation, presented in Table 6, show a much higher correlation of retrieved AODC with
AERONET over ocean. It is also interesting to note that the validation statistics for AODF seems to be
superior to that for AODC over land, and the situation is reversed. This can be explained by the fact that the
fine mode aerosols have higher abundance over land while coarse mode aerosol is dominant over ocean, i.e.
dynamic ranges are difference. Also, at longer wavelengths where the contribution of coarse particles to
radiation is significant, the land surface is very bright while ocean surface is practically dark. Over land
AODC in GRASP/HP and GRASP/Optimized products exhibit rather high BIAS of 0.05 and 0.03
correspondingly, that probably dominates the bias for the total AOD in both. For GRASP/Models product
biases in AODF and AODC over land have comparable magnitudes and different signs, and therefore
compensate each other in the total AOD.

**[Figure 5]**

**[Table 6]**

*Single Scattering Albedo*

Fig. 6 shows the validation of PARASOL/GRASP SSA (670 nm) with AERONET L2 inversion
products. The SSA products in AERONET L2 database provide the values only for moderate and high
AOD cases (AOD at 440 nm ≥ 0.4) to assure the highest quality of the inversion products (Dubovik et al.,
2000, 2002b). Following the same strategy, PARASOL/GRASP L2 and L3 products of SSA for low AOD
cases are also filtered out (Land: AOD 443 nm<0.3; Ocean: AOD 443 nm<0.02). The threshold for filtering
SSA over ocean is very low because using higher values would eliminate a significant fraction of the
retrievals. This low-AOD filtering is done over L2 products, and then L3 SSA is generated from filtered L2
products. The validation shows convincing correlation of all SSA PARASOL/GRASP products with those
from AERONET, although due to a rather small dynamic range (mostly 0.7-1.0) of SSA, the correlation
coefficients for SSA (670 nm) in Fig. 6 are notably lower than for other parameters. The highest correlation
is for GRASP/HP with R=0.536 and RMSE (0.056) compared with GRASP/Optimized (R=0.511;

RMSE=0.065) and GRASP/Models (R=0.324; RMSE=0.057), while GRASP/Models has a smallest BIAS

(-0.02) compared to GRASP/HP (BIAS=-0.03) and GRASP/Optimized (BIAS=-0.04).

Table 7 shows the statistics of PARASOL/GRASP spectral SSA (443, 670, 865, and 1020 nm) against AERONET SSA at four wavelengths (440, 675, 870, and 1020 nm). The statistics are given for combined land and ocean, because of the limited amount of validation points over ocean. The SSA correlation coefficients for GRASP/Optimized and GRASP/HP L3 products increase from 440 nm (~0.25)

to 1020 nm (~0.60), which is likely due to the increased dynamic range of SSA at longer wavelengths (e.g. see Dubovik et al., 2002b, SSA at 1020 can change from very low values for biomass burning aerosol to nearly unity for desert dust). Consequently, the RMSE also increases from 440 to 1020 nm.

In addition, Table 7 reports the statistics of SSA validation at different PARASOL AOD levels. The results clearly illustrate the improvement of retrieved SSA with the increase of aerosol abundance, in

ageement with the results of AERONET sensitivity studies by Dubovik et al., (2000). For example, the correlation coefficient for GRASP/Models SSA at 670 nm with AERONET significantly improves from 0.321 for all L3 products to 0.814 for AOD greater than 1.5.

**[Figure 6]**

**[Table 7]**

*Aerosol absorption optical depth*

Aerosol absorption optical depth (AAOD) is related to SSA and total AOD as:

$$AAOD(\lambda) = AOD(\lambda) \times [1 - SSA(\lambda)] \tag{5}$$

In the current PARASOL/GRASP L3 dataset, the AAOD value of each grid box (0.1° or 1° degree) is calculated based on Eq. (5) using average $AOD(\lambda)$ and $SSA(\lambda)$ of the grid box. Note that the

PARASOL/GRASP L3 $SSA(\lambda)$ values are aggregated based on moderate and high AOD cases (Land: AOD 443 nm≥0.3; Ocean: AOD 443 nm≥0.02), and again the very low threshold (≥0.02) for filtering SSA over ocean was chosen in order to retain sufficient number of SSA and AOD retrievals. Choosing even slightly higher values would eliminate the majority of retrieval over ocean. Thus, the direct use of L3 climatology





of AAOD may lead to overestimation of the global aerosol absorption, because the low AOD cases are

filtered. Similarly, the AERONET L2 database provides AAOD products only for moderate and high AOD

cases (AOD at 440 nm ≥ 0.4) to assure their highest quality (Dubovik et al., 2000).

The statistics of PARASOL/GRASP spectral AAOD (443, 670, 865 and 1020 nm) validation versus

AERONET AAOD (440, 675, 870, and 1020 nm) are shown in Table 8. The correlation coefficients of

AAOD are relatively low (0.4-0.55), which is certainly due to the low absolute value of AAOD, most cases

are less than 30% of total AOD. GRASP/HP and GRASP/Models AAOD products show the RMSE equal

to 0.042-0.018 from 443 nm to 1020 nm for Models, and 0.047-0.025 for HP. The BIAS is lowest for

PARASOL/Models AAOD: 0.00 at 440, 870 and 1020 nm and 0.01 at 670. Thus, PARASOL/GRASP

AAOD provide rather useful information about global AAOD values, even the uncertainties are rather

significant given the generally low magnitudes of AAOD. In contrast with SSA, the attempts to analyze the

AAOD accuracy for different AOD levels did not show any consistent improvement in accuracy with

increase of abundance.

**[Table 8]**

**3.2 Comparison of results obtained from validation of PARASOL and MODIS aerosol products
against AERONET**

In order to place the PARASOL/GRASP validation results into perspective, here we compare

PARASOL/GRASP ability to retrieve AOD, AE, AODF and AODC with other satellites. Specifically,

these products from MODIS, PARASOL/Operational and PARASOL/GRASP products are validated using

the same approach for the entire 2008 year and validation results were compared. MODIS aerosol products

have been extensively evaluated globally by the MODIS team in multiple studies (Gupta et al., 2018; Levy

et al., 2010, 2013, 2018; Lyapustin et al., 2018; Sayer et al., 2013, 2014, 2019) and PARASOL/Operational

aerosol products have been evaluated in Bréon et al. (2011); the present analyses is performed for reader

convenience and consistency of methodology across products. We confirmed that the statistic metrics that

we found for MODIS and PARASOL/Operational aerosol products validation in 2008 is similar to these

studies. This section is therefore focusing on a comprehensive evaluation of the consistencies and



differences between PARASOL and MODIS aerosol products using examples from one year. The year

2008 was chosen because it presents a generally good statistics of observations and all types of aerosol are

clearly present. The validation figures for the satellite products over land and ocean are presented

separately, because over land there are three MODIS products (DT, DB, and MAIAC), while only DT

product is provided over ocean. MAIAC products cover some land-containing ocean tiles, though as these

are spatially complete we do not consider MAIAC ocean products here. PARASOL/Operational AODF

products are provided over land and ocean, and total AOD products only over ocean.

Fig. 7 shows validation results for AOD at 550 nm for 3 PARASOL/GRASP products over land

with collocated AERONET measurements. Fig. 8 shows the validation results for MODIS DT, DB and

MAIAC AOD products over land. The products called MAIAC_0.1 and MAIAC_0.01 correspond to the

MODIS MAIAC original product aggregated to 0.1° and 0.01° grid boxes, respectively. In general MODIS

products have more matched points than PARASOL products due to MODIS' wider swath and higher

spatial resolution of measurements allowing better cloud detection. From low to high values the sequence

of obtained global correlation coefficients is: 0.870 (DB), 0.874 (MAIAC_0.01), 0.875

(GRASP/Optimized), 0.895 (MAIAC_0.1), 0.898 (DT), 0.908 (GRASP/HP) and 0.924 (GRASP/Models).

The GCOS fraction sequence is 28.8% (GRASP/Optimized), 32.4% (GRASP/HP), 46.1% (DT), 48.1%

(MAIAC_0.01), 48.8% (DB), 52.8% (MAIAC_0.1), and 53.2% (GRASP/Models). The high to low RMSE

sequence is: 0.157 (GRASP/HP), 0.150 (GRASP/Optimized), 0.126 (DB), 0.121 (GRASP/Models), 0.120

(DT), and 0.112 (MAIAC_0.1). The large to small total BIAS sequence is: 0.06 (GRASP/HP), 0.04

(GRASP/Optimized), -0.03 (MAIAC_0.01 and MAIAC_0.1), 0.02 (DT), -0.01 (DB) and 0.00

(GRASP/Models). The low to high sequence of regression slope values is: 0.780 (GRASP/Optimized),

0.793 (MAIAC_0.1), 0.794 (MAIAC_0.1), 0.841 (DB), 0.938 (GRASP/HP), 0.988 (DT), and 0.989

(GRASP/Models). The results illustrate that the overall accuracy of these AOD products are generally

comparable on a global scale. Note, however, that different products may have different regional strengths

and weaknesses (e.g. Sayer et al., 2014), motivating the mapped analysis later. The GRASP/Models AOD



yields overall the largest number of the best statistical indicators over land: with the highest correlation

(R=0.924), the highest GCOS fraction (53.2%), correlation slope (0.989) and the smallest total bias

(BIAS=0.00). The detailed statistics of PARASOL and MODIS AOD products against referenced

collocated AERONET AOD at 550 nm over land and ocean are presented in Table 9.

**[Figure 7]**

**[Figure 8]**

Fig. 9 presents validation of PARASOL/GRASP (Optimized, HP and Models),

PARASOL/Operational and MODIS/DT AOD products versus collocated AERONET measurements over

ocean in 2008. The detailed statistic metrics are presented in Table 9. The matching methodology is the

same as described in section 2.4. The total matched points in 2008 are ranging from minimum

GRASP/Optimized 116 to maximum MODIS/DT 218. In general, all AOD products show high correlation

over ocean, of which GRASP/Models has the highest R (0.963), following by Operational (0.954), DT

(0.952), GRASP/Optimized (0.950) and GRASP/HP (0.947). The high to low sequence of RMSE is 0.092

(GRASP/HP), 0.089 (GRASP/Optimized), 0.081 (DT), 0.077 (Operational) and 0.061 (GRASP/Models).

The slopes are quite similar: 1.165 (Operational), 1.145 (GRASP/Optimized), 1.074 (GRASP/HP), 0.965

(GRASP/Models) and 0.974 (DT). Overall, GRASP/Models show slightly better bias (0.02) and GCOS

fraction (62.9%), following by DT (BIAS=0.03, GCOS fraction=55.0%), Operational (BIAS=0.03, GCOS

fraction=52.2%), GRASP/Optimized (BIAS=0.06, GCOS fraction=42.2%) and GRASP/HP (BIAS=0.07,

GCOS fraction=26.6%). Altogether, GRASP/Models, PARASOL/Operational and DT AOD yield quite

similar performance over ocean with a better correlation statistics than GRASP/Optimized and GRASP/HP

AODs that correlate well with AERONET measurements but present a significant positive bias (0.06-0.07).

**[Figure 9]**

**[Table 9]**

In order to obtain more information about the quality of the retrieval products over different land

surfaces, the statistics of satellite validation against AERONET were also analyzed separately for different

land covers. Table 10 shows the statistic metrics for land surfaces with different Normalized Difference





Vegetation Index (NDVI). The statistics are presented for several categories: bare soil/desert surfaces (NDVI<0.2); mixture of bare soil and vegetated surfaces (0.2≤NDVI<0.4); surfaces covered different types of vegetation (0.4≤NDVI<0.6 and NDVI≥0.6). The global NDVI dataset is adopted from GRASP/Models L3 annual mean products for year 2008 (Fig. 10). The correlation metrics in Table 10 show that, in general,

all products show better performance over surface type with 0.2≤NDVI<0.6 than bright, bare surfaces (NDVI<0.2), and somewhat better than for dense vegetation surface (NDVI≥0.6). Overall the AOD product of GRASP/Models seems to show the best correlation with AERONET, with highest R over 3 of 4 surface classes. Over bright surfaces (NDVI<0.2), GRASP/HP has a highest R (0.915), but also rather high BIAS of 0.06. The GRASP/Models AOD also has zero BIAS for 3 surface classes except the dense vegetation

surface (NDVI≥0.6), where GRASP/Models AOD has total BIAS of 0.03, higher than that in any MODIS AODs.

**[Figure 10]**

**[Table 10]**

Fig. 11 shows the validation of AE for PARASOL/GRASP and MODIS DT and DB products over

land versus collocated AERONET measurements. The MODIS AE for DT and DB products were calculated based on 470 and 660 nm that are reported in both products; an equivalent for AERONET was calculated using AOD interpolated to 470 and 660 nm. PARASOL/GRASP products contain AOD at 440 and 870 nm, therefore AE (440/870) was directly used for validation PARASOL/GRASP results. MAIAC AE was not included because MAIAC reports AOD at two rather close wavelengths 470 and 550 nm and

calculation of AE using these such close channels could produce substantial uncertainties in AE. The threshold of satellite AOD (550 nm) > 0.2 was used in validation of AE over land and ocean. In general, PARASOL/GRASP (Optimized, HP and Models) AE correlate notably better with AERONET than MODIS (DT and DB), which is likely caused by the lower information content in regards to aerosol size in mono-viewing MODIS observations. In addition, both DT and DB algorithms rely on climatology for the

aerosol model selection, i.e. AE is rather predetermined than retrieved to some extent. For example, although AOD over land is reported by DT at 470 nm and 660 nm, the spectral dependence of the DT land

retrieval is mostly imposed by assumed aerosol models. The DT team makes a specific point of not reporting AE over land for that reason, and at best the spectral dependence might allow a binary inference of either fine mode or coarse mode dominated particles, but not a quantitative measure of the true spectral

dependence. The DT over ocean algorithm has greater flexibility in its mixing of models and does return a quantitative AE. The weaker performance of GRASP/Models approach compare to GRASP/HP and Optimized is caused by the limitation of maximum and minimum AE values allowed by the mixture of aerosol components used, even though GRASP/Models approach allows mixing of different components freely with no location specific constraints. As a result, GRASP/Models tends to overestimate AE for large

particles (low AE values), and underestimate AE for small particles (high AE values). Hence, GRASP/Models AE products are less appealing than those from GRASP/Optimized and GRASP/HP in terms of evaluation metrics. GRASP/HP tends to provide the best AE products over land.

**[Figure 11]**

**[Figure 12]**

**[Table 11]**

Fig. 12 presents the validation of AE over ocean from PARASOL/GRASP, PARASOL/Operational and MODIS/DT products against AERONET measurements. PARASOL/Operational AE was calculated based on 670 and 870 nm that are reported in the product, and AE (670/870) of AERONET was calculated using AOD interpolated to 670 and 870 nm. Although there are not many available points the satellite

derived AE over ocean are much better than that over land. GRASP/Models show R (0.949) higher than Operational (0.891), GRASP/HP (0.890), GRASP/Optimized (0.840) and DT (0.832). GRASP/Models and MODIS DT AE show an overestimation for large particles. Operational AE tends to overestimate both for large and small particles. At the same time, GRASP/HP AE correlation has the slope closer to 1:1 line with AERONET AE than other products, with the best linear fitting slope (0.810) and intercept (0.051). The

statistic metrics of AE validation over land and ocean are listed in Table 11.

AODF is often used to estimate anthropogenic aerosol climate effects (Bellouin et al., 2005) and surface air quality (e.g. $PM_{2.5}$) (Zhang and Li, 2015). MODIS started to report fine mode weighting





parameter (η) in the products from the second generation DT operational algorithm (Levy et al., 2007b), though η is weighted for reflectance not for AOD. Consequently η over land is a diagnostic that has little

physical meaning and the resulting AODF and AODC do not have physical meaning and generally are not recommended to be used. Therefore, it is not considered in the analysis. However, over ocean, based on single scattering approximation, η is also weighted for AOD (Remer et al., 2005). Therefore, MODIS fine and coarse mode AOD at 550 nm over ocean are derived according to the equations below:

$$AODF = AOD \times \eta \tag{6}$$

$$AODC = AOD \times (1.0 - \eta) \tag{7}$$

Fig. 13 shows the validation of AODF at 550 nm for PARASOL/GRASP and PARASOL/Operational AODF products over land against AERONET AODF products from the SDA algorithm. It's noticeable that the AODF products over land are only available from PARASOL MAP measurements. The results in Fig. 13 indicate the PARASOL/GRASP and PARASOL/Operational AODF

products are in good agreement with AERONET SDA products, for example, R>0.86. The GCOS fraction of AODF for PARASOL/GRASP and PARASOL/Operational products are at least 50%. GRASP/HP AODF shows the best correlation among all four AODF products over land, with rather similar performance for GRASP/Optimized AODF.

The AODF validation over ocean is shown in Fig. 14, and statistical metrics over land and ocean are

presented in Table 12. GRASP/Optimized and GRASP/HP AODF show generally consistent performance over ocean and over land, with correlation R around 0.9, while the BIAS for GRASP/Optimized and GRASP/HP AODF is higher over ocean than over land. At the same time, GRASP/Models AODF shows significant improvement over ocean, for example, the fitting line is much closer to 1:1 (dotted line), and the RMSE decreased dramatically. PARASOL/Operational AODF shows a slight decrease of R from land

(R=0.886) to ocean (R=0.780), also reported in Bréon et al. (2011), while the fitting line, RMSE and BIAS show improvement from land to ocean. This is likely due to higher information content about aerosols in satellite observations over dark ocean surfaces compared to brighter land surfaces.

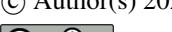


**[Figure 13]**

**[Figure 14]**

**[Table 12]**

The validation of AODC over land and ocean are shown in Figs. 15 and 16, respectively. Even though AODC products over land are only provided for PARASOL/GRASP, for completeness we present this over land for year 2008 in Fig. 15. Similarly to the results from the total PARASOL/GRASP archive, AODC over ocean is more accurate than over land. The overall best results of AODC are provided by

GRASP/HP with highest R (0.771) and the best linear fitting (slope is reaching 1 and intercept is close to 0) over land. Yet, the BIAS of GRASP/HP AODC is 0.05, which is higher than GRASP/Models (0.01) and GRASP/Optimized (0.03), which results in higher GCOS fraction for GRASP/Models AODC (63.7%) than GRASP/HP (45.8%) and GRASP/Optimized (45.6%). At the same time, as mentioned above, over dark ocean the sensitivity of the observed signal to aerosol is stronger allowing for retrieval of particle size

information that is more challenging over land. The GRASP/Models AODC shows the best R (0.966) and RMSE (0.040) while MODIS/DT AODC has the smallest bias (0.00) against AERONET over ocean, following by Operational (0.01), GRASP/Models (-0.01), GRASP/Optimized (0.03), and GRASP/HP (0.05). Although AODC is not included in PARASOL/Operational product list, over ocean we subtract AODF from total AOD to obtain Operational AODC, which shows a rather good agreement with

AERONET (R=0.936, Slope=0.971, RMSE=0.045, BIAS=0.01). However, over land, only AODF is provided in the PARASOL/Operational product.

**[Figure 15]**

**[Figure 16]**

**[Table 13]**

The statistics in comparison of each single product vary due to the differences in product coverage: coverage of MODIS/AQUA is wider than PARASOL; at the same time, the products have different limitations and availability:

- MODIS/DT has limited products over bright land surfaces, and AODF and AODC are only available over ocean;

- MODIS/DB and MAIAC AOD products are only over land, and do not include AODF and AODC;

- PARASOL/Operational over land provides only AODF;

- Quality screening is different (even between PARASOL/GRASP products).

Therefore, the approaches chosen in this paper for considering all above factors could have some effects on

the results and their interpretation. At the same time, the correlations with AERONET obtained in these studies for known products including MODIS DT, DB and MAIAC and PARASOL/Operational in general agree with the results of previously mentioned studies.

### 3.3 Evaluation of PARASOL and MODIS validation results over different AERONET sites

In this section, we compare the validation metrics of PARASOL/GRASP and MODIS aerosol

products over spatially distributed AERONET sites. PARASOL/Operational AOD products are provided over ocean only, hence are not included in this section. The AOD validation was conducted over all AERONET sites that had available data in 2008. At the same time, and to increase statistical robustness only sites with at least 10 matchup points were included in the analysis. However, the different products can also have different number of matchup points over different AERONET sites due to various factors (as

discussed previously). Therefore, to evaluate the validation performance of different products, the percentage (%) of the cases when the product of each algorithm showed the best statistic metrics, observed among all the products (e.g. the highest R, GCOS Fraction, and the lowest RMSE, BIAS, etc.) was used as an indicator for the performance evaluation.

Fig. 17 shows the percentage score for each algorithm at AERONET sites for statistical metrics R,

RMSE, BIAS and GCOS Fraction respectively. The detailed statistics for the performance of each AOD products is shown in Fig. 18 (only the 1st ranking statistics over each site are present in the maps.). All PARASOL/GRASP products have fewer sites with at least 10 matchup points than MODIS AOD products.



There are 102, 124, and 95 sites having sufficient matchup points for GRASP/Optimized, GRASP/HP and

GRASP/Models respectively, lower than DT (153), DB (172), MAIAC_0.1 (169) and MAIAC_0.01 (172)

by 20%~45%. Regarding the correlation coefficient R, GRASP/Models, DT and MAIAC_0.1 are the 3

algorithms showing higher scores for 37.9%, 28.1% and 24.9% sites where respective products were

provided. As shown in Fig. 18a, these 3 algorithms show good performance worldwide, e.g. North America,

Europe and East Asia. There are no MODIS AOD products showing the best R over Australia (only 4 sites

are available there). The 3 GRASP algorithms show high percentage for products over dust and biomass

burning regions, e.g. South America, southern Africa, central Africa, central Australia, etc. At the same

time, the GRASP/Optimized and GRASP/HP AOD products performed less well over North America.

**[Figure 17]**

**[Figure 18]**

In terms of the percentage of sites with the best RMSE, GRASP/Models and MAIAC_0.1 are the

top 2 algorithms showing the best RMSE results over 60.0% and 33.1% of AERONET sites with available

GRASP/Models and MAIAC_0.1 retrievals. Thus, overall, all these results indicate that GRASP/Models

presents a strong ability to provide AOD that agrees well with AERONET measurements. Both

GRASP/Models and MAIAC_0.1 show the best RMSE over Europe and North America (Fig. 18b), which

also have the highest density of AERONET sites. GRASP/Models shows the best BIAS over 36.8% sites,

followed by DB (27.3%) and MAIAC_0.01 (23.3%). For the best GCOS Fraction, GRASP/Models leads

this for 57.9% over its total 95 sites. After, MAIAC_0.1 has the highest GCOS Fraction for 30.8% over

total 169 sites. In Fig. 18d, the best GRASP/Models sites are globally distributed. Over the Eastern United

States, DB and MAIAC_0.1 products tend to have more sites with the best GCOS Fraction.

Using a similar concept as the AOD analysis above, the PARASOL/GRASP and MODIS (DT and

DB) AE validation over AERONET sites were compared. Only cases with satellite AOD (550 nm) > 0.2

were included in the analysis; due to the reduced data volume from this threshold, the requirement on

minimum matchups was reduced to 5. Fig. 19 shows the detailed statistics for the performance of each AE

products. Fig. 20 shows the best performing algorithm at each site according to R and RMSE respectively.



In general, GRASP/HP and GRASP/Optimized AE products outperform the other AE products in the site

level validation. The best sites are globally distributed (see Fig. 20). There are 44.1%, 38.6% and 34.7%

sites showing the best R for GRASP/Optimized, GRASP/HP and GRASP/Models, somewhat higher than

DT (12.0%) and DB (8.2%). GRASP/HP AE has the best RMSE over 43.0% AERONET sites, higher than

GRASP/Optimized (34.3%), DB (28.6%), GRASP/Models (24.5%) and DT (17.0%).

**[Figure 19]**

**[Figure 20]**

### 4 Inter-comparison of satellite products at global scale

This section presents the inter-comparison of different satellite products for the year 2008 data on a

global scale, i.e. not only over AERONET sites. Specifically, we want to know if the consistency of the

satellite products remains the same in the areas where no AERONET observations are available. In the first

part, we compare PARASOL/GRASP and PARASOL/Operational at a spatial resolution of 0.2° x 0.2°,

which represents a compromise between PARASOL/Operational (18.5 km) and PARASOL/GRASP Level

3 (0.1°) resolutions. In the second part of this section, the global inter-comparison is done between

PARASOL/GRASP and MODIS aerosol products in a spatial resolution of 0.1° x 0.1°, close to the DT and

DB product native resolution of 10 km, and only use MAIAC_0.1 data that is of similar resolution. Only

GRASP/HP and GRASP/Models products for PARASOL/GRASP are used in the consideration of this

section since GRASP/Optimized shows rather similar results to GRASP/HP. Since the focus of this section

is global pixel-to-pixel comparison of satellite aerosol products, we use all available data of the highest

quality for each dataset (Table 2).

### 4.1 Comparisons between PARASOL/GRASP and PARASOL/Operational aerosol products

To begin, we investigate two independent aerosol products derived from PARASOL measurements,

PARASOL/GRASP and PARASOL/Operational, globally for 2008. As mentioned above,

PARASOL/Operational provides only AODF over land, while over ocean AOD, AE and AODF are

available. We subtract AODF from total AOD to obtain Operational AODC over ocean.

Table 14 presents the pixel-to-pixel statistic metrics (R, Slope, Intercept, RMSE and BIAS) between

Operational and GRASP aerosol products (note here the BIAS should be interpreted as an offset rather than

true bias as the "truth" is unknown; we retain the name of the metric for consistency with the earlier

analysis). We took Operational products as a reference as these were the original PARASOL aerosol

products released by AERIS/ICARE; hence, the BIAS is defined as GRASP - Operational. All the statistics

for AOD, AODF and AODC are given for the midvisible wavelength (550 nm), while AE is calculated

based on 670 and 870 nm. The statistical metrics are reported both for global comparisons and over

AERONET pixels only (the numbers in the brackets). It can be seen from Table 14 the global comparison

between PARASOL/GRASP and PARASOL/Operational is rather consistent for AOD over ocean and

AODF over land, for which, the global pixel-to-pixel correlations between GRASP/HP, GRASP/Models

and Operational products are generally higher than 0.85 based on more than 5 million pairs. However, the

agreement of AODF over ocean decreases to 0.63-0.73 for R. The slight decreasing of correlation against

AERONET from land to ocean for Operational AODF products is also recorded in Table 12 and previous

study by Bréon et al. (2011). The AODC over ocean for the Operational product is derived from AOD and

AODF, hence, the number of matched pairs is lower than for AODF. The overall agreement has a

correlation coefficient of  ~0.7. GRASP/HP AODC is ~0.05 higher than Operational, but the difference

between GRASP/Models and Operational is ~0.0, which are in line with the validation against AERONET

in Table 13. The pixel-to-pixel agreement for PARASOL/GRASP and PARASOL/Operational AE is less

convincing (R<0.6) than any other parameters, even though they are all well correlated with AERONET

(R>0.8) over ocean. One possible reason is that the AE here is calculated at different wavelengths (670 and

870 nm) than for the comparisons with AERONET (470/660 nm and 440/870 nm). Besides, the increase of

AE agreement for global correlation (R) compared to that over AERONET pixels is more notable than

other parameters. This may explain that the AE products resulting from LUT-based algorithms are more

determined by climatological assumptions about the aerosol models than retrieved.

**[Table 14]**

**4.2 AOD comparisons between PARASOL/GRASP and MODIS products**

In order to further clarify the level of consistency of satellite products (PARASOL/GRASP and MODIS), the global correlations of different satellite products were extensively analyzed for the year 2008 at a spatial resolution of 0.1° x 0.1°. Fig. 21 shows the seasonal pattern of AOD (550 nm) from PARASOL (GRASP/HP and GRASP/Models) and MODIS (DT, DB, and MAIAC) products. Any grid box with less than 3 measurements for a season was omitted. Fig. 22 shows the differences of AOD (550 nm) by season

between PARASOL and MODIS aerosol products using GRASP/Models as the reference. A positive value indicates that the MODIS product had a higher mean value. Note that Fig. 22 is not a simple difference of the seasonal means shown in Fig. 21.    Instead, to decrease sampling-related differences, a difference between the products was calculated at the pixel level, and these pixel-to-pixel differences were then averaged for a season. In addition, we require at least three matched points in a season to be plotted on the

map. Since the analysis in Section 3 suggested that the AOD products over land and ocean from the GRASP/Models processing have the lowest biases, this was used as a reference product in Fig. 22. It should be noted that in order to show the intrinsic difference between the products, the overall bias from AERONET values (using validation metrics in Table 9) were subtracted from the AOD products before obtaining the seasonal differences shown in Fig. 22.

In addition, the global correlations between different satellite products and GRASP/Models data at 550 nm were calculated for the complete year 2008. Also, in order to evaluate the consistency of different MODIS products over land, the inter-comparisons were done against MAIAC AOD (Land) product chosen as a reference, as MAIAC provides the most universal coverage over land. Table 15 presents the pixel-to-pixel statistic metrics (R, Slope, Intercept, RMSE and BIAS) between AOD products compared to the

reference of GRASP/Models (Land and Ocean) and MAIAC AOD (Land) products. The statistical metrics are reported both for global comparisons and over AERONET pixels only (numbers in brackets).

[Figure 21]

[Figure 22]

[Table 15]





Each of these global correlations was based on several dozens of millions of pairs, and less noisy

compared to the AERONET correlations (based on only a few thousand points). In spite of this significant

difference in volume, the outcome of the global satellite comparisons is rather consistent with the results of

validation against AERONET. For example, all AOD products are in close agreement over ocean, with the

correlation coefficients above 0.9 and slope lines close to 1:1 (Table 15). Specifically, the three aerosol

products (GRASP/HP, GRASP/Models and DT) over ocean agree with R>0.92 for any two products. Also,

in line with the validation over AERONET sites, GRASP/HP AOD (550 nm) consistently has a positive

offset ~0.05-0.16 from low to high AOD conditions with respect to GRASP/Models. DT and

GRASP/Models AOD show good agreement over ocean, R=0.92 for all points and R=0.97 for AERONET

pixels, in addition, the BIAS (DT–GRASP/Models) equals to -0.01 for all points and 0.00 for low AOD

(<0.2), while the negative BIAS of -0.06 appears when AOD is greater than 0.7. Statistics over ocean rely

on ~65 million pairs between GRASP/HP and GRASP/Models, and ~32 million pairs between DT and

GRASP/Models.

        However, over land surfaces the situation is quite different, and MODIS/MAIAC and DB AOD

products show evidently better agreement with GRASP/Models over AERONET pixels than the rest of

globe. The correlations over AERONET pixels both for MAIAC versus GRASP/Models and DB versus

GRASP/Models are of ~0.89 that is generally in line with the correlation coefficient values with

AERONET shown in Table 9. In a contrast, the corresponding correlation coefficients decrease to 0.76 and

0.77 for global statistics. The other statistical parameters (e.g. Slope, Offset, RMSE and BIAS) showed the

same trend, indicating a better agreement over AERONET pixels. For comparisons of GRASP/HP and DT

versus GRASP/Models AOD such tendency is not evident. Even though, the correlation coefficient drops

from 0.90 over AERONET to 0.85 globally, the rest of statistical indicators do not show significant

changes, whether over an AERONET site or elsewhere. It is interesting to note that MODIS products show

better agreement (especially in correlations) with other MODIS products over AERONET stations and

globally than between PARASOL and MODIS products over AERONET stations and globally (Table 15).

This phenomenon can be explained by several factors. First, the inputs from the two satellites differ significantly. The multi-angle polarization information from PARASOL offers algorithms many more degrees of freedom from which to constrain environmental factors and invert aerosol parameters than does a single view radiometer like MODIS. Second, because of this extra information the PARASOL/GRASP retrievals do not have location specific assumption about aerosol and conduct their retrievals in the exactly

the same manner globally. In contrast, all three MODIS retrievals use some regional assumptions over land about aerosol types, surface properties, etc. Even though each algorithm's assumptions are different, the need for a priori constraints could draw the MODIS products closer together. Therefore, the similarities in global performance of three algorithms can probably be explained by somewhat similar a priori assumptions about aerosol types, etc. used in MODIS algorithms. Third, as can be seen from the Table 15

and Fig. 21, GRASP/Models, GRASP/HP and MAIAC have wider coverage over land than DB and DT, because of the lack of retrievals over bright surfaces for DT and reduced number of retrievals over dark vegetation for DB (although some of this was improved in DB Collection 6.1; Sayer et al., 2019). Specifically, for the year 2008, there are more than 64 millions of pairs MAIAC/GRASP/Models AOD over land, which is much higher than the number of pairs obtained with other two AOD products. Thus, the

collocation statistics for MAIAC/GRASP, DT/GRASP, DB/GRASP as well as MAIAC/DT and MAIAC/DB were based on different data sets. Fourth, the different representation of various natural conditions in the global statistics and statistics over AERONET can be non-identical and, therefore, the average performance indications can differ. For example, there is only a certain fraction of AERONET sites in desert areas while land cover with bright surface may have notably higher or lower fraction in global

statistics. Correspondingly, if the product agreement is non-identical over different land surfaces, then the statistics with different representations of various surfaces can differ.

In order to explore the last factor, the statistics of the comparisons were sorted by land surface type. The Tables 16 and 17 show pixel-to-pixel statistic metrics with reference AOD from GRASP/Models and

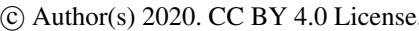

MAIAC respectively, over different land coverage using four classes of land surface by NDVI (as before,

NDVI<0.2, 0.2≤NDVI<0.4, 0.4≤NDVI<0.6, and NDVI≥0.6).

**[Table 16]**

**[Table 17]**

Tables 16 and 17 show that over very bright land surfaces (NDVI < 0.2), the global correlations

between MODIS (especially DB and MAIAC) with PARASOL/Models products were significantly lower

than over other surfaces and showed a most notable drop (>0.1) in global correlations compared to the

correlation over AERONET sites. Such a large drop was not seen between different PARASOL products or

between different MODIS products. Therefore, these differences are likely related to the fact that MODIS

retrievals rely on regional climatological aerosol assumptions or surface assumptions derived from

atmospheric correction at (unevenly-distributed) AERONET sites while in PARASOL/GRASP retrievals

no location specific assumptions are used. Another issue maybe related is that MODIS has much higher

spatial resolution for cloud detection than PARASOL. The possible sub-pixel cloud contamination for

PARASOL may affect the global inter-comparison statistics, since the validation against AERONET brings

additional cloud clearing filter from AERONET. As a result, PARASOL/GRASP retrievals are expected to

be rather consistent globally, while MODIS retrievals are more closely tied to AERONET statistics and

may perform less well in the areas with a lack of AERONET sites. At the same time, the fraction of pairs

over bright surfaces in inter-satellite product comparisons is higher than in AERONET statistics since there

are only a limited number of AERONET sites in desert areas. This latter statement does not necessarily

apply to MODIS DT because it often does not retrieve over deserts; however, although the sample size is

very small, Table 16 shows that it actually matches GRASP/Models less well at AERONET sites than

globally for NDVI < 0.2.

Interestingly, the maps in Fig. 22 of seasonal AOD difference indicate lower AOD (550 nm) for

PARASOL/Models over bright surfaces compared to MODIS products, while the global comparisons of

PARASOL/Models and MODIS DB and MAIAC products did not show significant BIAS in AOD (550

nm). At the same time, the global comparisons (Table 16) between PARASOL/Models and MODIS DB,





MAIAC show a significant BIAS for different ranges of AODs. MODIS DB and MAIAC had a positive

BIAS of ~0.06-0.04 for the situation with lower aerosol loadings (AOD 550 nm<0.2) and a notable

negative BIAS (0.02~0.06) for moderate (0.2 < AOD 550 nm < 0.7) and especially large for high aerosol

loadings (AOD 550 nm > 0.7) that reached ~0.3. A very similar tendency can be seen from the statistics of

validation against AERONET in Table 10: both PARASOL/Models and MODIS/DB have very small BIAS

of -0.01, while the distribution of BIAS is quite different for the situations with different loadings: 0.01 and

0.03 for low AODs, -0.03 and -0.05 for moderate AODs and for high AOD 0.01 and -0.16. This suggests

that the observed positive differences when MODIS/DB and MAIAC show higher AOD over bright

surfaces occurs mainly during low AOD conditions. This conclusion is supported by the fact that seasonal

means from all products do not show high AOD over the northern Sahara between 20°N and 30°N latitude.

Also, both DB and MAIAIC show significant underestimation of AOD over the Taklamakan desert where

seasonal mean AOD retrieved by PARASOL is high, which agrees with the negative offset between

MODIS DB, MAIAC and PARASOL/Models products over bright and bare soil land surfaces (NDVI<0.4).

      The negative BIAS between MODIS and PARASOL products is clearly seen on the maps of

seasonal AOD from different products for African biomass burning events. The results of correlation

analysis over green vegetation (NDVI ≥ 0.6) in Table 16 also show a significant negative BIAS in all

MODIS products compared to PARASOL/Models over green vegetation that increases for medium and

high aerosol loading. The validation against AERONET in Table 10 shows the highest BIAS of 0.06 to

0.07 for PARASOL/Models is over green vegetation (NDVI ≥ 0.6) when 0.2 < AOD < 0.7, while the

MODIS products tend to be less biased (DT BIAS = 0.03) or negatively biased (MAIAC BIAS = -0.04 to -

0.06, and DB BIAS = -0.04) for this surface type and AOD range. This pattern continues for DB and

MAIAC through all the vegetated surfaces with NDVI > 0.2. MODIS DB and MAIAC continue to be more

negatively biased against AERONET for moderate to high aerosol loading than PARASOL/Models is.

Thus, the results suggest that observed differences for African biomass burning events can be explained by

two potential reasons: a combination of overestimations of AOD by PARASOL/GRASP retrievals and

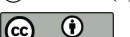

underestimation of AOD by MODIS products for cases of moderate to high aerosol loading. However, the

DT retrievals also show this negative bias against PARASOL/GRASP in the African biomass burning

(Figure 22), but do not follow the same trends against MODIS as DB and MAIAC. Other factors, such as

differences in cloud-screening, data amount, aggregation and quality screening approaches must also

contribute to these differences and need to be investigated in future analysis.

**4.3 AE comparisons between PARASOL/GRASP and MODIS products**

The seasonal pattern of AE from PARASOL (GRASP/HP and GRASP/Models) and MODIS (DT

and DB) products is presented in Fig. 23, as well as, AE differences by season between PARASOL and

MODIS aerosol products in Fig. 24. Table 18 shows the global pixel-to-pixel statistic metrics between AE

products based on references of GRASP/HP; in the brackets, the values corresponding to validation results

over AERONET pixels only. As before, the statistic metrics split into four classes of land surface by NDVI

are presented in Table 19. The GRASP/HP AE products are chosen to be a reference taking into account

the highest obtained correlation in the validation with AERONET in the Section 3. Again, note that

although AOD over land is reported by DT at 470 nm and 660 nm, the spectral dependence of the DT land

retrieval is mostly imposed by assumed aerosol models, and thus DT AE over land is at most a binary

indication of fine and coarse particles, and not a quantitative parameter. We expect no correlation with

GRASP/HP over land. AE over land from DB is similarly prescribed, not retrieved, when AOD < 0.2 (Hsu

et al., 2013). On the other hand, the DT AE over ocean is a true quantitative measure.

**[Figure 23]**

**[Figure 24]**

**[Table 18]**

**[Table 19]**

The differences between PARASOL/GRASP and MODIS DT and DB AE products are pronounced

in all comparisons. From Fig. 23, the seasonal variations for DB and DT are minor, which likely implies

utilization of similar climatological information in the DB and DT algorithms. Even though the differences

for GRASP/Models and GRASP/HP shown in Fig. 24 are not small (mainly due to the limited dynamic range of aerosol components used in the GRASP/Models approach), the overall pixel-to-pixel correlation between GRASP/Models and GRASP/HP is the highest between any two products (0.70 over land, 0.74 over ocean). The correlations for AE over land between MODIS DT and DB AE versus GRASP/HP are lower than 0.5 for all land surface types (Table 19), which is not surprising for the aforementioned reasons.

Over ocean, all available products (GRASP/HP, GRASP/Models and DT) show good agreement with AERONET measurements, with R>0.8 (Fig. 11 and Table 11), however, the pixel-to-pixel correlation between DT and GRASP/HP for ocean pixels globally decreases to 0.46. The cause of the drop in correlation for global statistics is presently unknown. It could be due to assumptions in the DT retrieval, but could also be linked to differences in calibration between POLDER and MODIS, as AE is particularly

sensitive to nuanced spectral changes in calibration in the lower-AOD conditions often seen over ocean.

## 4.4 AODF and AODC comparisons between PARASOL/GRASP and MODIS products

This section compares AODF and AODC at 550 nm from PARASOL/GRASP (GRASP/HP and GRASP/Models) and MODIS DT algorithms. As discussed earlier, the quantitative fine mode fraction (η) provided by the DT algorithm can be used to derive AODF and AODC only over ocean. Therefore, the

comparison of AODF and AODC over land is between GRASP/HP and GRASP/Models. The seasonal distribution of AODF and AODC are shown in Fig. 25 and Fig. 27 respectively. The seasonal differences between GRASP/Models, DT and GRASP/HP are shown in Fig. 26 (AODF) and Fig. 28 (AODC). GRASP/Models AODF is higher than GRASP/HP over dust source and downwind regions, while it is lower than GRASP/HP over biomass burning and urban areas, which is consistent with the validation

versus AERONET measurements in Figs. 13-16.

**[Figure 25]**

**[Figure 26]**

**[Figure 27]**

**[Figure 28]**



Globally, GRASP/Models and GRASP/HP AODF show a consistent agreement over land (R=0.87)

and ocean (R=0.89), as presented in Table 20. MODIS/DT AODF and AODC over ocean have good

agreement with GRASP/HP with R 0.86 and 0.84 respectively. GRASP/Models AODC shows a better

agreement with GRASP/HP over ocean than over land, while differences are less pronounced, R of 0.89

and 0.71, respectively. As was mentioned above, this tendency can be explained by a stronger sensitivity of

the observed signal to aerosol over dark ocean surface. Another interesting tendency is that correlations for

AODF over land are generally higher than for AODC, while over ocean the situation is inverse and the

correlations are higher for AODC, especially over AERONET. This can probably be explained by the two

facts that dominating oceanic aerosol has a pronounced coarse mode and that at the longer wavelengths,

where the contribution of coarse mode is the strongest, the ocean is practically dark. The land reflectance is,

however, higher than ocean at long wavelengths, even for relatively dark vegetated surfaces. The statistics

of pixel-to-pixel comparison (GRASP/HP and GRASP/Models) over different land surface types, as

discriminated by different NDVI categories, are also reported in Table 22 (AODF) and Table 23 (AODC).

**[Table 20]**

**[Table 21]**

**[Table 22]**

**[Table 23]**

In conclusion, the differences in more detailed aerosol characteristics including AE, AODF and

AODC (Tables 18-23) derived from PARASOL and MODIS are pronounced over both land and ocean.

This is in contrast to the results for the total AOD from PARASOL and MODIS, which are close over

ocean and in a reasonable agreement over land. This conclusion can likely be generalised by the fact that

retrieval accuracy of detailed aerosol properties is expected to be significantly higher from MAP products

than from mono-viewing photometric imagery.



**5 Summary and conclusions**

The new PARASOL/GRASP products were extensively evaluated using validations against
AERONET and comparisons with the original POLDER algorithm (PARASOL/Operational), and MODIS
Collection 6 aerosol products. The study was focused on the main aerosol parameters AOD, AE, AODF,
AODC, SSA and AAOD included in all PARASOL/GRASP products. Level 3 data quality filtered and
aggregated to 0.1 degree spatial resolution were used. The validation of PARASOL/GRASP spectral
products (443 – 1020 nm) was done for the full PARASOL archive (2005-2013) against all available
AERONET products. In addition to the direct validation of the full archive of PARASOL satellite products,
the PARASOL/GRASP products were intensively inter-compared with the widely used MODIS/AQUA
aerosol products from DT, DB and MAIAC (land only) algorithms and PARASOL/Operational aerosol
products for one full year, 2008, at 0.1 degree (~10 km) resolution. A global comparison with AERONET
for the year 2008 was performed for all products and the results inter-compared. The percentage of the
cases when the product of each algorithm showed the best statistical metrics among all the products was
used as an indicator for the performance evaluation. In addition, in order to further clarify the level of
consistency of the satellite products, the comparisons of seasonal means as well as the global correlations
of different satellite products at 0.1 degree or 0.2 degree were comprehensively analyzed for the year 2008.
In terms of data volume and geographic extent, the global comparisons analyses are more representative of
the global aerosol system than the subset based on colocations with AERONET.

The results show that the PARASOL/GRASP retrieval provided reliable aerosol products, and
important advancement over the reference MODIS aerosol products:

- *Total AOD*

- the PARASOL spectral products including AOD for six wavelengths in the range 443 to 1020 nm
  agree well with AERONET AOD measurements, e.g. for PARASOL/Models AOD correlation
  coefficients R are ≥ 0.86 over land and ≥ 0.94 over ocean with BIAS not exceeding 0.01 over land
  and 0.02 over ocean for all wavelengths;



- the AOD (550 nm) products from PARASOL/GRASP (especially GRASP/Models) correlate with AERONET generally similar or better than the correlations of MODIS AOD (550 nm) results both over ocean and land:

  - over ocean: all PARASOL (including Operational) and MODIS DT algorithms provide comparable and well correlated retrieval results;

  - over land: PARASOL/GRASP provides full land coverage products that correlate generally better with AERONET; MAIAC shows the highest percentage falling with the GCOS criteria and lowest RMSE among MODIS products, but greater overall BIAS than either DT or DB.

- the correlation between different PARASOL/GRASP products obtained only over AERONET sites and globally are rather consistent, while the correlations between PARASOL and MODIS products for global analysis over land notably degrade compared for those obtained only over AERONET sites, especially for MAIAC and DB. This finding suggests possible dependence of MODIS retrievals on AERONET regional assumptions of aerosol types or AERONET-assisted atmospheric correction to determine surface reflectance, while GRASP retrievals do not use any location specific aerosol or surface assumptions.

- *AE:*

- the PARASOL products agree with AERONET generally similar to the MODIS DT product over ocean and significantly better over land;

- all PARASOL/GRASP products (Optimized, HP and Models) provide AE values globally over land and ocean that agree between themselves consistently over AERONET sites and globally;

- *AODF and AODC:*

- all PARASOL/GRASP products (Optimized, HP and Models) provide spectral AODF and AODC values globally over ocean and all land covers including bright surfaces, and the different products agree between themselves consistently over AERONET sites and globally;



- the PARASOL/GRASP uniquely provides AODF and AODC with global coverage; PARASOL/Operational provides only AODF over land, while MODIS AODF and AODC products are only available over ocean;

- the PARASOL/GRASP AODF and AODC products agree with AERONET as well as MODIS (and PARASOL/Operational) and somewhat better over ocean;

- *Aerosol absorption:*

- all PARASOL/GRASP products (Optimized, HP and Models) provide SSA and AAOD spectral values that are generally not accessible from MODIS and other satellite products;

- the validation of PARASOL/GRASP shows robust correlation of the retrieved SSA and AAOD spectral values with AERONET (440–1020 nm), correlations increase for retrievals corresponding to the events with higher AOD. For AAOD retrievals overall the BIAS does not exceed 0.01, suggesting that PARASOL products can be used for making global estimations of AAOD at such level of uncertainty.

Analysis presented in this paper suggests that the data from PARASOL, and therefore from multi-angle polarimeters (MAP) in general, allow not only solid retrievals of conventional aerosol products (e.g. AOD at 550 nm), but also detailed aerosol properties such as AOD for the whole spectrum of observations (e.g. for PARASOL from 443 to 1020 nm), and aerosol SSA and AAOD that are practically not accessible from mono- and bi- viewing photometric satellite observations, as well as improved AE, AODF, and AODC at a global scale. It is also important to emphasize that PARASOL/GRASP retrievals are based on rigorous optimized inversion that searches for statistically optimized fitting in a continuous space of solution without using widely used Look-up-Tables. As a result, it provides a globally-consistent product using exactly the same aerosol modeling approach over land and ocean, unique set of a priori constraints and initial guess, while retrieving surface reflectance properties simultaneously with aerosol. It is expected that similar type of approaches will become more common and evolve further in the coming era of multiple MAP instruments, e.g. 3MI, DPC, Aerosol-UA, SPEXone and HARP2, etc. (see more in Dubovik et al.,

2019). The multi-dimensional aerosol information derived from MAPs is expected to improve quality and utility of atmospheric aerosol characterization from space.

One key finding of this work is that the best retrieval of total AOD is provided by the GRASP/Models approach, which restrains the retrieval to a priori aerosol model components, vastly reducing the number of free parameters for retrieval. The more complex GRASP/HP retrieval with many more retrieval parameters seemed to offer more accurate detailed aerosol parameters such as AE, AODF, AODF and SSA. Future efforts on improving the GRASP retrieval will be aimed at achieving accurate

retrievals within one approach. However, this situation also reveals the challenge of a developing unique approach that can provide a retrieval of all parameters with highest accuracy from MAP observations. Indeed, multi-angular polarimetric observations have sensitivity to different aerosol properties, and therefore the MAP algorithms tend to be designed for the retrieval of large number of parameters, while in the situations with low aerosol presence the information may be not sufficient to retrieve all parameters

reliably. Nonetheless, the presented results demonstrate an overall clear advantages of MAP aerosol retrievals compare with photometric mono-viewing product and support high expectations from future MAP missions with improved instrumental and algorithmic developments.

**Data availability**

The PARASOL/GRASP Optimized, HP and Models products are publicly available the official GRASP

algorithm website (https://www.grasp-open.com/products) and the AERIS/ICARE Data and Services Center (http://www.icare.univ-lille.fr). The dataset used in the current study is registered under: http://doi.org/10.5281/zenodo.3887265 (Chen et al., 2020).

**Author contribution**

The GRASP aerosol products evaluation exercise has been implemented and investigated by the GRASP

team (OD, DF, PL, TL AL, CC). CC and OD carried out this study and analysis. The results were widely

discussed with POLDER/Operational and MODIS (DT, DB and MAIAC) aerosol team, who are co-authors of this paper. CC and OD wrote the manuscript with contributions from all authors. FD, JD, MA, LB, DM, AH and CF carried through the POLDER data processing based on the GRASP-OPEN software.

**Competing interests**

The authors declare that they have no conflict of interest.

**Acknowledgements**

The authors would like to acknowledge the use of POLDER data, POLDER/PARASOL Level-1 data originally provided by CNES and AERIS/ICARE Data and Services Center (http://www.icare.univ-lille.fr/). The PARASOL/GRASP results are generated by Laboratoire d'Optique Atmosphérique and Cloudflight Austria GmbH with the GRASP-OPEN software (https://www.grasp-open.com). We would like to thank the AERONET team and all PIs of AERONET stations for maintaining the instrument and making the data available for the community. The authors are also grateful to the MODIS aerosol team, especially DT, DB and MAIAC groups, for providing the data used in this study. We would like to acknowledge to the CaPPA (Chemical and Physical Properties of the Atmosphere) project funded by ANR (ANR-ll-LABX-0005-01), AERIS/ICARE Data and Services Center and GRASP-SAS for data.

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





**Table 1. List of aerosol parameters in PARASOL/GRASP products**

| Parameters | Dimension | Description |
|---|---|---|
| AOD | (λ, Latitude, Longitude) | Aerosol optical depth |
| AExp | (Latitude, Longitude) | AE (670/865 nm) |
| AODF | (λ, Latitude, Longitude) | Fine mode AOD |
| AODC | (λ, Latitude, Longitude) | Coarse mode AOD |
| AAOD | (λ, Latitude, Longitude) | Aerosol absorption optical depth |
| SSA | (λ, Latitude, Longitude) | Single scattering albedo |
| RealRefIndSpect | (λ, Latitude, Longitude) | Real part of refractive index |
| ImagRefIndSpect | (λ, Latitude, Longitude) | Imaginary part of refractive index |
| SizeDistrLogNormBin | (5, Latitude, Longitude) | 5 Bins of size distribution |
| SphereFraction | (Latitude, Longitude) | Sphere fraction |
| VertProfileHeight | (Latitude, Longitude) | Aerosol scale height (unit: m) |
| LandPercentage | (Latitude, Longitude) | Land percentage (%) |
| ResidualRelative | (Latitude, Longitude) | Relative residual |

λ = 443, 490, 565, 670, 865, and 1020 nm


**Table 2. Strategies used to select quality assured PARASOL and MODIS aerosol products**

| | | Land | Ocean |
|---|---|---|---|
| POLDER | GRASP/Optimized | "ResidualRelative"<0.05 | "ResidualRelative"<0.1 |
| | GRASP/HP | "ResidualRelative"<0.05 | "ResidualRelative"<0.1 |
| | GRASP/Models | "ResidualRelative"<1.0 | "ResidualRelative"<1.0 |
| | Operational | 0.8≤Quality Index≤1.0 | 0.8≤Quality Index≤1.0 |
| MODIS | DT | QA Flag = 3 | QA Flag = 3 |
| | DB | QA Flag = 3 | -[1] |
| | MAIAC | QA = '0000' | -[2] |

[1] DB aerosol product is not available over ocean.

[2] MAIAC aerosol product is presently only available for tiles containing land, so the ocean retrievals are not considered in this study.

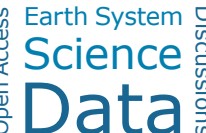

**Table 3. Global statistics of PARASOL/GRASP spectral AOD vs. AERONET AOD over land and ocean. The best performing of three approaches by each metric is labelled in bold.**

| Land/Ocean | Band (nm) | Products | R | Slope | Offset | RMSE | GCOS (%) | BIAS | BIAS τ<0.2 | BIAS 0.2≤ τ ≤0.7 | BIAS τ >0.7 |
|---|---|---|---|---|---|---|---|---|---|---|---|
| Land | 443 | Optimized (41268) | 0.900 | 0.867 | 0.104 | 0.179 | 26.7 | 0.06 | 0.09 | 0.06 | -0.06 |
| | | HP (42202) | 0.915 | 0.981 | 0.072 | 0.181 | 32.7 | 0.07 | 0.07 | 0.07 | 0.05 |
| | | Models (28449) | **0.932** | **1.013** | **0.003** | **0.140** | **49.3** | **0.01** | **0.01** | **0.00** | **0.02** |
| | 490 | Optimized (41268) | 0.892 | 0.879 | 0.099 | 0.171 | 26.8 | 0.06 | 0.08 | 0.06 | -0.04 |
| | | HP (42202) | 0.909 | **1.000** | 0.069 | 0.174 | 33.2 | 0.07 | 0.07 | 0.07 | 0.07 |
| | | Models (28449) | **0.929** | 1.025 | **0.003** | **0.131** | **51.6** | **0.01** | **0.01** | **0.01** | **0.03** |
| | 550 | Optimized (41268) | 0.876 | 0.847 | 0.101 | 0.162 | 27.5 | 0.06 | 0.08 | 0.05 | -0.08 |
| | | HP (42202) | 0.898 | 0.973 | 0.074 | 0.163 | 34.0 | 0.07 | 0.07 | 0.07 | 0.04 |
| | | Models (28449) | **0.922** | **1.023** | **0.005** | **0.123** | **54.2** | **0.01** | **0.01** | **0.01** | **0.03** |
| | 565 | Optimized (41268) | 0.877 | 0.877 | 0.096 | 0.161 | 27.3 | 0.06 | 0.08 | 0.06 | -0.05 |
| | | HP (42202) | 0.898 | **1.004** | 0.069 | 0.165 | 34.0 | 0.07 | 0.07 | 0.07 | 0.07 |
| | | Models (28449) | **0.920** | 1.011 | **0.006** | **0.120** | **54.4** | **0.01** | **0.01** | **0.00** | **0.02** |
| | 670 | Optimized (41268) | 0.858 | 0.823 | 0.099 | 0.152 | 28.4 | 0.06 | 0.08 | 0.05 | -0.10 |
| | | HP (42202) | 0.886 | **0.955** | 0.077 | 0.153 | 35.0 | 0.07 | 0.07 | 0.07 | **0.02** |
| | | Models (28449) | **0.911** | 0.954 | **0.016** | **0.108** | **58.6** | **0.01** | **0.01** | **-0.01** | -0.03 |
| | 865 | Optimized (41268) | 0.816 | 0.785 | 0.093 | 0.142 | 31.3 | 0.05 | 0.07 | 0.03 | -0.15 |
| | | HP (42202) | 0.856 | 0.932 | 0.074 | 0.142 | 37.6 | 0.06 | 0.06 | 0.07 | **-0.02** |
| | | Models (284449) | **0.880** | **0.935** | **0.018** | **0.105** | **60.3** | **0.01** | 0.02 | -0.01 | -0.04 |
| | 1020 | Optimized (40148) | 0.791 | 0.772 | 0.089 | 0.139 | 32.8 | 0.05 | 0.07 | 0.02 | -0.17 |
| | | HP (41016) | 0.837 | 0.924 | 0.073 | 0.138 | 38.8 | 0.06 | 0.06 | 0.06 | **-0.03** |
| | | Models (27551) | **0.856** | **0.943** | **0.023** | **0.109** | **59.5** | **0.01** | 0.02 | **0.00** | -0.04 |
| Ocean | 443 | Optimized (1495) | 0.938 | **1.028** | 0.049 | 0.084 | 40.5 | 0.05 | 0.05 | 0.07 | **0.03** |
| | | HP (1551) | 0.939 | 1.043 | 0.046 | 0.083 | 41.2 | 0.05 | 0.05 | 0.06 | 0.05 |
| | | Models (2064) | **0.940** | 0.970 | **0.026** | **0.066** | **60.6** | 0.02 | 0.02 | 0.03 | -0.06 |
| | 490 | Optimized (1495) | 0.939 | 1.064 | 0.041 | 0.079 | 43.2 | 0.05 | 0.04 | 0.07 | **0.05** |
| | | HP (1551) | 0.942 | 1.077 | 0.039 | 0.079 | 43.1 | 0.05 | 0.05 | 0.07 | 0.09 |
| | | Models (2064) | **0.946** | **0.969** | **0.023** | **0.057** | **65.1** | 0.02 | 0.02 | 0.02 | -0.05 |
| | 550 | Optimized (1495) | 0.936 | 1.060 | 0.035 | 0.071 | 48.4 | 0.05 | 0.04 | 0.06 | **0.04** |
| | | HP (1551) | 0.940 | 1.083 | 0.036 | 0.074 | 46.4 | 0.05 | 0.04 | 0.07 | 0.11 |
| | | Models (2064) | **0.950** | **0.960** | **0.019** | **0.050** | **70.3** | **0.01** | **0.01** | **0.01** | -0.05 |
| | 565 | Optimized (1495) | 0.939 | 1.090 | 0.033 | 0.072 | 48.5 | 0.05 | 0.04 | 0.07 | **0.05** |
| | | HP (1551) | 0.943 | 1.105 | 0.033 | 0.074 | 46.7 | 0.05 | 0.04 | 0.07 | 0.12 |
| | | Models (2064) | **0.950** | **0.939** | **0.020** | **0.048** | **71.2** | **0.01** | **0.01** | **0.00** | -0.07 |
| | 670 | Optimized (1495) | 0.936 | **1.071** | 0.030 | 0.064 | 55.8 | 0.04 | 0.04 | 0.06 | **0.02** |
| | | HP (1551) | 0.943 | 1.099 | 0.032 | 0.068 | 50.9 | 0.05 | 0.04 | 0.07 | 0.11 |
| | | Models (2064) | **0.951** | 0.876 | **0.021** | **0.043** | **77.3** | **0.00** | **0.01** | **-0.02** | -0.13 |



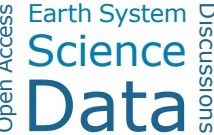

| | | | | | | | | | | |
|---|---|---|---|---|---|---|---|---|---|---|
| 865 | Optimized (1495) | 0.931 | **1.077** | 0.020 | 0.053 | 66.0 | 0.03 | 0.03 | 0.05 | 0.15 |
| | HP (1551) | 0.942 | 1.129 | 0.024 | 0.060 | 58.3 | 0.04 | 0.03 | 0.06 | 0.17 |
| | Models (2064) | **0.955** | 0.852 | **0.015** | **0.038** | **82.1** | **0.00** | **0.00** | **-0.03** | **-0.13** |
| 1020 | Optimized (1431) | 0.927 | **1.063** | 0.017 | 0.049 | 71.3 | 0.02 | 0.02 | 0.04 | 0.15 |
| | HP (1501) | 0.940 | 1.143 | 0.021 | 0.058 | 60.9 | 0.04 | 0.03 | 0.07 | 0.18 |
| | Models (2002) | **0.957** | 0.865 | **0.013** | **0.035** | **84.6** | **0.00** | **0.00** | **-0.03** | **-0.11** |





**Table 4. Global statistics of PARASOL/GRASP AE vs. AERONET AE (440/870) over land and ocean.**

**The best performing of three approaches by each metric is labelled in bold.**

|       |                   | R     | Slope | Offset | RMSE  | BIAS  |
|-------|-------------------|-------|-------|--------|-------|-------|
| Land  | Optimized (18594) | 0.797 | 0.680 | 0.213  | 0.358 | -0.10 |
|       | HP (19093)        | **0.843** | **0.716** | **0.139** | **0.336** | -0.14 |
|       | Models (11468)    | 0.681 | 0.415 | 0.511  | 0.420 | **-0.04** |
| Ocean | Optimized (363)   | 0.935 | 0.773 | 0.199  | 0.210 | **0.01** |
|       | HP (391)          | 0.949 | **0.817** | **0.092** | **0.193** | -0.05 |
|       | Models (522)      | **0.958** | 0.620 | 0.451  | 0.292 | 0.16  |






**Table 5. Global statistics of PARASOL/GRASP AODF vs. AERONET SDA AODF at 550 nm over land and ocean. The best performing of three approaches by each metric is labelled in bold.**

| | | R | Slope | Offset | RMSE | GCOS Fraction (%) | BIAS | BIAS $\tau_{f(550)}<0.2$ | BIAS $0.2\leq \tau_{f(550)}\leq0.7$ | BIAS $\tau_{f(550)}>0.7$ |
|---|---|---|---|---|---|---|---|---|---|---|
| Land | Optimized (31902) | 0.922 | 0.840 | 0.044 | 0.100 | 54.9 | 0.02 | 0.03 | **0.01** | -0.16 |
| | HP (32973) | **0.924** | **0.892** | 0.029 | 0.098 | 60.9 | **0.01** | 0.02 | **0.01** | **-0.10** |
| | Models (23653) | 0.868 | 0.662 | **0.028** | **0.094** | **64.5** | -0.02 | **0.00** | -0.07 | -0.37 |
| Ocean | Optimized (1074) | 0.901 | **0.958** | 0.042 | 0.058 | 56.7 | 0.04 | 0.04 | 0.05 | **-0.24** |
| | HP (1155) | **0.908** | 0.932 | **0.028** | **0.043** | 76.3 | **0.02** | **0.02** | 0.03 | -0.27 |
| | Models (1338) | 0.834 | 0.746 | 0.035 | 0.048 | **77.5** | **0.02** | **0.02** | **-0.03** | -0.33 |

**Table 6. Global statistics of PARASOL/GRASP AODC vs. AERONET SDA AODC at 550 nm over land and ocean. The best performing of three approaches by each metric is labelled in bold.**

| | | R | Slope | Offset | RMSE | GCOS Fraction (%) | BIAS | BIAS $\tau_{c(550)}<0.2$ | BIAS $0.2\leq \tau_{c(550)}\leq0.7$ | BIAS $\tau_{c(550)}>0.7$ |
|---|---|---|---|---|---|---|---|---|---|---|
| Land | Optimized (31903) | 0.686 | 0.744 | 0.062 | 0.117 | 43.1 | 0.04 | 0.05 | **-0.03** | -0.18 |
| | HP (32973) | **0.742** | **0.933** | 0.057 | 0.124 | 47.7 | 0.05 | 0.05 | **0.03** | **-0.02** |
| | Models (23651) | 0.571 | 0.653 | **0.040** | **0.112** | **64.3** | **0.02** | **0.03** | -0.08 | -0.28 |
| Ocean | Optimized (1076) | 0.871 | **0.942** | **0.009** | 0.046 | 77.6 | **0.01** | **0.00** | **0.01** | **0.03** |
| | HP (1156) | 0.915 | 1.119 | 0.015 | 0.051 | 70.0 | 0.02 | 0.02 | 0.06 | 0.20 |
| | Models (1337) | **0.922** | 0.754 | 0.010 | **0.036** | **84.8** | **-0.01** | **0.00** | -0.06 | -0.09 |




**Table 7. Global statistics of PARASOL/GRASP spectral SSA vs. AERONET SSA stratified by PARASOL AOD (565 nm) levels. The best performing at each wavelength of three approaches by each metric is labelled in bold.**

| AOD Level | Band (nm) | Products | R | Slope | Offset | RMSE | BIAS |
|---|---|---|---|---|---|---|---|
| All L3 data | 443 | Optimized (7192) | 0.285 | 0.292 | 0.631 | 0.051 | -0.01 |
| | | HP (7450) | 0.254 | 0.266 | 0.666 | 0.051 | **0.00** |
| | | Models (6095) | **0.348** | **0.349** | **0.582** | **0.047** | -0.01 |
| | 670 | Optimized (7192) | 0.511 | 0.608 | 0.324 | 0.065 | -0.04 |
| | | HP (7450) | **0.536** | **0.648** | **0.299** | **0.056** | -0.03 |
| | | Models (6095) | 0.321 | 0.334 | 0.602 | 0.057 | **-0.02** |
| | 865 | Optimized (7192) | 0.566 | 0.667 | 0.267 | 0.068 | -0.04 |
| | | HP (7450) | **0.594** | **0.698** | **0.253** | **0.058** | -0.03 |
| | | Models (6095) | 0.360 | 0.347 | 0.597 | 0.059 | **-0.01** |
| | 1020 | Optimized (7192) | 0.596 | 0.705 | 0.230 | 0.072 | -0.04 |
| | | HP (7450) | **0.627** | **0.730** | **0.223** | **0.060** | -0.03 |
| | | Models (6095) | 0.372 | 0.334 | 0.615 | 0.062 | **0.00** |
| AOD>0.5 | 443 | Optimized (3695) | 0.315 | 0.312 | 0.619 | 0.045 | -0.01 |
| | | HP (4235) | 0.242 | 0.242 | 0.691 | 0.047 | **0.00** |
| | | Models (2424) | **0.413** | **0.345** | **0.594** | **0.037** | **0.00** |
| | 670 | Optimized (3695) | 0.534 | 0.612 | 0.327 | 0.056 | -0.04 |
| | | HP (4235) | **0.552** | **0.642** | **0.307** | 0.051 | -0.03 |
| | | Models (2424) | 0.455 | 0.355 | 0.592 | **0.042** | **-0.01** |
| | 865 | Optimized (3695) | 0.593 | 0.668 | 0.274 | 0.059 | -0.04 |
| | | HP (4235) | **0.615** | **0.699** | **0.254** | 0.052 | -0.03 |
| | | Models (2424) | 0.535 | 0.387 | 0.571 | **0.043** | **0.00** |
| | 1020 | Optimized (3695) | 0.627 | 0.703 | 0.240 | 0.061 | -0.04 |
| | | HP (4235) | **0.647** | **0.726** | **0.228** | 0.054 | -0.03 |
| | | Models (2424) | 0.564 | 0.376 | 0.586 | **0.046** | **0.00** |
| AOD>1.0 | 443 | Optimized (715) | 0.478 | **0.459** | **0.499** | 0.034 | **0.00** |
| | | HP (976) | 0.398 | 0.366 | 0.587 | 0.037 | **0.00** |
| | | Models (463) | **0.585** | 0.457 | **0.499** | **0.027** | 0.01 |
| | 670 | Optimized (715) | **0.674** | **0.712** | **0.252** | 0.036 | -0.02 |
| | | HP (976) | 0.664 | 0.687 | 0.277 | 0.036 | -0.02 |
| | | Models (463) | 0.665 | 0.464 | 0.497 | **0.031** | **-0.01** |
| | 865 | Optimized (715) | 0.702 | **0.699** | **0.264** | 0.039 | -0.02 |
| | | HP (976) | 0.704 | 0.692 | 0.272 | 0.037 | -0.02 |
| | | Models (463) | **0.737** | 0.487 | 0.483 | **0.033** | **0.00** |
| | 1020 | Optimized (715) | 0.715 | 0.694 | 0.268 | 0.042 | -0.02 |
| | | HP (976) | 0.723 | **0.699** | **0.265** | 0.040 | -0.02 |
| | | Models (463) | **0.757** | 0.453 | 0.519 | **0.038** | 0.01 |
| AOD>1.5 | 443 | Optimized (212) | 0.544 | **0.536** | **0.430** | 0.030 | **0.00** |
| | | HP (317) | 0.527 | 0.518 | 0.459 | 0.031 | **0.00** |
| | | Models (116) | **0.639** | 0.491 | 0.472 | **0.022** | **0.00** |
| | 670 | Optimized (212) | 0.734 | 0.752 | 0.220 | 0.030 | -0.01 |
| | | HP (317) | 0.752 | **0.804** | **0.171** | 0.029 | -0.01 |
| | | Models (116) | **0.814** | 0.567 | 0.402 | **0.023** | **0.00** |
| | 865 | Optimized (212) | 0.760 | 0.688 | 0.283 | 0.032 | -0.01 |
| | | HP (317) | 0.770 | **0.738** | **0.235** | 0.030 | -0.01 |
| | | Models (116) | **0.876** | 0.602 | 0.375 | **0.025** | **0.00** |
| | 1020 | Optimized (212) | 0.770 | 0.666 | 0.303 | 0.035 | **-0.01** |
| | | HP (317) | 0.779 | **0.716** | **0.256** | 0.034 | **-0.01** |
| | | Models (116) | **0.889** | 0.556 | 0.423 | **0.032** | 0.01 |



**Table 8. Global statistics of PARASOL/GRASP spectral AAOD vs. AERONET AAOD. The best performing at each wavelength of three approaches by each metric is labelled in bold**

| Band (nm) | Products | R | Slope | Offset | RMSE | BIAS |
|---|---|---|---|---|---|---|
| 443 | Optimized (7192) | 0.486 | 0.475 | 0.040 | 0.046 | 0.01 |
| | HP (7450) | 0.498 | **0.536** | **0.034** | 0.047 | **0.00** |
| | Models (8046) | **0.538** | 0.509 | 0.035 | **0.042** | **0.00** |
| 670 | Optimized (7192) | 0.480 | 0.571 | 0.033 | 0.034 | 0.02 |
| | HP (7450) | **0.517** | **0.673** | 0.028 | 0.034 | 0.02 |
| | Models (8046) | 0.480 | 0.492 | **0.023** | **0.026** | **0.01** |
| 865 | Optimized (7192) | 0.393 | 0.476 | 0.029 | 0.028 | 0.02 |
| | HP (7450) | 0.438 | **0.574** | 0.024 | 0.028 | 0.01 |
| | Models (8046) | **0.444** | 0.439 | **0.017** | **0.020** | **0.00** |
| 1020 | Optimized (7192) | 0.343 | 0.430 | 0.026 | 0.025 | 0.01 |
| | HP (7450) | 0.394 | **0.526** | 0.022 | 0.025 | 0.01 |
| | Models (8046) | **0.414** | 0.409 | **0.015** | **0.018** | **0.00** |



**Table 9. Statistics of PARASOL and MODIS AOD products against collocated AERONET AOD at 550 nm over land and ocean**

| | | R | Slope | Offset | RMSE | GCOS Fraction (%) | BIAS | BIAS $\tau_{550}<0.2$ | BIAS $0.2\leq\tau_{550}\leq0.7$ | BIAS $\tau_{550}>0.7$ |
|---|---|---|---|---|---|---|---|---|---|---|
| | | | | | | Reference: AERONET AOD ($\tau_{550}$) | | | | |
| Land | GRASP/Optimized (3647) | 0.875 | 0.780 | 0.098 | 0.150 | 28.8 | 0.04 | 0.07 | 0.02 | -0.13 |
| | GRASP/HP (4777) | 0.908 | 0.938 | 0.078 | 0.157 | 32.4 | 0.06 | 0.07 | 0.06 | 0.02 |
| | GRASP/Models (3111) | **0.924** | **0.989** | **0.005** | 0.121 | **53.2** | **0.00** | 0.01 | **-0.01** | 0.02 |
| | DT (6858) | 0.898 | 0.988 | 0.021 | 0.120 | 46.1 | 0.02 | 0.02 | 0.02 | **0.00** |
| | DB (8409) | 0.870 | 0.841 | 0.026 | 0.126 | 48.8 | -0.01 | 0.01 | -0.04 | -0.14 |
| | MAIAC_0.1 (8164) | 0.895 | 0.793 | 0.007 | **0.112** | 52.8 | -0.03 | -0.01 | -0.08 | -0.19 |
| | MAIAC_0.01 (9054) | 0.874 | 0.796 | 0.014 | 0.125 | 48.1 | -0.03 | **0.00** | -0.08 | -0.19 |
| Ocean | GRASP/Optimized (116) | 0.950 | 1.145 | 0.033 | 0.089 | 42.4 | 0.06 | 0.05 | 0.09 | 0.17 |
| | GRASP/HP (154) | 0.947 | 1.074 | 0.054 | 0.092 | 26.6 | 0.07 | 0.06 | 0.08 | 0.13 |
| | GRASP/Models (205) | **0.963** | 0.965 | 0.024 | **0.061** | **62.9** | **0.02** | 0.02 | **0.02** | -0.04 |
| | Operational (207) | 0.954 | 1.165 | **-0.009** | 0.077 | 52.2 | 0.03 | **0.01** | 0.04 | 0.18 |
| | DT (218) | 0.952 | **0.974** | 0.037 | 0.081 | 55.0 | 0.03 | 0.03 | 0.04 | **0.00** |





**Table 10. Statistics of PARASOL and MODIS AOD products against collocated AERONET AOD at 550 nm classified by NDVI. In each individual NDVI level, the best performing metric is indicated in bold. The number of matched pairs is included in brackets.**

| | | R | Slope | Offset | RMSE | GCOS Fraction (%) | BIAS | BIAS $\tau_{550}<0.2$ | BIAS $0.2\leq \tau_{550}\leq0.7$ | BIAS $\tau_{550}>0.7$ |
|---|---|---|---|---|---|---|---|---|---|---|
| | | | | | | Reference: AERONET AOD ($\tau_{550}$) | | | | |
| Land NDVI<0.2 | GRASP/Optimized (1055) | 0.892 | 0.711 | 0.105 | 0.153 | 23.0 | 0.02 | 0.08 | **-0.01** | -0.20 |
| | GRASP/HP (1410) | **0.915** | 0.860 | 0.104 | 0.155 | 26.0 | 0.06 | 0.09 | 0.05 | -0.03 |
| | GRASP/Models (786) | 0.873 | **0.888** | **0.023** | 0.159 | 39.8 | **-0.01** | **0.01** | -0.03 | **0.01** |
| | DT (99) | 0.792 | 0.878 | 0.073 | 0.216 | **44.4** | 0.05 | 0.04 | 0.12 | -0.04 |
| | DB (1327) | 0.845 | 0.790 | 0.044 | 0.153 | 44.2 | **-0.01** | 0.03 | -0.05 | -0.16 |
| | MAIAC_0.1 (1853) | 0.883 | 0.734 | 0.032 | **0.139** | 42.4 | -0.03 | 0.02 | -0.09 | -0.22 |
| | MAIAC_0.01 (2087) | 0.853 | 0.734 | 0.041 | 0.155 | 35.7 | -0.03 | 0.03 | -0.09 | -0.22 |
| Land 0.2≤NDVI<0.4 | GRASP/Optimized (1106) | 0.881 | 0.777 | 0.101 | 0.161 | 31.9 | 0.04 | 0.07 | 0.03 | -0.16 |
| | GRASP/HP (1479) | 0.928 | 0.911 | 0.074 | 0.145 | 39.4 | 0.05 | 0.05 | 0.06 | **-0.03** |
| | GRASP/Models (1020) | **0.953** | **1.062** | -0.014 | **0.125** | **52.7** | **0.00** | **0.00** | -0.01 | 0.04 |
| | DT (1847) | 0.895 | 0.947 | 0.029 | 0.145 | 40.6 | 0.02 | 0.03 | **0.01** | -0.05 |
| | DB (2204) | 0.888 | 0.883 | 0.010 | 0.142 | 46.3 | -0.02 | 0.00 | -0.05 | -0.09 |
| | MAIAC_0.1 (2049) | 0.901 | 0.825 | -0.005 | 0.133 | 53.9 | -0.04 | -0.01 | -0.11 | -0.18 |
| | MAIAC_0.01 (2363) | 0.896 | 0.826 | **0.002** | 0.134 | 51.5 | -0.04 | 0.00 | -0.10 | -0.18 |
| Land 0.4≤NDVI<0.6 | GRASP/Optimized (958) | 0.880 | 0.868 | 0.083 | 0.138 | 31.7 | 0.05 | 0.07 | 0.03 | -0.02 |
| | GRASP/HP (1249) | 0.903 | 1.069 | 0.047 | 0.173 | 33.2 | 0.07 | 0.06 | 0.07 | 0.12 |
| | GRASP/Models (1074) | **0.920** | 0.952 | 0.014 | **0.086** | **61.5** | **0.00** | 0.01 | **0.00** | -0.04 |
| | DT (2702) | 0.907 | **0.994** | 0.012 | 0.112 | 46.6 | 0.01 | **0.01** | 0.02 | **-0.01** |
| | DB (2718) | 0.866 | 0.808 | 0.030 | 0.120 | 50.0 | -0.01 | **0.01** | -0.03 | -0.20 |
| | MAIAC_0.1 (2193) | 0.911 | 0.821 | -0.009 | 0.093 | 53.5 | -0.04 | -0.03 | -0.07 | -0.19 |
| | MAIAC_0.01 (2530) | 0.899 | 0.827 | **-0.002** | 0.097 | 50.5 | -0.03 | -0.02 | -0.07 | -0.17 |
| Land NDVI≥0.6 | GRASP/Optimized (194) | 0.832 | 0.932 | 0.108 | 0.145 | 23.7 | 0.09 | 0.10 | 0.08 | 0.09 |
| | GRASP/HP (287) | 0.853 | **1.001** | 0.107 | 0.160 | 21.3 | 0.11 | 0.10 | 0.11 | 0.10 |
| | GRASP/Models (231) | **0.910** | 1.115 | 0.011 | 0.083 | 61.9 | 0.03 | 0.02 | 0.07 | 0.06 |
| | DT (943) | **0.910** | 1.118 | **-0.005** | 0.076 | 55.0 | 0.01 | **0.01** | 0.03 | 0.17 |
| | DB (907) | 0.855 | 0.884 | 0.015 | 0.076 | 60.2 | **0.00** | **0.01** | -0.04 | 0.07 |
| | MAIAC_0.1 (651) | 0.826 | 0.837 | **-0.005** | **0.063** | **66.2** | -0.02 | -0.02 | -0.06 | **-0.04** |
| | MAIAC_0.01 (669) | 0.840 | 0.929 | -0.013 | 0.074 | 61.3 | -0.02 | -0.02 | -0.04 | 0.10 |





**Table 11. Statistics of PARASOL and MODIS AE products against collocated AERONET AE over land and ocean, with a threshold of satellite AOD (550 nm) > 0.2.**

| | | R | Slope | Offset | RMSE | BIAS |
|---|---|---|---|---|---|---|
| | Reference: AERONET AE | | | | | |
| Land | GRASP/Optimized (2035) | 0.745 | 0.641 | 0.168 | 0.435 | -0.19 |
| | GRASP/HP (2791) | **0.772** | **0.654** | **0.122** | **0.425** | -0.21 |
| | GRASP/Models (1253) | 0.686 | 0.407 | 0.507 | 0.443 | -0.06 |
| | DT (2589) | 0.390 | 0.372 | 0.514 | 0.599 | -0.31 |
| | DB (3279) | 0.563 | 0.650 | 0.444 | 0.573 | **0.04** |
| Ocean | GRASP/Optimized (55) | 0.840 | 0.724 | 0.183 | 0.279 | -0.02 |
| | GRASP/HP (80) | 0.890 | 0.810 | **0.051** | **0.229** | **-0.08** |
| | GRASP/Models (92) | **0.949** | 0.625 | 0.431 | 0.291 | 0.20 |
| | Operational (57) | 0.891 | **0.812** | 0.841 | 0.782 | 0.75 |
| | DT (106) | 0.832 | 0.610 | 0.317 | 0.305 | **0.08** |



**Table 12. Statistics of PARASOL and MODIS AODF products against collocated AERONET AODF over land and ocean**

| | | R | Slope | Offset | RMSE | GCOS Fraction (%) | BIAS | BIAS $\tau_{f(550)}<0.2$ | BIAS $0.2\leq\tau_{f(550)}\leq0.7$ | BIAS $\tau_{f(550)}>0.7$ |
|---|---|---|---|---|---|---|---|---|---|---|
| | Reference: AERONET AODF ($\tau_{f\,550}$) | | | | | | | | | |
| Land | GRASP/Optimized (2634) | 0.923 | 0.762 | 0.043 | **0.104** | 58.0 | **0.00** | 0.02 | -0.01 | -0.23 |
| | GRASP/HP (3507) | **0.926** | **0.828** | 0.036 | 0.106 | 59.5 | 0.01 | 0.02 | **0.00** | **-0.17** |
| | GRASP/Models (2795) | 0.868 | 0.587 | **0.035** | 0.124 | **63.8** | -0.02 | **0.00** | -0.08 | -0.46 |
| | Operational (2619) | 0.886 | 0.546 | 0.052 | 0.162 | 50.5 | -0.04 | **0.00** | -0.08 | -0.45 |
| Ocean | GRASP/Optimized (91) | 0.893 | 1.397 | 0.023 | 0.079 | 40.7 | 0.06 | 0.05 | 0.10 | -- |
| | GRASP/HP (129) | **0.924** | 1.118 | 0.018 | **0.049** | **75.2** | 0.03 | 0.03 | 0.05 | -- |
| | GRASP/Models (168) | 0.866 | **1.046** | 0.028 | 0.054 | 65.5 | 0.03 | 0.03 | 0.02 | -- |
| | Operational (82) | 0.780 | 1.082 | **0.017** | 0.061 | 67.1 | **0.02** | **0.02** | **0.00** | -- |
| | DT (119) | 0.808 | 0.887 | 0.048 | 0.067 | 56.3 | 0.04 | 0.04 | -0.01 | |

**Table 13. Statistics of PARASOL and MODIS AODC products against collocated AERONET AODC over land and ocean**

| | | R | Slope | Offset | RMSE | GCOS Fraction (%) | BIAS | BIAS $\tau_{c(550)}<0.2$ | BIAS $0.2\leq\tau_{c(550)}\leq0.7$ | BIAS $\tau_{c(550)}>0.7$ |
|---|---|---|---|---|---|---|---|---|---|---|
| | Reference: AERONET AODC ($\tau_{c\,550}$) | | | | | | | | | |
| Land | GRASP/Optimized (2634) | 0.700 | 0.678 | 0.058 | **0.114** | 45.6 | 0.03 | 0.04 | -0.06 | -0.24 |
| | GRASP/HP (3506) | **0.771** | **0.912** | 0.060 | 0.127 | 45.8 | 0.05 | 0.06 | **0.02** | **0.00** |
| | GRASP/Models (2795) | 0.536 | 0.596 | **0.043** | 0.125 | **63.7** | **0.01** | **0.03** | -0.12 | -0.28 |
| | DT (1472) | 0.501 | **0.912** | 0.246 | 0.332 | 11.8 | 0.24 | 0.24 | 0.22 | 0.06 |
| Ocean | GRASP/Optimized (91) | 0.936 | 1.033 | 0.021 | 0.062 | 59.3 | 0.03 | 0.02 | 0.05 | **0.03** |
| | GRASP/HP (129) | 0.961 | 1.113 | 0.033 | 0.070 | 45.0 | 0.05 | 0.04 | 0.07 | 0.20 |
| | GRASP/Models (168) | **0.966** | 0.827 | **0.008** | **0.040** | **81.5** | -0.01 | **0.00** | -0.05 | -0.09 |
| | Operational (82) | 0.936 | **0.971** | 0.014 | 0.045 | 74.4 | 0.01 | 0.01 | **-0.01** | -- |
| | DT (119) | 0.911 | 0.806 | 0.025 | 0.045 | 68.9 | **0.00** | 0.01 | -0.03 | -0.11 |






**Table 14. Pixel to pixel (0.2° x 0.2°) statistical metrics between PARASOL/Operational and PARASOL/GRASP aerosol products; the statistics for only AERONET pixels are presented in brackets.**

| | | | R | Slope | Offset | RMSE | BIAS | BIAS $\tau/\tau_f/\tau_c<0.2$ | BIAS $0.2\leq\tau/\tau_f/\tau_c\leq0.7$ | BIAS $\tau/\tau_f/\tau_c>0.7$ |
|---|---|---|---|---|---|---|---|---|---|---|
| | | | | | | Reference: PARASOL/Operational | | | | |
| Land | AODF | GRASP/HP 8 209 015 (7801) | 0.88 (0.91) | 1.22 (1.13) | 0.03 (0.02) | 0.13 (0.13) | 0.06 (0.05) | 0.04 (0.04) | 0.10 (0.05) | 0.29 (0.18) |
| | | GRASP/Models 8 117 246 (7427) | 0.85 (0.83) | 0.94 (0.74) | 0.03 (0.04) | 0.10 (0.13) | 0.02 (-0.01) | 0.03 (0.01) | 0.02 (-0.04) | -0.05 (-0.19) |
| Ocean | AOD | GRASP/HP 5 702 109 (93) | 0.88 (0.98) | 1.19 (1.22) | 0.04 (0.04) | 0.10 (0.16) | 0.06 (0.11) | 0.06 (0.08) | 0.10 (0.10) | 0.22 (0.38) |
| | | GRASP/Models 5 988 842 (91) | 0.94 (0.99) | 1.09 (1.14) | 0.00 (0.01) | 0.04 (0.09) | 0.01 (0.05) | 0.01 (0.04) | 0.03 (0.03) | 0.10 (0.24) |
| | AE | GRASP/HP 5 703 282 (93) | 0.59 (0.67) | 0.53 (0.62) | -0.04 (-0.18) | 0.62 (0.77) | -0.48 (-0.66) | -- | -- | -- |
| | | GRASP/Models 5 988 842 (91) | 0.38 (0.53) | 0.24 (0.40) | 0.69 (0.41) | 0.44 (0.53) | -0.03 (-0.34) | -- | -- | -- |
| | AODF | GRASP/HP 5 704 665 (126) | 0.63 (0.41) | 0.52 (0.24) | 0.05 (0.12) | 0.07 (0.15) | 0.02 (0.00) | 0.03 (0.05) | -0.12 (-0.09) | -0.40 (-0.64) |
| | | GRASP/Models 5 991 408 (125) | 0.73 (0.74) | 0.57 (0.58) | 0.04 (0.09) | 0.06 (0.11) | 0.01 (0.02) | 0.02 (0.05) | -0.12 (-0.06) | -0.31 (-0.23) |
| | AODC | GRASP/HP 2 692 908 (--) | 0.68 (--) | 1.08 (--) | 0.04 (--) | 0.10 (--) | 0.05 (--) | 0.04 (--) | 0.17 (--) | 0.53 (--) |
| | | GRASP/Models 2 949 016 (--) | 0.70 (--) | 0.69 (--) | 0.02 (--) | 0.06 (--) | 0.00 (--) | 0.00 (--) | -0.01 (--) | 0.04 (--) |




**Table 15. Pixel to pixel (0.1° x 0.1°) statistical metrics between AOD (550 nm) products based on references of GRASP/Models (Land and Ocean) and MAIAC AOD (Land); the statistics for only AERONET pixels are presented in brackets.**

| | | R | Slope | Offset | RMSE | BIAS | BIAS $\tau_{550}<0.2$ | BIAS $0.2\leq\tau_{550}\leq0.7$ | BIAS $\tau_{550}>0.7$ |
|---|---|---|---|---|---|---|---|---|---|
| colspan | Reference: GRASP/Models AOD | | | | | | | | |
| Land | GRASP/HP AOD 53 656 407 (8564) | 0.85 (0.90) | 0.81 (0.84) | 0.11 (0.12) | 0.19 (0.19) | 0.06 (0.06) | 0.07 (0.08) | 0.06 (0.06) | -0.07 (-0.02) |
| | DT AOD 13 069 294 (3 432) | 0.85 (0.90) | 0.84 (1.01) | 0.00 (0.06) | 0.15 (0.18) | -0.04 (0.06) | 0.00 (0.04) | -0.07 (0.07) | -0.15 (0.07) |
| | DB AOD 36 348 953 (4 972) | 0.76 (0.89) | 0.69 (0.89) | 0.06 (0.03) | 0.18 (0.16) | -0.01 (0.00) | 0.03 (0.02) | -0.04 (-0.02) | -0.26 (-0.06) |
| | MAIAC AOD 64 921 447 (10 830) | 0.77 (0.89) | 0.66 (0.84) | 0.05 (0.02) | 0.18 (0.15) | -0.03 (-0.02) | 0.02 (0.01) | -0.08 (-0.04) | -0.29 (-0.11) |
| Ocean | GRASP/HP AOD 65 551 501 (300) | 0.94 (0.97) | 1.10 (1.04) | 0.04 (0.05) | 0.09 (0.10) | 0.05 (0.06) | 0.05 (0.05) | 0.07 (0.06) | 0.16 (0.11) |
| | DT AOD 32 486 105 (130) | 0.92 (0.97) | 0.88 (0.99) | 0.01 (-0.01) | 0.05 (0.05) | -0.01 (-0.01) | 0.00 (-0.02) | -0.04 (-0.01) | -0.06 (-0.04) |
| | DB AOD | -- | -- | -- | -- | -- | -- | -- | -- |
| | MAIAC AOD | -- | | -- | -- | -- | -- | -- | -- |
| colspan | Reference: MAIAC AOD | | | | | | | | |
| Land | GRASP/HP AOD 54 693 580 (8 679) | 0.81 (0.90) | 0.91 (0.91) | 0.10 (0.11) | 0.20 (0.17) | 0.08 (0.08) | 0.08 (0.09) | 0.10 (0.09) | 0.00 (0.02) |
| | DT AOD 21 272 908 (5 836) | 0.91 (0.93) | 1.05 (1.07) | 0.01 (0.06) | 0.10 (0.15) | 0.02 (0.08) | 0.01 (0.05) | 0.01 (0.11) | 0.07 (0.12) |
| | DB AOD 53 758 759 (7 681) | 0.86 (0.93) | 0.92 (0.98) | 0.04 (0.03) | 0.13 (0.12) | 0.02 (0.02) | 0.03 (0.02) | 0.02 (0.02) | -0.05 (0.00) |







**Table 16. Pixel to pixel (0.1° x 0.1°) statistical metrics between AOD products based on reference of GRASP/Models over land pixels with four classes of surface (NDVI<0.2, 0.2≤NDVI<0.4, 0.4≤NDVI<0.6, and NDVI≥0.6); the statistics for only AERONET pixels are presented in brackets.**

| | | R | Slope | Offset | RMSE | BIAS | BIAS $\tau_{550}<0.2$ | BIAS $0.2\leq \tau_{550}\leq0.7$ | BIAS $\tau_{550}>0.7$ |
|---|---|---|---|---|---|---|---|---|---|
| | | | | | | Reference: GRASP/Models AOD | | | |
| Land NDVI<0.2 | GRASP/HP AOD 31 341 330 (2069) | 0.78 (0.81) | 0.68 (0.66) | 0.15 (0.19) | 0.20 (0.28) | 0.06 (0.08) | 0.09 (0.12) | 0.07 (0.09) | -0.20 (-0.21) |
| | DT AOD 542 625 (38) | 0.74 (0.88) | 0.95 (1.27) | 0.09 (0.16) | 0.17 (0.31) | 0.09 (0.24) | 0.09 (0.18) | 0.07 (0.26) | 0.12 (0.47) |
| | DB AOD 17 834 405 (1013) | 0.66 (0.82) | 0.59 (0.74) | 0.11 (0.08) | 0.22 (0.21) | 0.01 (0.00) | 0.06 (0.05) | -0.02 (-0.03) | -0.33 (-0.17) |
| | MAIAC AOD 31 329 712 (2357) | 0.67 (0.79) | 0.60 (0.68) | 0.08 (0.08) | 0.21 (0.21) | -0.01 (-0.01) | 0.04 (0.05) | -0.06 (-0.04) | -0.32 (-0.26) |
| Land 0.2≤NDVI<0.4 | GRASP/HP AOD 11 667 461 (3596) | 0.90 (0.93) | 0.84 (0.91) | 0.08 (0.08) | 0.16 (0.16) | 0.04 (0.04) | 0.05 (0.06) | 0.05 (0.05) | -0.06 (0.00) |
| | DT AOD 3 784 302 (1547) | 0.81 (0.89) | 0.86 (0.99) | 0.01 (0.09) | 0.16 (0.21) | -0.02 (0.08) | 0.01 (0.06) | -0.06 (0.10) | -0.08 (0.08) |
| | DB AOD 7 767 588 (1911) | 0.85 (0.92) | 0.78 (0.95) | 0.03 (0.02) | 0.14 (0.16) | -0.01 (0.00) | 0.01 (0.01) | -0.04 (0.00) | -0.15 (-0.03) |
| | MAIAC AOD 13 927 469 (4133) | 0.87 (0.92) | 0.69 (0.89) | 0.04 (0.02) | 0.15 (0.15) | -0.03 (-0.02) | 0.01 (0.01) | -0.08 (-0.04) | -0.26 (-0.08) |
| Land 0.4≤NDVI<0.6 | GRASP/HP AOD 7 879 243 (2641) | 0.92 (0.93) | 0.98 (0.95) | 0.07 (0.08) | 0.16 (0.14) | 0.06 (0.06) | 0.06 (0.07) | 0.05 (0.06) | 0.06 (0.06) |
| | DT AOD 5 431 789 (1605) | 0.86 (0.91) | 0.83 (1.01) | -0.01 (0.04) | 0.16 (0.14) | -0.06 (0.04) | -0.02 (0.04) | -0.08 (0.05) | -0.17 (0.05) |
| | DB AOD 7 146 072 (1763) | 0.88 (0.91) | 0.80 (0.89) | 0.01 (0.02) | 0.13 (0.13) | -0.03 (-0.01) | 0.00 (0.01) | -0.07 (-0.02) | -0.16 (-0.06) |
| | MAIAC AOD 12 624 553 (3660) | 0.88 (0.92) | 0.74 (0.90) | 0.01 (0.00) | 0.14 (0.11) | -0.05 (-0.03) | -0.01 (-0.01) | -0.10 (-0.05) | -0.24 (-0.08) |
| Land NDVI≥0.6 | GRASP/HP AOD 2 766 521 (258) | 0.94 (0.89) | 1.00 (0.90) | 0.10 (0.10) | 0.19 (0.13) | 0.10 (0.08) | 0.08 (0.09) | 0.11 (0.07) | 0.13 (0.01) |
| | DT AOD 3 305 544 (242) | 0.91 (0.89) | 0.86 (0.85) | -0.01 (0.01) | 0.13 (0.09) | -0.05 (-0.02) | -0.02 (0.00) | -0.08 (-0.04) | -0.17 (-0.15) |
| | DB AOD 3 598 331 (285) | 0.90 (0.77) | 0.72 (0.55) | 0.00 (0.02) | 0.12 (0.10) | -0.05 (-0.05) | -0.02 (-0.02) | -0.11 (-0.12) | -0.25 (-0.28) |
| | MAIAC AOD 7 029 548 (680) | 0.90 (0.85) | 0.73 (0.71) | 0.00 (0.00) | 0.14 (0.10) | -0.06 (-0.05) | -0.02 (-0.03) | -0.11 (-0.10) | -0.28 (-0.30) |





**Table 17. The same as Table 16, but for reference of MAIAC AOD**

| | | R | Slope | Offset | RMSE | BIAS | BIAS $\tau_{550}<0.2$ | BIAS $0.2\leq \tau_{550}\leq0.7$ | BIAS $\tau_{550}>0.7$ |
|---|---|---|---|---|---|---|---|---|---|
| Land NDVI<0.2 | GRASP/HP AOD 32 768 635 (2207) | 0.78 (0.86) | 0.80 (0.83) | 0.12 (0.13) | 0.20 (0.18) | 0.08 (0.08) | 0.08 (0.09) | 0.08 (0.10) | -0.11 (-0.08) |
| | DT AOD 885 841 (83) | 0.77 (0.71) | 1.28 (1.43) | 0.05 (0.33) | 0.15 (0.51) | 0.08 (0.42) | 0.07 (0.34) | 0.14 (0.56) | 0.33 (0.72) |
| | DB AOD 26 151 234 (1500) | 0.85 (0.89) | 0.86 (0.96) | 0.06 (0.02) | 0.15 (0.16) | 0.04 (0.01) | 0.04 (0.02) | 0.03 (-0.01) | -0.09 (0.00) |
| Land 0.2≤NDVI<0.4 | GRASP/HP AOD 11 919 986 (3641) | 0.85 (0.91) | 1.03 (0.92) | 0.06 (0.09) | 0.17 (0.17) | 0.07 (0.07) | 0.06 (0.07) | 0.09 (0.08) | 0.08 (0.02) |
| | DT AOD 5 857 865 (2314) | 0.88 (0.94) | 1.03 (1.05) | 0.00 (0.08) | 0.11 (0.17) | 0.01 (0.10) | 0.01 (0.06) | 0.00 (0.13) | 0.08 (0.13) |
| | DB AOD 11 668 355 (2922) | 0.87 (0.94) | 1.02 (1.00) | 0.01 (0.02) | 0.12 (0.13) | 0.01 (0.02) | 0.01 (0.02) | 0.02 (0.04) | 0.05 (0.01) |
| Land 0.4≤NDVI<0.6 | GRASP/HP AOD 7 489 541 (2526) | 0.87 (0.91) | 1.09 (0.96) | 0.09 (0.11) | 0.21 (0.17) | 0.11 (0.10) | 0.09 (0.10) | 0.14 (0.10) | 0.17 (0.08) |
| | DT AOD 8 401 731 (2416) | 0.92 (0.94) | 1.03 (1.06) | 0.00 (0.05) | 0.10 (0.13) | 0.01 (0.07) | 0.01 (0.05) | 0.00 (0.09) | 0.05 (0.12) |
| | DB AOD 10 298 915 (2628) | 0.88 (0.95) | 0.97 (0.95) | 0.02 (0.04) | 0.11 (0.09) | 0.02 (0.03) | 0.02 (0.03) | 0.01 (0.03) | 0.00 (0.00) |
| Land NDVI≥0.6 | GRASP/HP AOD 2 512 741 (305) | 0.88 (0.84) | 1.13 (.94) | 0.14 (0.15) | 0.26 (0.18) | 0.17 (0.14) | 0.13 (0.14) | 0.24 (0.13) | 0.26 (0.08) |
| | DT AOD 5 539 285 (548) | 0.94 (0.94) | 1.10 (1.06) | 0.00 (0.02) | 0.09 (0.07) | 0.02 (0.03) | 0.01 (0.02) | 0.04 (0.05) | 0.09 (0.05) |
| | DB AOD 5 253 920 (520) | 0.88 (0.78) | 0.91 (0.73) | 0.02 (0.03) | 0.09 (0.06) | 0.01 (0.00) | 0.01 (0.01) | -0.02 (-0.05) | -0.07 (-0.27) |

Reference: MAIAC AOD





**Table 18. Pixel to pixel statistical metrics between AE products based on references of GRASP/HP; the statistics for only AERONET**

**pixels are presented in brackets.**

| | Reference: GRASP/HP AE | | | | |
|---|---|---|---|---|---|
| | | R | Slope | Offset | RMSE | BIAS |
| Land | GRASP/Models AE 27 385 356 (5 517) | 0.70 (0.68) | 0.51 (0.43) | 0.45 (0.47) | 0.39 (0.39) | 0.12 (-0.05) |
| | DT AE 6 017 122 (2 335) | 0.31 (0.30) | 0.32 (0.29) | 0.84 (0.64) | 0.66 (0.59) | 0.11 (-0.15) |
| | DB AE 19 317 232 (3 121) | 0.40 (0.43) | 0.53 (0.49) | 0.39 (0.68) | 0.67 (0.65) | 0.09 (0.21) |
| Ocean | GRASP/Models AE 49 987 062 (285) | 0.74 (0.88) | 0.52 (0.68) | 0.63 (0.47) | 0.45 (0.33) | 0.35 (0.23) |
| | DT AE 18 564 876 (123) | 0.46 (0.55) | 0.49 (0.78) | 0.55 (0.82) | 0.53 (0.83) | 0.25 (0.60) |





**Table 19. Pixel to pixel statistical metrics between AE products based on references of GRASP/HP over land pixels with four classes of surface (NDVI<0.2, 0.2≤NDVI<0.4, 0.4≤NDVI<0.6, and NDVI≥0.6)**

| Reference: GRASP/HP AE | | R | Slope | Offset | RMSE | BIAS |
|---|---|---|---|---|---|---|
| Land NDVI<0.2 | GRASP/Models AE 15 916 616 (1205) | 0.40 (0.53) | 0.38 (0.48) | 0.49 (0.48) | 0.42 (0.42) | 0.23 (0.24) |
| | DT AE 203 121 (25) | 0.16 (0.36) | 0.14 (0.15) | 0.71 (0.47) | 0.65 (0.48) | 0.32 (-0.30) |
| | DB AE 12 223 721 (764) | 0.12 (0.35) | 0.21 (0.60) | 0.37 (0.40) | 0.65 (0.65) | 0.02 (0.21) |
| Land 0.2≤NDVI<0.4 | GRASP/Models AE 5 220 459 (2425) | 0.79 (0.69) | 0.54 (0.41) | 0.42 (0.47) | 0.35 (0.39) | 0.05 (-0.11) |
| | DT AE 1 923 619 (1168) | 0.30 (0.33) | 0.30 (0.27) | 0.79 (0.58) | 0.69 (0.59) | 0.16 (-0.20) |
| | DB AE 3 157 768 (1256) | 0.21 (0.24) | 0.23 (0.26) | 0.86 (0.98) | 0.77 (0.71) | 0.24 (0.24) |
| Land 0.4≤NDVI<0.6 | GRASP/Models AE 4 516 281 (1743) | 0.80 (0.65) | 0.57 (0.48) | 0.38 (0.43) | 0.34 (0.37) | -0.11 (-0.16) |
| | DT AE 2 723 494 (1024) | 0.28 (0.26) | 0.30 (0.29) | 0.90 (0.71) | 0.65 (0.58) | 0.08 (-0.12) |
| | DB AE 2 896 017 (999) | 0.23 (0.27) | 0.21 (0.30) | 1.04 (1.00) | 0.64 (0.58) | 0.15 (0.19) |
| Land NDVI≥0.6 | GRASP/Models AE 1 730 292 (144) | 0.76 (0.73) | 0.57 (0.67) | 0.41 (0.24) | 0.31 (0.29) | -0.08 (-0.14) |
| | DT AE 1 166 000 (118) | 0.19 (-0.01) | 0.22 (-0.01) | 1.00 (1.30) | 0.65 (0.60) | 0.09 (0.14) |
| | DB AE 1 039 192 (102) | 0.18 (-0.07) | 0.16 (-0.11) | 1.21 (1.44) | 0.59 (0.63) | 0.25 (0.17) |






**Table 20: Pixel to pixel statistical metrics between GRASP/HP AODF with other AODF products; the statistics for only AERONET pixels are presented in brackets.**

| | | R | Slope | Offset | RMSE | BIAS |
|---|---|---|---|---|---|---|
| | Reference: GRASP/HP AODF | | | | | |
| Land | GRASP/Models AODF 53 656 407 (8 564) | 0.87 (0.91) | 0.75 (0.68) | 0.02 (0.03) | 0.10 (0.15) | -0.01 (-0.06) |
| Ocean | GRASP/Models AODF 65 551 501 (300) | 0.89 (0.67) | 0.78 (0.90) | 0.02 (0.03) | 0.05 (0.11) | 0.00 (0.01) |
| | DT AODF 17 513 511 (116) | 0.86 (0.70) | 0.66 (0.64) | 0.02 (0.04) | 0.06 (0.09) | -0.02 (-0.03) |

**Table 21: The same as Table 20, but for AODC**

| | | R | Slope | Offset | RMSE | BIAS |
|---|---|---|---|---|---|---|
| | Reference: GRASP/HP AODC | | | | | |
| Land | GRASP/Models AODC 53 656 407 (8 564) | 0.71 (0.63) | 0.65 (0.67) | 0.02 (0.06) | 0.16 (0.18) | -0.04 (0.00) |
| Ocean | GRASP/Models AODC 65 551 501 (300) | 0.89 (0.98) | 0.56 (0.64) | 0.00 (0.01) | 0.09 (0.14) | -0.05 (-0.07) |
| | DT AODC 17 513 511 (116) | 0.84 (0.90) | 0.58 (0.69) | 0.01 (0.00) | 0.08 (0.10) | -0.04 (-0.04) |






**Table 22: Pixel to pixel statistical metrics between AODF products based on references of GRASP/HP over land pixels with four classes of surface (NDVI<0.2, 0.2≤NDVI<0.4, 0.4≤NDVI<0.6, and NDVI≥0.6); the statistics for only AERONET pixels are presented in brackets.**

| | | | | | | |
|---|---|---|---|---|---|---|
| | Reference: GRASP/HP AODF | | | | | |
| | | R | Slope | Offset | RMSE | BIAS |
| Land NDVI<0.2 | GRASP/Models AODF 31 340 947 (2069) | 0.68 (0.82) | 0.91 (0.84) | 0.02 (0.03) | 0.09 (0.12) | 0.00 (0.01) |
| Land 0.2≤NDVI<0.4 | GRASP/Models AODF 11 667 461 (3596) | 0.90 (0.93) | 0.79 (0.68) | 0.02 (0.03) | 0.09 (0.16) | -0.02 (-0.07) |
| Land 0.4≤NDVI<0.6 | GRASP/Models AODF 7 879 243 (2641) | 0.93 (0.92) | 0.73 (0.67) | 0.01 (0.02) | 0.13 (0.16) | -0.06 (-0.09) |
| Land NDVI≥0.6 | GRASP/Models AODF 2 766 521 (258) | 0.94 (0.88) | 0.76 (0.71) | -0.01 (-0.01) | 0.16 (0.12) | -0.09 (-0.08) |

**Table 23: The same as Table 22, but for AODC**

| | | | | | | |
|---|---|---|---|---|---|---|
| | Reference: GRASP/HP AODC | | | | | |
| | | R | Slope | Offset | RMSE | BIAS |
| Land NDVI<0.2 | GRASP/Models AODC 31 340 947 (2069) | 0.69 (0.67) | 0.64 (0.72) | 0.01 (-0.01) | 0.18 (0.23) | -0.07 (-0.08) |
| Land 0.2≤NDVI<0.4 | GRASP/Models AODC 11 667 461 (3596) | 0.77 (0.64) | 0.75 (0.69) | 0.01 (0.08) | 0.13 (0.18) | -0.02 (0.03) |
| Land 0.4≤NDVI<0.6 | GRASP/Models AODC 7 879 243 (2641) | 0.76 (0.60) | 0.69 (0.69) | 0.04 (0.06) | 0.13 (0.13) | 0.01 (0.03) |
| Land NDVI≥0.6 | GRASP/Models AODC 2 766 521 (258) | 0.77 (0.65) | 0.66 (0.70) | 0.04 (0.03) | 0.14 (0.08) | -0.01 (0.01) |

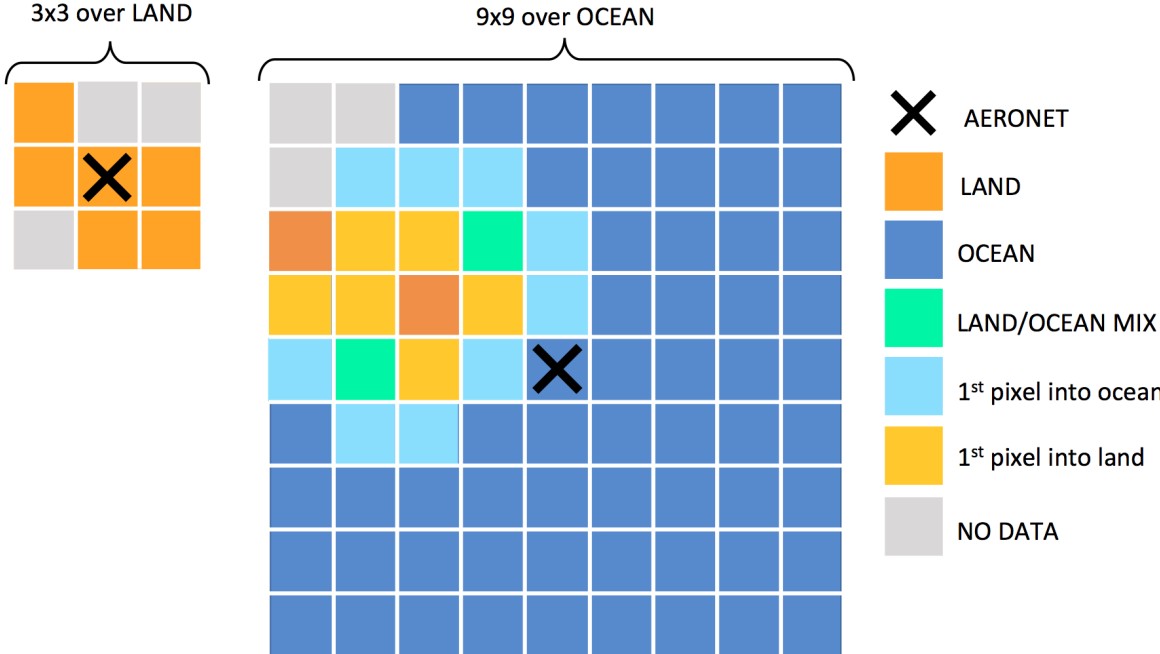

**Figure 1: Schematic diagram for satellite data selection over (left) land and (right) ocean.**

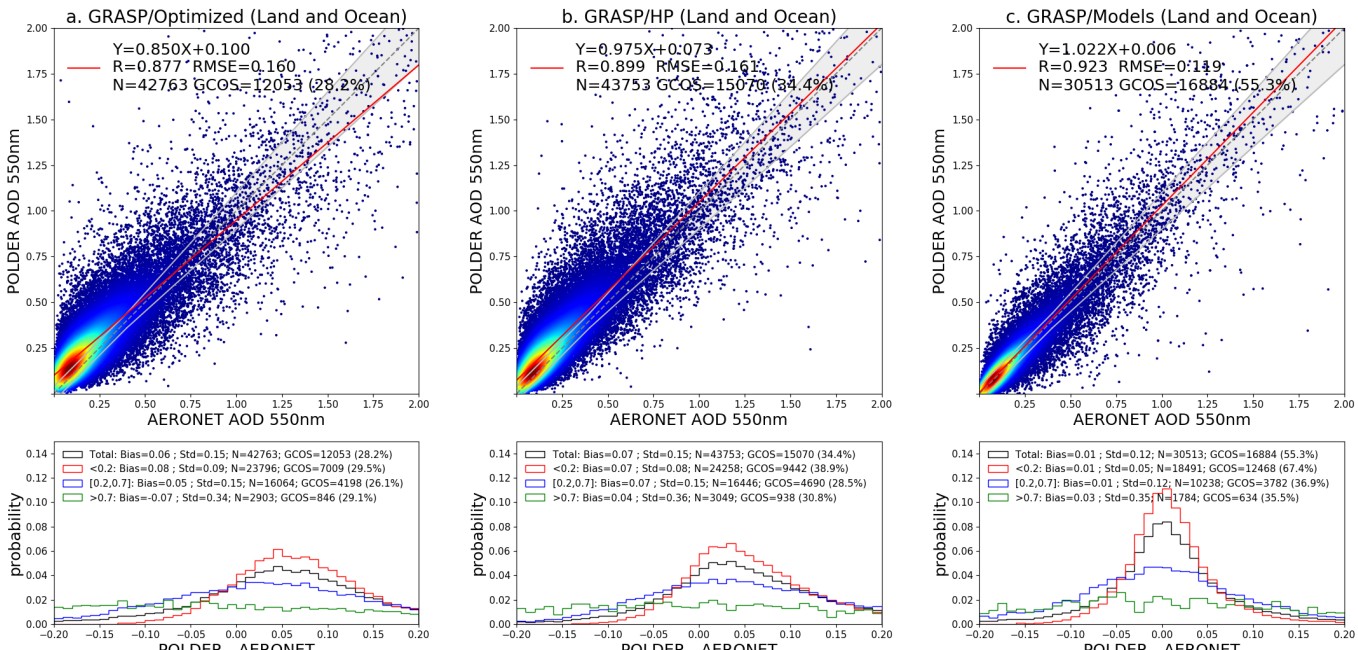

Figure 2: Evaluation of three archives PARASOL/GRASP AOD at 550 nm against AERONET, (a) GRASP/Optimized; (b) GRASP/HP; (c) GRASP/Models. The gray dashed line and the red solid line are the 1:1 reference line and the linear regression line. The gray envelope indicates GCOS requirement: max (0.04 or 0.1AOD). The probability density functions of differences (POLDER-AERONET) are present in the lower panel. The black, red, blue and green solid lines indicate all AOD conditions: any AOD, AOD<0.2, 0.2≤AOD≤0.7 and AOD>0.7 respectively.



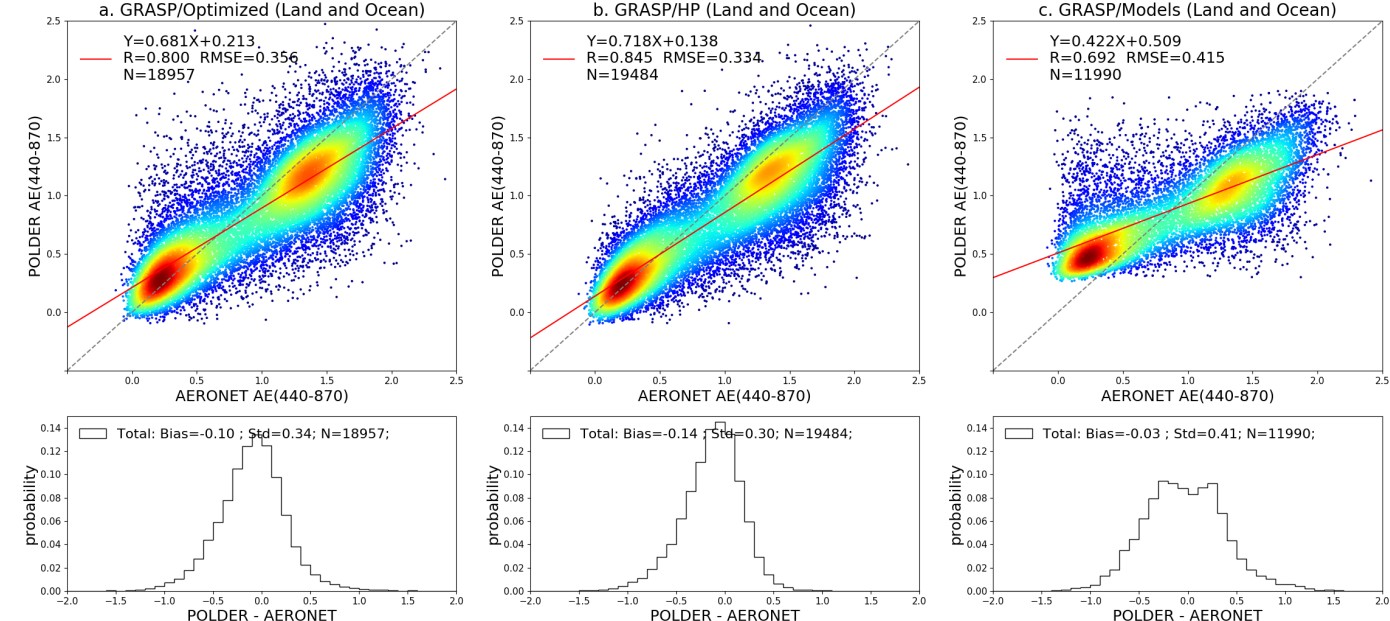


**Figure 3: Evaluation of all archive PARASOL/GRASP AE (440/870) against AERONET, (a) GRASP/Optimized; (b) GRASP/HP; (c) GRASP/Models. The gray dashed line and the red solid line are the 1:1 reference line and the linear regression line. The probability density functions of differences (POLDER-AERONET) are present in the lower panel.**



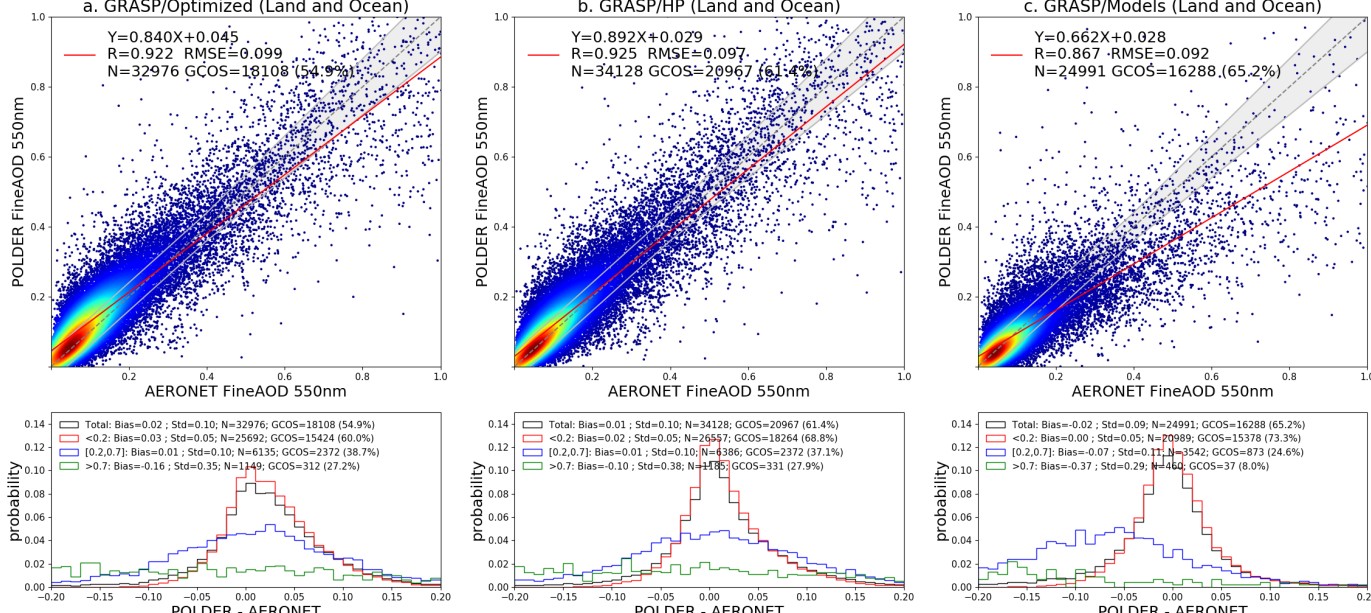

**Figure 4: Evaluation of all archive PARASOL/GRASP AODF at 550 nm with AERONET SDA AODF, (a) GRASP/Optimized; (b) GRASP/HP; (c) GRASP/Models. The gray dashed line and the red solid line are the 1:1 reference line and the linear regression line. The gray envelope indicates GCOS requirement applied for AODF: max (0.04 or 0.1AODF). The probability density functions of differences (POLDER-AERONET) are present in the lower panel. The black, red, blue and green solid lines indicate all AODF conditions: any AODF, AODF<0.2, 0.2≤AODF≤0.7 and AODF>0.7 respectively.**

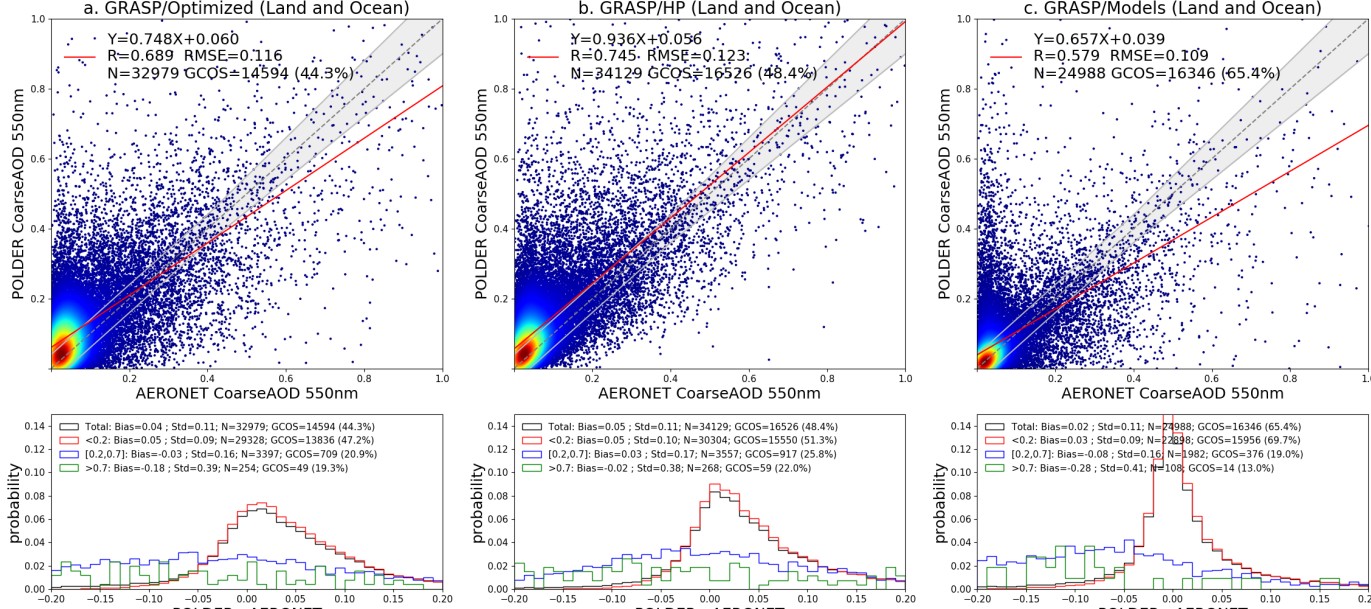

**Figure 5: The same as Figure 4, but for AODC at 550 nm.**

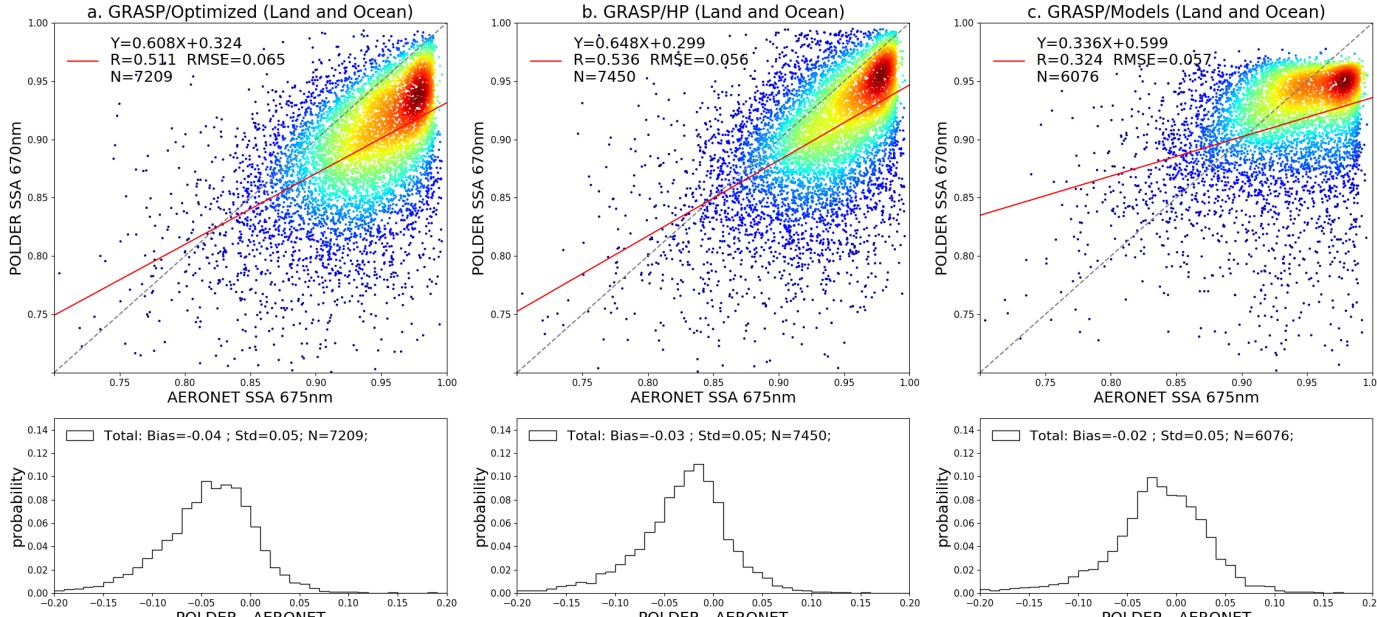

**Figure 6: Evaluation of all archive PARASOL/GRASP SSA at 670 nm with AERONET SSA at 675 nm, (a) GRASP/Optimized; (b) GRASP/HP; (c) GRASP/Models. The gray dashed line and the red solid line are the 1:1 reference line and the linear regression line.**

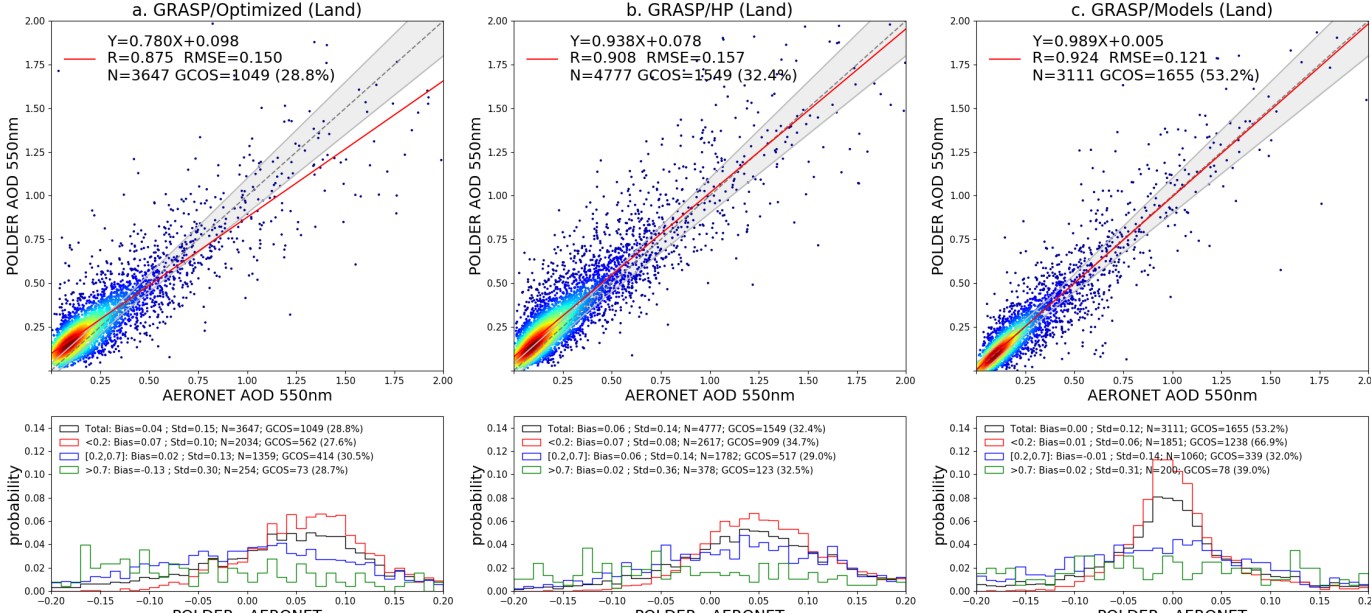

**Figure 7: Validation of PARASOL/GRASP AOD at 550 nm over land in 2008, (a) GRASP/Optimized; (b) GRASP/HP; (c) GRASP/Models. The gray dashed line and the red solid line are the 1:1 reference line and the linear regression line. The gray envelope indicates GCOS requirement: max (0.04 or 0.1AOD). The probability density functions of differences (POLDER-AERONET) are present in the lower panel. The black, red, blue and green solid lines indicate all AOD conditions: any AOD, AOD<0.2, 0.2≤AOD≤0.7 and AOD>0.7 respectively.**



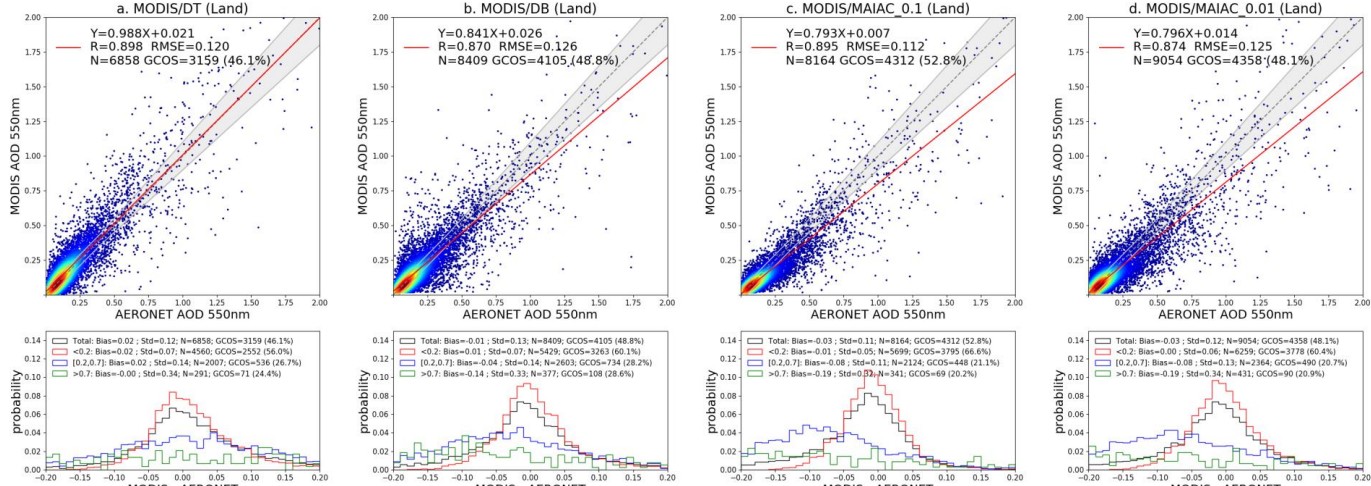

**Figure 8: Validation of MODIS AOD at 550 nm over land in 2008, (a) DT; (b) DB; (c) MAIAC_0.1; (d) MAIAC_0.01. The gray dashed line and the red solid line are the 1:1 reference line and the linear regression line. The gray envelope indicates GCOS requirement: max (0.04 or 0.1AOD). The probability density functions of differences (MODIS-AERONET) are present in the lower panel. The black, red, blue and green solid lines indicate all AOD conditions: any AOD, AOD<0.2, 0.2≤AOD≤0.7 and AOD>0.7 respectively.**



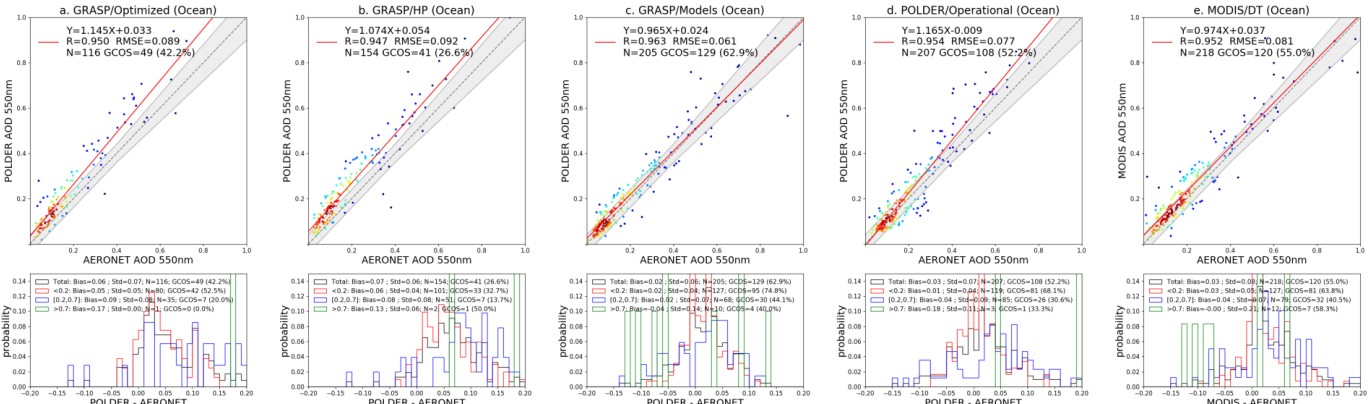

**Figure 9: Validation of PARASOL/GRASP, PARASOL/Operational and MODIS/DT AOD at 550 nm over ocean in 2008. (a) GRASP/Optimized; (b) GRASP/HP; (c) GRASP/Models; (d) Operational; (e) DT. The gray dashed line and the red solid line are the 1:1 reference line and the linear regression line. The gray envelope indicates GCOS requirement: max (0.04 or 0.1AOD). The probability density functions of differences (Satellite-AERONET) are present in the lower panel. The black, red, blue and green solid lines indicate all AOD conditions: any AOD, AOD<0.2, 0.2≤AOD≤0.7 and AOD>0.7 respectively.**





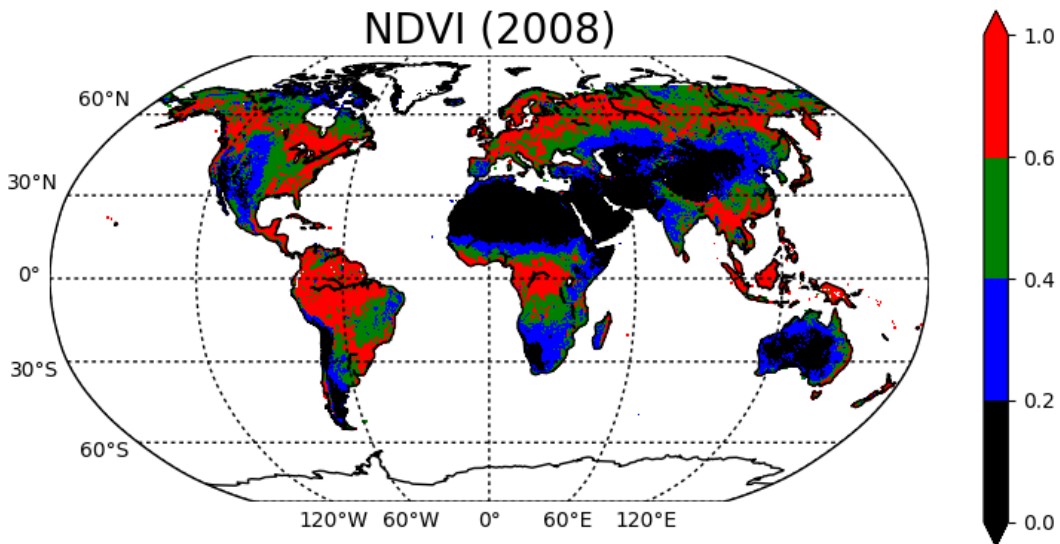


**Figure 10: Spatial distribution of annual mean NDVI for 2008 from GRASP/Models L3 products.**



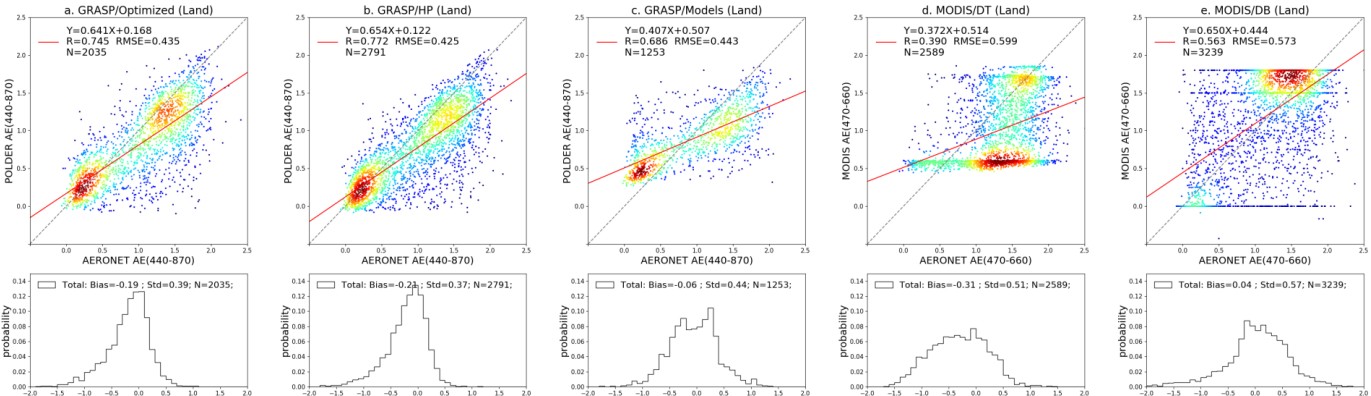

**Figure 11: Validation of PARASOL/GRASP (a. GRASP/Optimized, b. GRASP/HP and c. GRASP/Models) and MODIS (d. DT and e. DB) AE over land in 2008. The gray dashed line and the red solid line are the 1:1 reference line and the linear regression line. The probability density functions of differences (Satellite-AERONET) are present in the lower panel.**



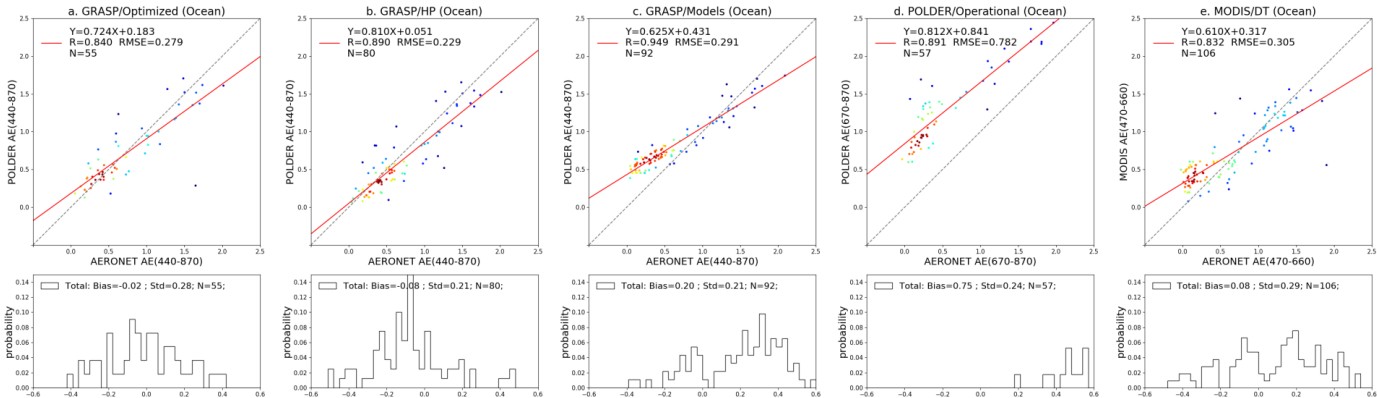

**Figure 12: Validation of PARASOL/GRASP (a. GRASP/Optimized; b. GRASP/HP and c. GRASP/Models), (d) PARASOL/Operational**
**and (e) MODIS/DT AE over ocean in 2008. The gray dashed line and the red solid line are the 1:1 reference line and the linear**
**regression line. The probability density functions of differences (Satellite-AERONET) are present in the lower panel.**



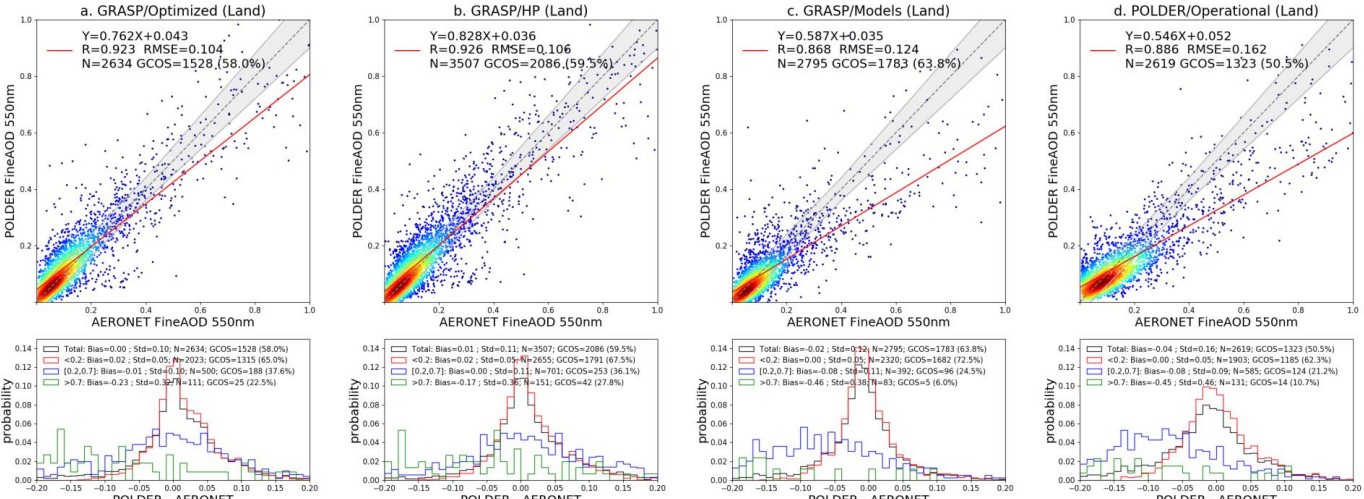

**Figure 13: Validation of PARASOL/GRASP (a. GRASP/Optimized; b. GRASP/HP and c. GRASP/Models) and (d) PARASOL/Operational fine mode AOD at 550 nm over land in 2008. The gray envelope indicates GCOS requirement applied for AODF: max (0.04 or 0.1AODF). The probability density functions of differences (POLDER-AERONET) are present in the lower panel. The black, red, blue and green solid lines indicate all AODF conditions: any AODF, AODF<0.2, 0.2≤AODF≤0.7 and AODF>0.7 respectively.**

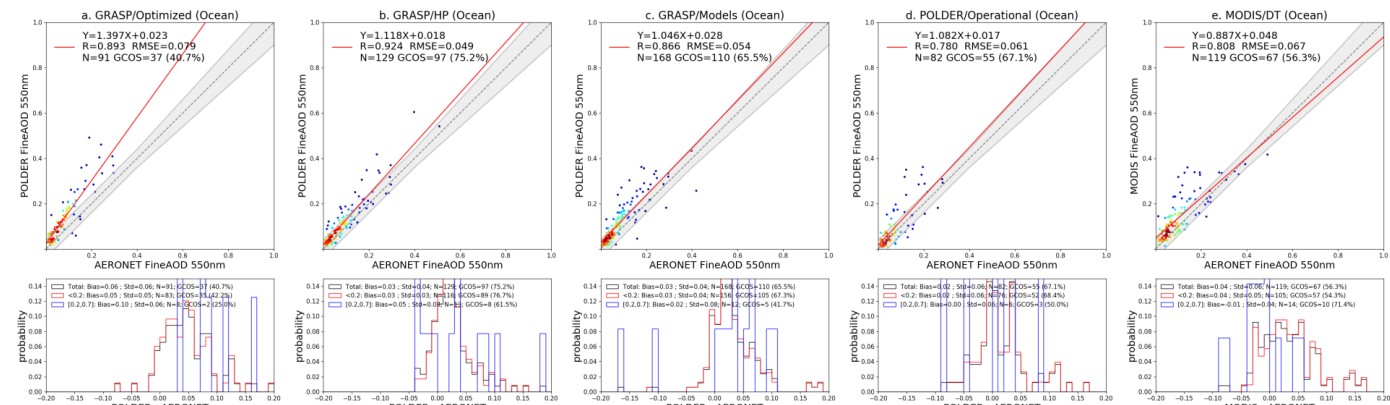


**Figure 14: Validation of PARASOL/GRASP (a. GRASP/Optimized; b. GRASP/HP and c. GRASP/Models), (d) PARASOL/Operational and (e) MODIS/DT fine mode AOD at 550 nm over ocean in 2008. The gray envelope indicates GCOS requirement applied for AODF: max (0.04 or 0.1AODF). The probability density functions of differences (Satellite-AERONET) are present in the lower panel. The black, red, blue and green solid lines indicate all AODF conditions: any AODF, AODF<0.2, 0.2≤AODF≤0.7 and AODF>0.7 respectively.**




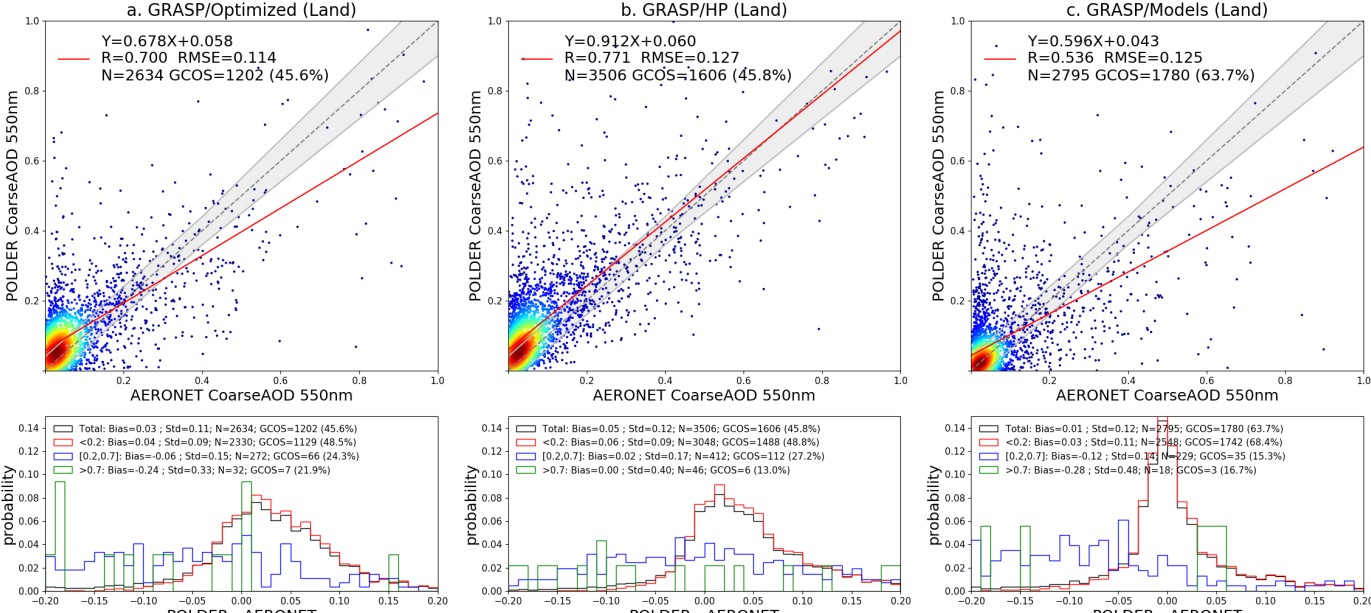

**Figure 15: Validation of PARASOL/GRASP (a. GRASP/Optimized; b. GRASP/HP and c. GRASP/Models) coarse mode AOD at 550 nm over land in 2008. The gray envelope indicates GCOS requirement applied for AODC: max (0.04 or 0.1AODC). The probability density functions of differences (POLDER-AERONET) are present in the lower panel. The black, red, blue and green solid lines indicate all AODC conditions: any AODC, AODC<0.2, 0.2≤AODC≤0.7 and AODC>0.7 respectively.**




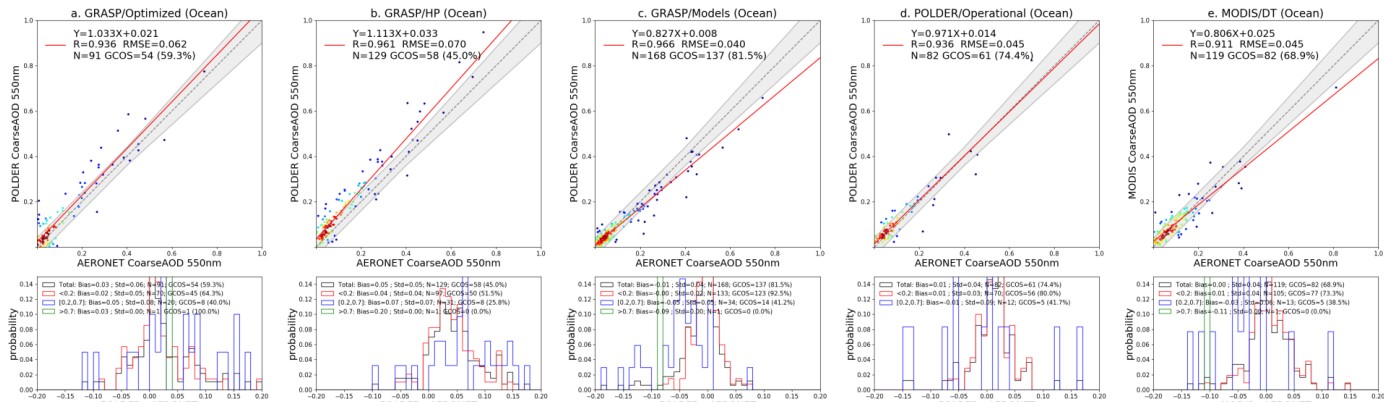

**Figure 16: Validation of PARASOL/GRASP (a. GRASP/Optimized; b. GRASP/HP and c. GRASP/Models), (d) PARASOL/Operational and (e) MODIS/DT coarse mode AOD at 550 nm over ocean in 2008. The gray envelope indicates GCOS requirement applied for AODC: max (0.04 or 0.1AODC). The probability density functions of differences (POLDER-AERONET) are present in the lower panel. The black, red, blue and green solid lines indicate all AODC conditions: any AODC, AODC<0.2, 0.2≤AODC≤0.7 and AODC>0.7 respectively.**



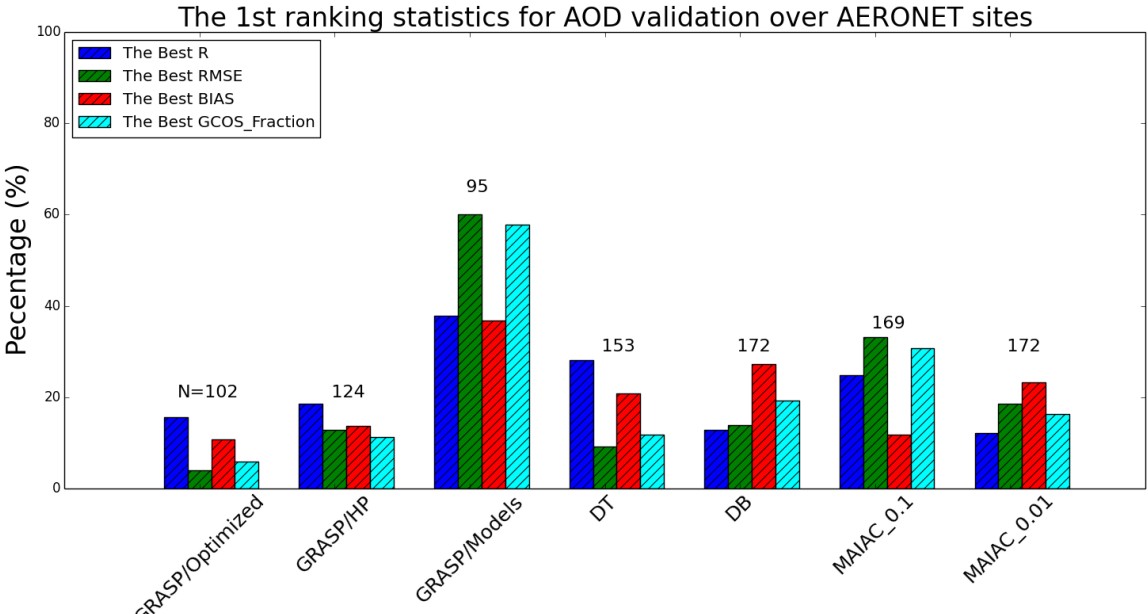

**Figure 17: Percentage of the AERONET sites where each product shows the best statistical metric (R, RMSE, BIAS and GCOS Fraction) between 7 PARASOL/GRASP and MODIS AOD products. The number on top of each product is the number of sites where this product has sufficient matchup points for the comparison.**



**Figure 18: Maps showing statistical metrics (a) R; (b) RMSE; (c) BIAS; (d) GCOS Fraction (%) for the best performed AOD products (1st ranking statistics among 7 PARASOL/GRASP and MODIS products) over each AERONET site. Note that only the 1st ranking statistics over each site are present in the maps.**



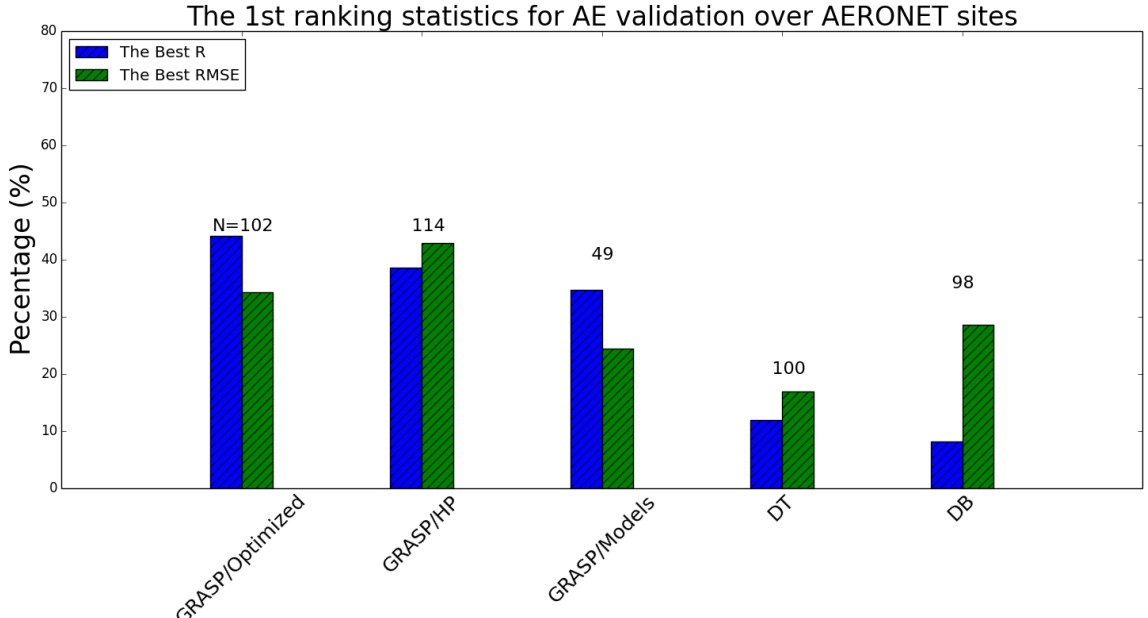

**Figure 19: Percentage of the AERONET sites where each product shows the best statistical metric (R and RMSE) between 5 PARASOL/GRASP and MODIS AE products. The number on top of each product is the number of sites where this product has sufficient matchup points for the comparison.**

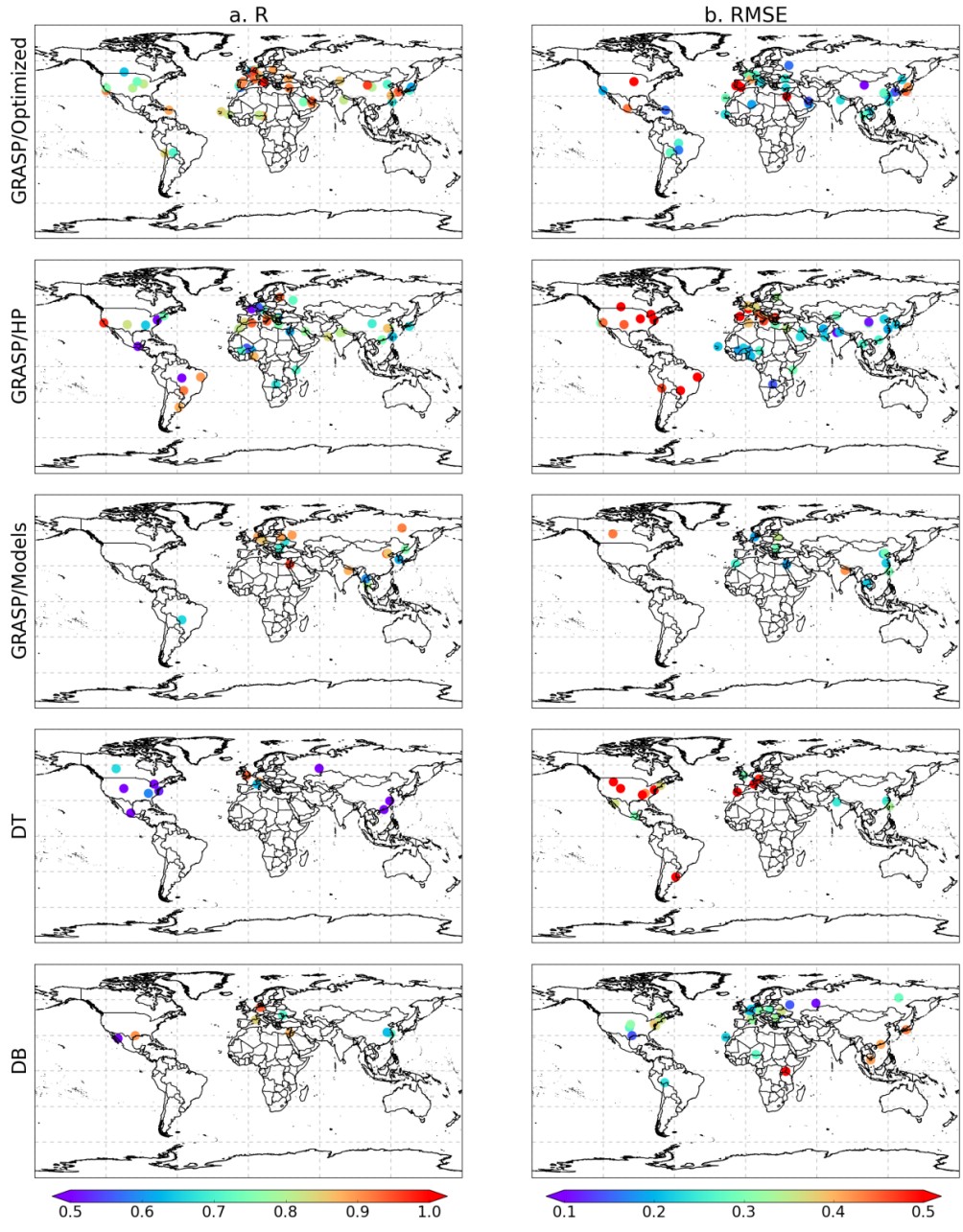

**Figure 20: Maps showing statistical metrics (a) R; (b) RMSE; for the best performed AE products (1st ranking statistics among 5 PARASOL/GRASP and MODIS products) over each AERONET site. Note that only the 1st ranking statistics over each site are present in the maps.**

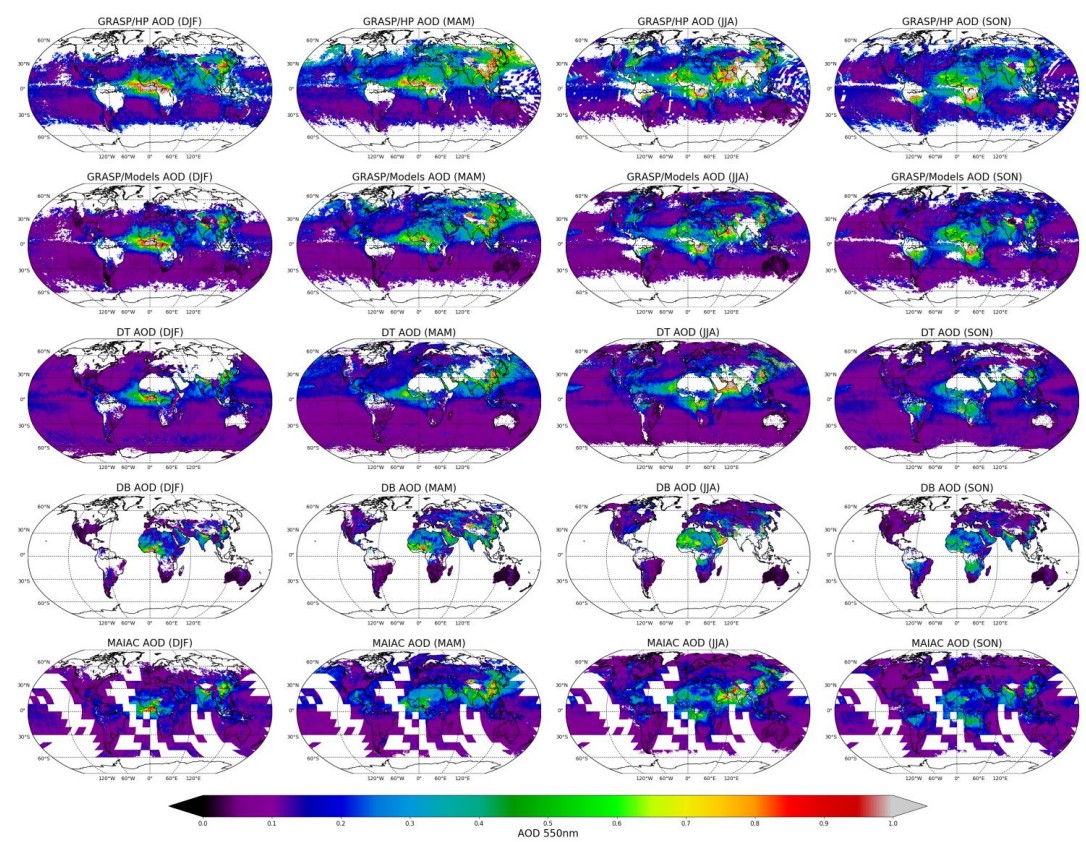

**Figure 21: Spatial distribution of 0.1° x 0.1° seasonal AOD (550 nm) from PARASOL (GRASP/HP and GRASP/Models) and MODIS (DT, DB, and MAIAC) products. DJF – December / January / February; MAM – March / April / May; JJA – June / July / August; SON – September / October / November.**

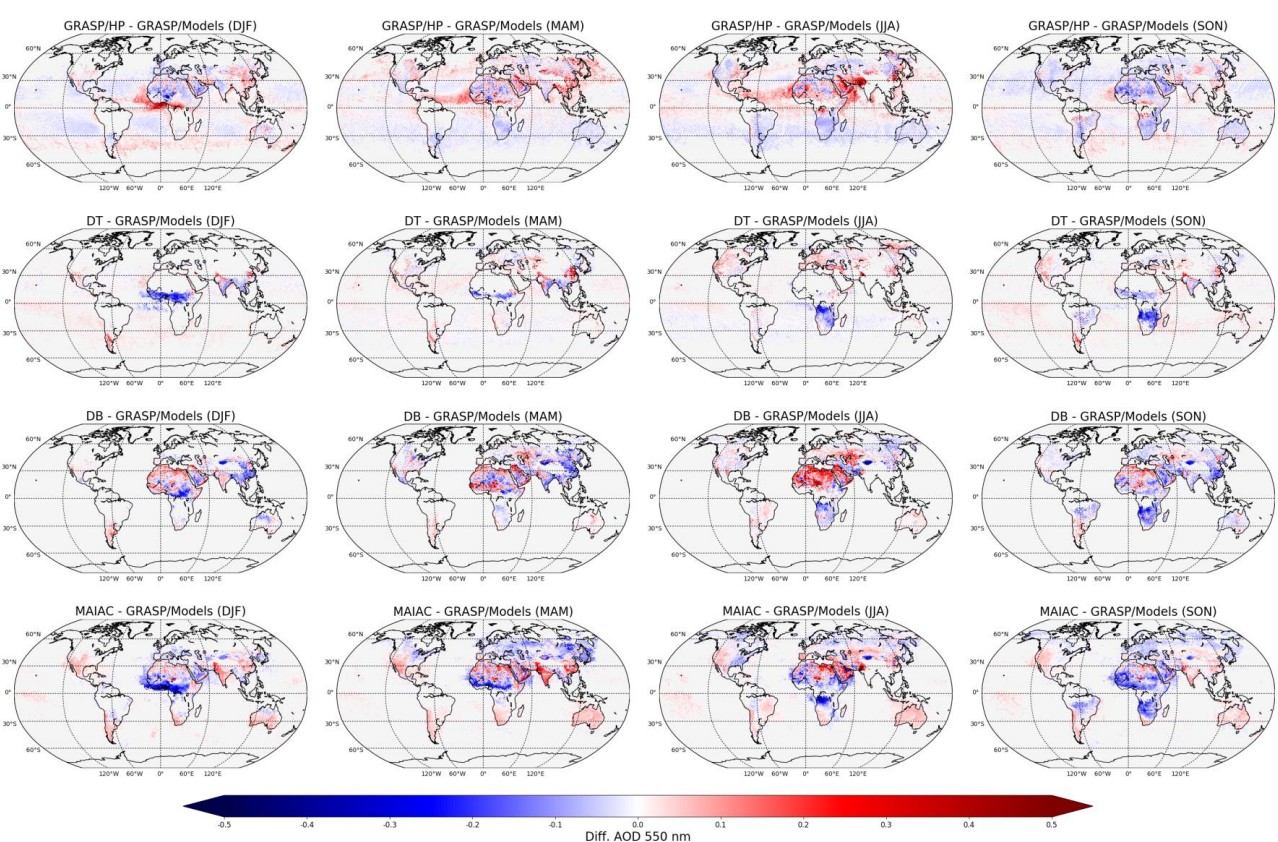

**Figure 22: Spatial distribution of 0.1° x 0.1° seasonal AOD (550 nm) differences between PARASOL and MODIS aerosol products, referenced to GRASP/Models.**

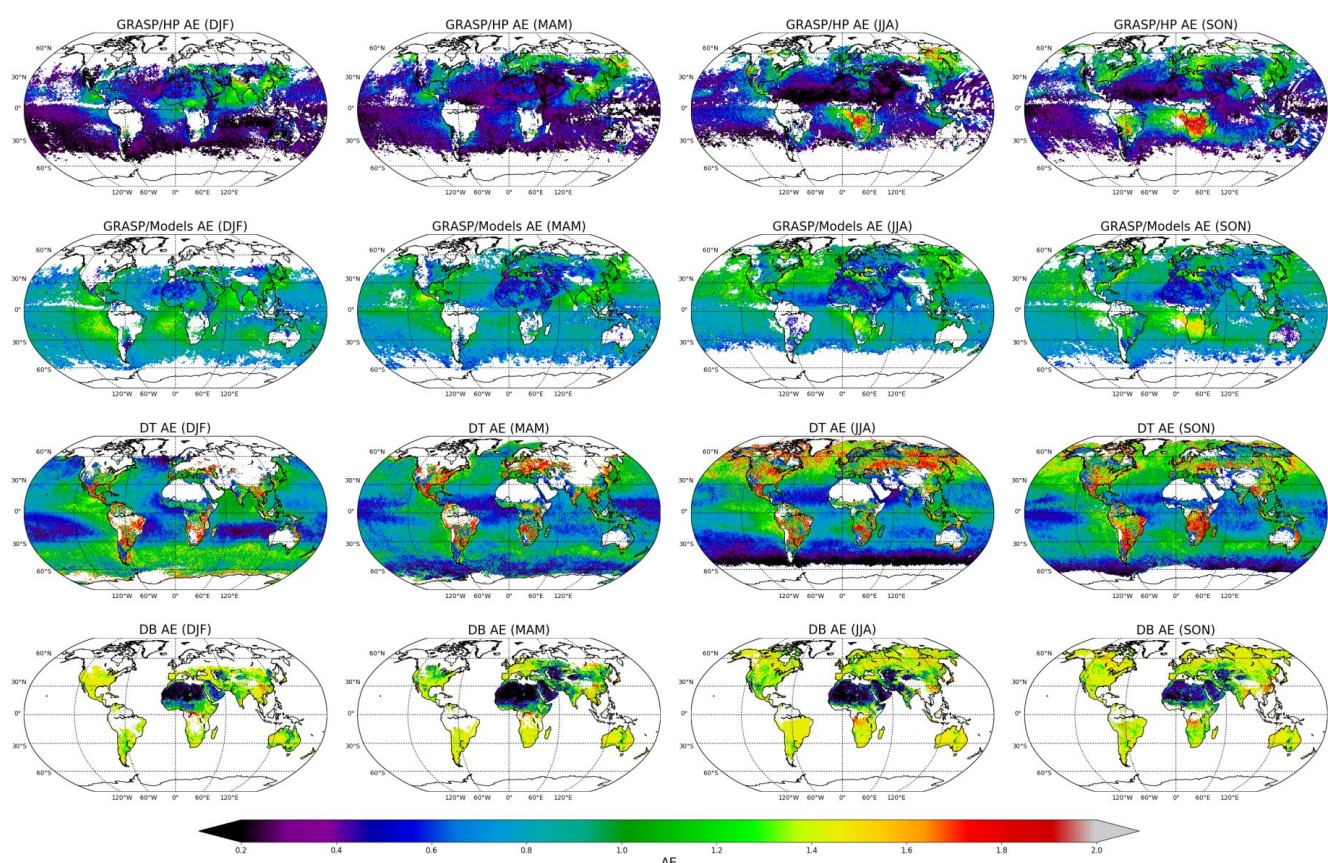

**Figure 23: Spatial distribution of 0.1° x 0.1° seasonal AE from PARASOL (GRASP/HP and GRASP/Models) and MODIS (DT and DB) products.**

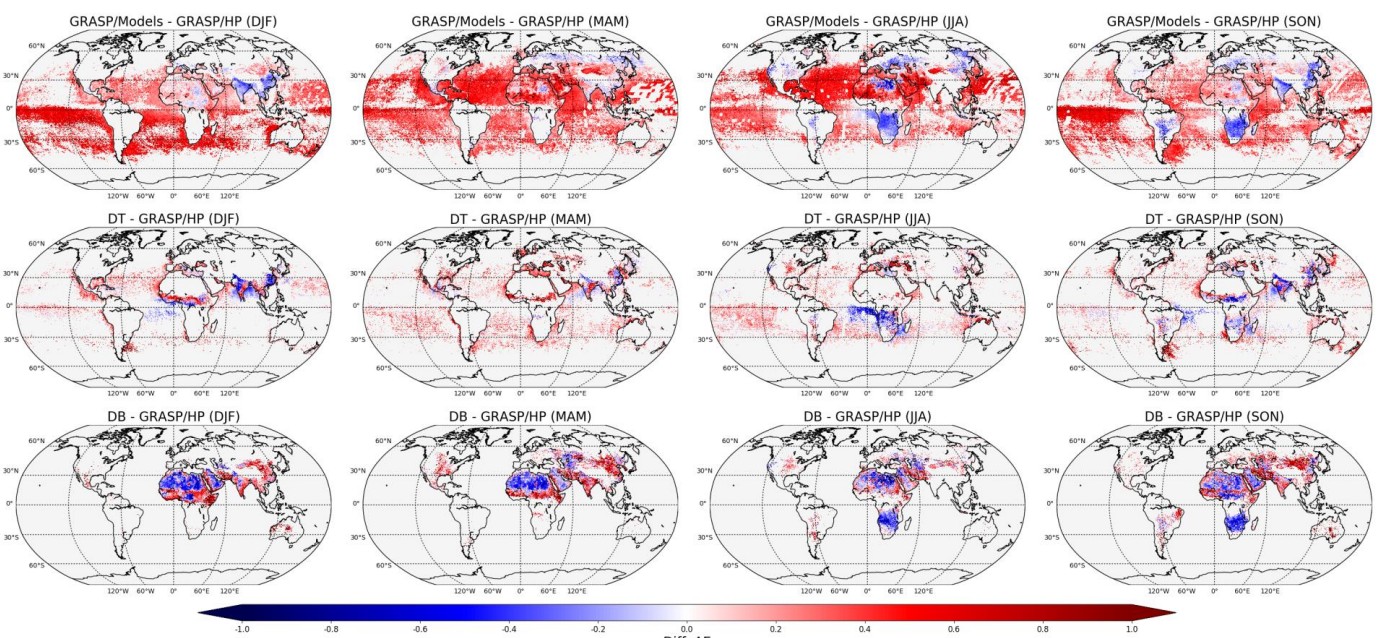

**Figure 24: Spatial distribution of seasonal AE differences between PARASOL and MODIS aerosol products, referenced to GRASP/HP.**



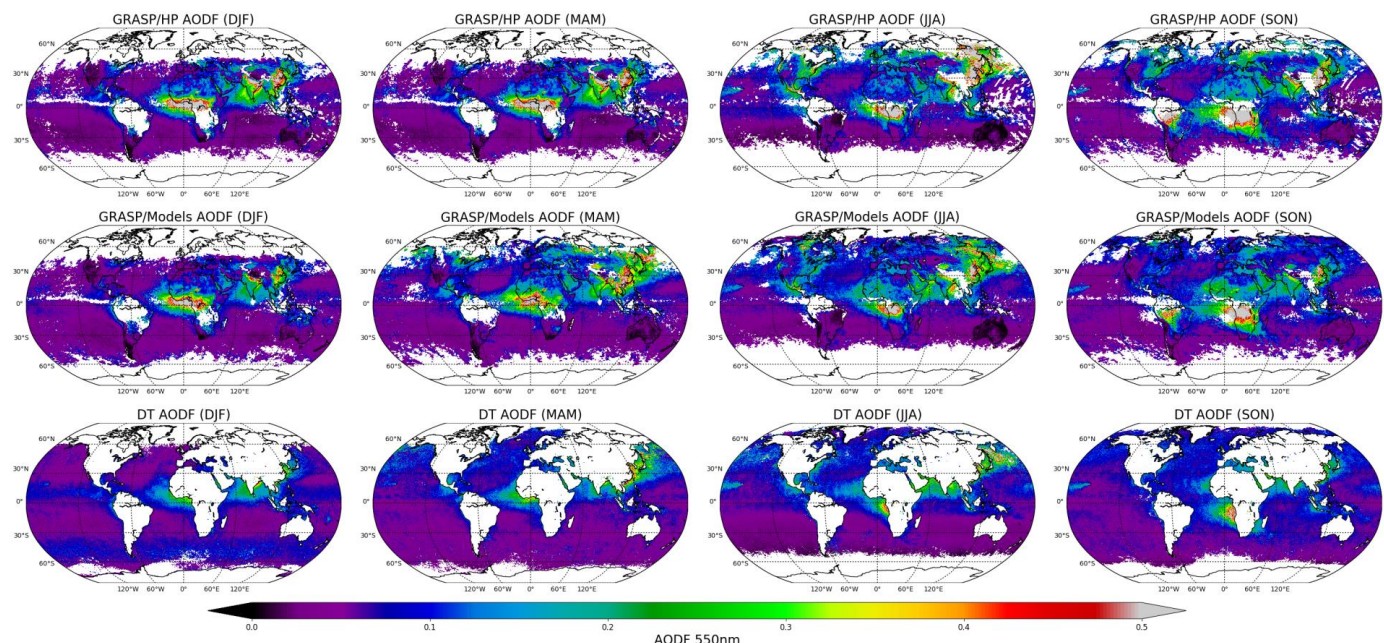

**Figure 25: Spatial distribution of 0.1° x 0.1° seasonal AODF (550 nm) from PARASOL (GRASP/HP and GRASP/Models) and MODIS (DT) products.**



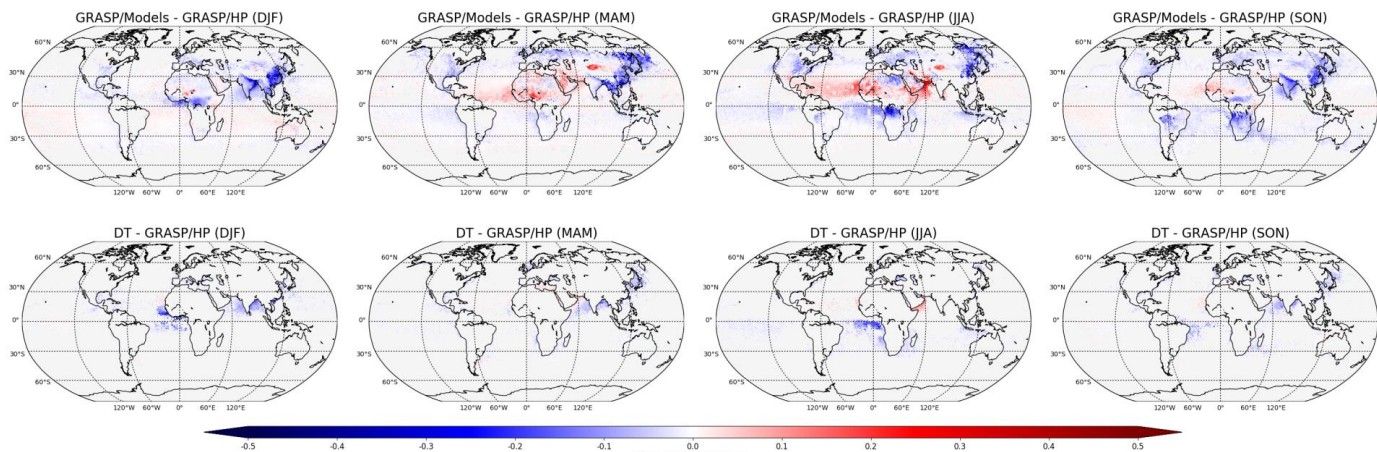

**Figure 26: Spatial distribution of seasonal AODF (550 nm) differences between PARASOL and MODIS aerosol products, referenced to GRASP/HP.**

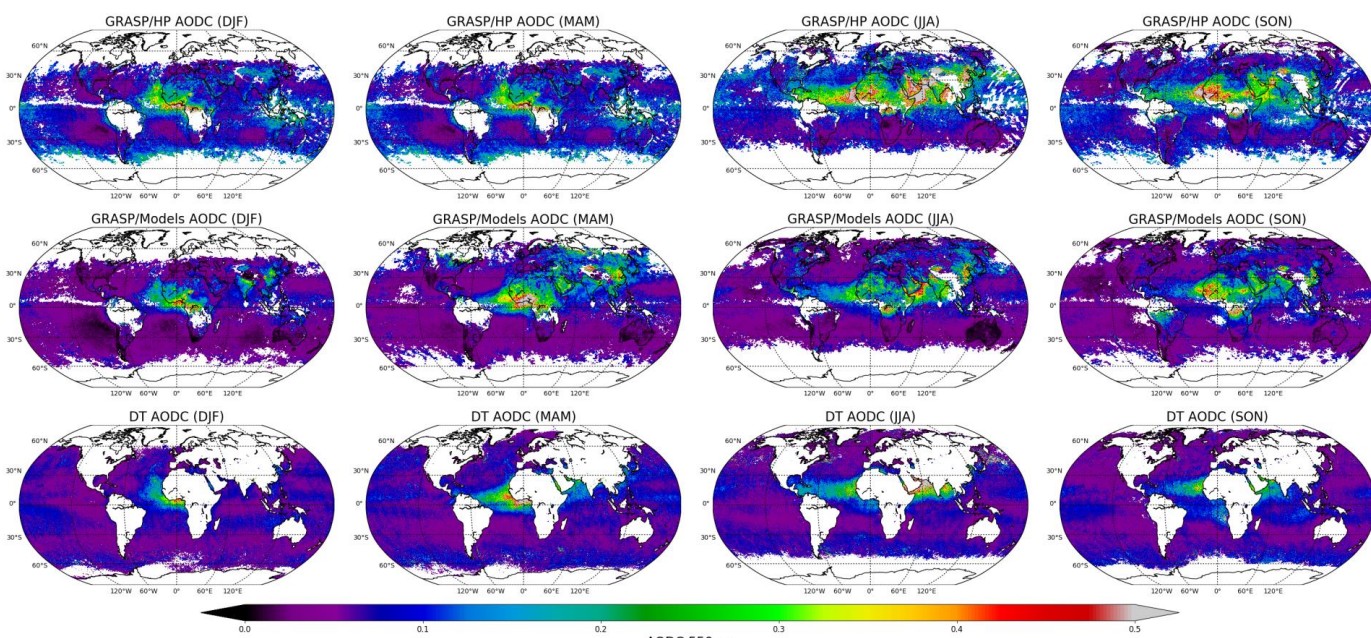

**Figure 27: The same as Figure 25, but for AODC (550 nm).**



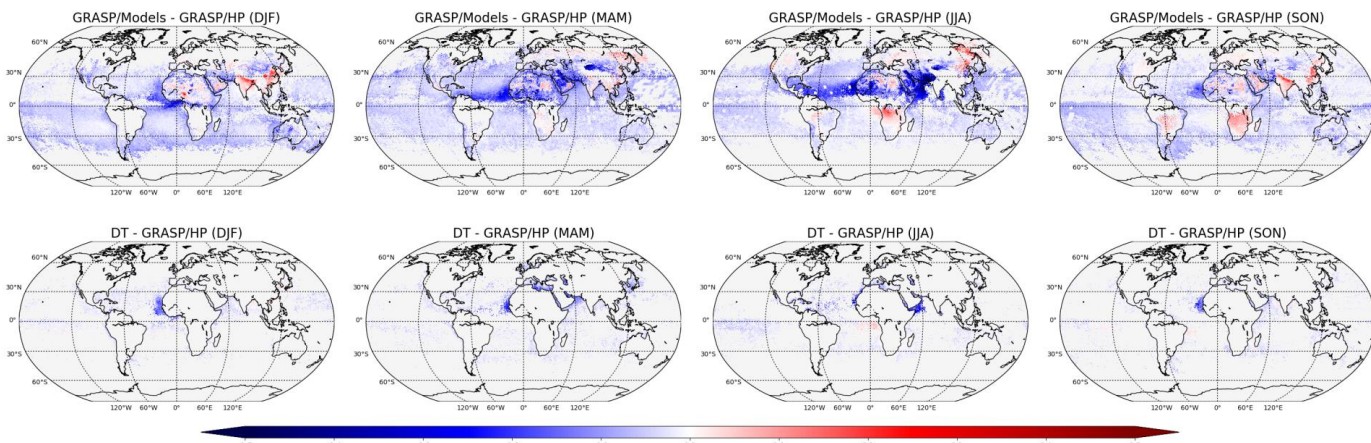

**Figure 28: The same as Figure 26, but for AODC (550 nm).**
