# Peer review of "Validation of GRASP algorithm product from POLDER/PARASOL data and assessment of multi-angular polarimetry potential for aerosol monitoring"

_Earth System Science Data, 2020_

## Referee Comment (RC1) · Otto Hasekamp (Referee) · 8 Sep 2020

The paper describes a thorough evaluation of aerosol products retrieved by the GRASP algorithm (in different configurations) from POLDER-3/PARASOL. First, a comparison to AERONET for the full data set is presented. Second, a comparison with MODIS aerosol products is performed. It is concluded that the GRASP/Models AOD product is at least as good as (and probably better than) the MODIS AOD products and that the GRASP/HP product is superior for retrieving SSA and AE.

[Figure]

Overall, the paper is well written and the conclusions are sound. The part on the comparison with MODIS is quite detailed and sometimes a bit hard to follow (because of the comparison of 3 GRASP products with 3 MODIS products). I think this part can be shortened by removing the part of fine- and coarse mode AOD as I believe the AOD+AE comparison already tells the story.

I recommend publication of this paper after addressing my comments I added to the pdf file of the manuscript, most of which are minor.

Two comments I'd like to highlight here: - It seems that the GRASP/Models product has significantly less valid retrievals than the GRASP/HP product (∼31000 vs ∼44000). What is the reason? Is the filter for GRASP/Models stricter? This is not clear from the text (in fact the opposite is suggested). May this be the reason for the better performance? Some discussion is needed here. - The evaluation puts large focus on the correlation coefficient when comparing the performance of different products. This is not always a good metric because it is heavily influenced by the range, i.e. a limited number of points at the end of the range can have large effect on the correlation. I recommend to put more emphasis on other metric such as RMSE and MAE (Mean Absolute Error).

Please also note the supplement to this comment:
https://essd.copernicus.org/preprints/essd-2020-224/essd-2020-224-RC1-supplement.pdf

**Supplement:**

[revised manuscript text omitted]

---

## Referee Comment (RC2) · Kirk Knobelspiesse (Referee) · 9 Sep 2020

Review of https://doi.org/10.5194/essd-2020-224 Chen et al., Validation of GRASP algorithm product from POLDER/PARASOL data and assessment of multi-angular polarimetry potential for aerosol monitoring

I commend the authors for what is clearly a comprehensive analysis of POLDER/PARASOL aerosol data products. Some version of this paper should be published, but I have a number of somewhat serious comments and concerns.

[Figure]

First and foremost, I'm not sure this manuscript is within the scope of ESSD. The "Aims and scope" portion of the ESSD website (https://www.earth-system-science-data.net/about/aims_and_scope.html) says:

"Earth System Science Data (ESSD) is an international, interdisciplinary journal for the publication of articles on original research data (sets), furthering the reuse of high-quality data of benefit to Earth system sciences. The editors encourage submissions on original data or data collections which are of sufficient quality and have the potential to contribute to these aims.

The journal maintains sections for regular-length articles, brief communications (e.g. on additions to data sets) and commentaries, as well as review articles and special issues.

Articles in the data section may pertain to the planning, instrumentation, and execution of experiments or collection of data. Any interpretation of data is outside the scope of regular articles. Articles on methods describe nontrivial statistical and other methods employed (e.g. to filter, normalize, or convert raw data to primary published data) as well as nontrivial instrumentation or operational methods. Any comparison to other methods is beyond the scope of regular articles."

I would think that an ESSD style manuscript on this topic would simply describe the various GRASP algorithms (briefly) and where they are archived. The majority of the paper is indeed comparisons to other data sets. I think it is far more appropriate for Atmospheric Measurement Techniques, for example (to pick another Copernicus Journal). I imagine this is something the editor needs to weigh in upon, but I wouldn't look to ESSD for this type of manuscript.

Secondly, about the length. The manuscript review version is 108 pages long with 23 tables and 28 figures. You've chosen an extensive set of data to compare and contrast the various versions of POLDER data to the various versions of MODIS and then again to AERONET. What I'm missing is a concise set of objectives and how those are met.

While I'm sure they are in the manuscript, they are lost among the noise. Because of the scale of the task, you need to be creative in finding ways to condense all of this analysis into something that easily and simply supports your work.

Given the above two points, you may consider splitting this manuscript. For example, you could make an ESSD manuscript that describes basics of the dataset and its creation and archive location. It would point to one (or more) manuscripts that contain analysis. One could be on continuity with MODIS (and VIIRS?) and another on the full set of PARASOL products and comparison to AERONET. This is just a suggestion.

Since the core of this manuscript is comparisons of other datasets, the choice of statistical metrics is very important. Table 3, for example, shows the use of nine different metrics, although Pearson's linear correlation coefficient is most commonly employed in the text. Unfortunately, some of these metrics, and especially the linear correlation coefficient, are not suitable for use on non-gaussian distributed data. The correlation coefficient is an expression of association, not agreement (Altman and Bland, 1983 and Bland and Altman, 1986), and is subject to numerical distribution, outliers, and sample range. So, for example, the R values for SSA are lower, but is that because of the lower success of the PARASOL retrieval (what you want to know) or because of the truncated numerical distribution of SSA which makes it non-gaussian distributed? Additionally, what threshold of R can be considered a success? To that end, I think your metric for percentage within the GCOS requirements is a much more appropriate measure, and should instead be emphasized, although you need to take care to account for measurement uncertainty in both POLDER and MODIS or AERONET. Seegers et al. 2018 is a nice overview of these issues in ocean color data products, but equally appropriate here. They also identify regression slope and root mean square error as problematic, while noting that mean bias (which you use) and mean absolute error as appropriate. Bland and Altman suggest something similar, also recommending the pairwise mean bias and the "limits of agreement" which is similar to the mean absolute error. Variable measurement uncertainty can also be incorporated into these

techniques (Knobelspiesse et al, 2019) which addresses the salient question "Do measurements agree to within stated uncertainties?" Ultimately, you should revise (and perhaps simplify) the metrics you use to assess your results.

Altman, D. G. and Bland, J. M.: Measurement in medicine: the analysis of method comparison studies, The statistician, 307–317 , 1983.

Bland, J. M. and Altman, D.: Statistical methods for assessing agreement between two methods of clinical measurement, The lancet, 327(8476), 307–310 , 1986.

Knobelspiesse, K., Tan, Q., Bruegge, C., Cairns, B., Chowdhary, J., van Diedenhoven, B., Diner, D., Ferrare, R., van Harten, G., Jovanovic, V., Ottaviani, M., Redemann, J., Seidel, F., and Sinclair, K.: Intercomparison of airborne multi-angle polarimeter observations from the Polarimeter Definition Experiment, Appl. Optics, 58(3), 650–669 , 2019.

Seegers, B. N., Stumpf, R. P., Schaeffer, B. A., Loftin, K. A., and Werdell, P. J.: Performance metrics for the assessment of satellite data products: an ocean color case study, Optics express, 26(6), 7404–7422 , 2018.

Co author Sayer has a publication about the numerical distribution of AOD and its impact on averages of data which is relevant to your matchup methodology and your use of Level 3 products. Were you calculating arithmetic or geometric means? I assume they are arithmetic since you don't mention otherwise, however this can in some cases cause an artificial bias in the results.

Sayer, A. M. and Knobelspiesse, K. D.: How should we aggregate data? Methods accounting for the numerical distributions, with an assessment of aerosol optical depth, Atmos. Chem. Phys., 19(23), 15023–15048 , https://doi.org/10.5194/acp-19-15023-2019, 2019.

I'm a little surprised that you make no mention of the retrieval algorithms for PARASOL (Hasekamp et al. 2011). In the beginning of section 2.1 you mention "A unique aspect

of GRASP is that it can perform radiative transfer (RT) computations fully accounting for multiple interactions of the scattered solar light in the atmosphere, and that it can perform it online without the use of traditional LUTs." You show that GRASP is an extremely power retrieval algorithm, but it is certainly not unique in its use of iterative RT computations, and I find it problematic that you make this claim and do not mention similar algorithms. Without acknowledging that there are other algorithms, even for POLDER, the manuscript sounds more like a sales pitch for GRASP and less an dispassionate piece of peer reviewed literature. In addition to Hasekamp et al, 2011, which is applied to POLDER/PARASOL, many others come to mind including those listed below.

Di Noia, A., Hasekamp, O. P., Wu, L., van Diedenhoven, B., Cairns, B., and Yorks, J. E.: Combined neural network/Phillips–Tikhonov approach to aerosol retrievals over land from the NASA Research Scanning Polarimeter, Atmos. Meas. Tech., 10(11), 4235–4252 , https://doi.org/10.5194/amt-10-4235-2017, 2017.

Fu, G. and Hasekamp, O.: Retrieval of aerosol microphysical and optical properties over land using a multimode approach, Atmos. Meas. Tech., 11(12), 6627–6650 , https://doi.org/10.5194/amt-11-6627-2018, 2018.

Gao, M., Zhai, P.-W., Franz, B., Hu, Y., Knobelspiesse, K., Werdell, P. J., Ibrahim, A., Xu, F., and Cairns, B.: Retrieval of aerosol properties and water-leaving reflectance from multi-angular polarimetric measurements over coastal waters, Optics express, 26(7), 8968–8989 , 2018.

Hasekamp, O. P. and Landgraf, J.: Retrieval of aerosol properties over land surfaces: capabilities of multiple-viewing-angle intensity and polarization measurements, Appl. Optics, 46(16), 3332–3344 , 2007.

Hasekamp, O. P., Litvinov, P., and Butz, A.: Aerosol properties over the ocean from PARASOL multiangle photopolarimetric measurements, J. Geophys. Res, 116(D14), D14204 , 2011.

Stamnes, S., Hostetler, C., Ferrare, R., Burton, S., Liu, X., Hair, J., Hu, Y., Wasilewski, A., Martin, W., van Diedenhoven, B., Chowdhary, J., Cetinic, I., Berg, L. K., Stamnes, K., and Cairns, B.: Simultaneous polarimeter retrievals of microphysical aerosol and ocean color parameters from the MAPP algorithm with comparison to high-spectral-resolution lidar aerosol and ocean products, Appl. Optics, 57(10), 2394–2413 , https://doi.org/10.1364/AO.57.002394, 2018.

Xu, F., Dubovik, O., Zhai, P.-W., Diner, D. J., Kalashnikova, O. V., Seidel, F. C., Litvinov, P., Bovchaliuk, A., Garay, M. J., van Harten, G., and Davis, A. B.: Joint retrieval of aerosol and water-leaving radiance from multi-spectral, multi-angular and polarimetric measurements over ocean, Atmospheric Measurement Techniques Discussions, 2016, 1–90 , https://doi.org/10.5194/amt-2015-394, 2016.

Xu, F., Diner, D. J., Dubovik, O., and Schechner, Y.: A Correlated Multi-Pixel Inversion Approach for Aerosol Remote Sensing, Remote Sensing, 11(7) , https://doi.org/10.3390/rs11070746, 2019.

Regarding the different versions of GRASP, I'm curious what the difference between GRASP optimized and GRASP high precision. What specific parameters are changed?

Section 2.2: PARASOL and Aqua were in the same orbit for a subset of the total PARASOL lifetime, please note this.

Paragraph starting at Line 435: so, if you find both valid ocean and land pixels in the grid box are they averaged for the estimate? If so please state this more directly.

Paragraph at line 605: For comparisons over ocean to AERONET, I presume you're using AERONET-OC (since AERONET-MAN does not include SSA retrievals). I think the distinctions of the AERONET subsets should be made more clear, and also the case that AERONET-OC is restricted to platforms near shore. So the validation against AERONET does not include deep ocean scenes. While this is similar to other studies, it should be noted.

Why is aerosol absorption optical depth included in the assessment when it is derived from two other parameters (AOD and SSA) that are already assessed?

So, the data in Figure 7 represent a subset of what is plotted in Figure 2? Could this be a bit redundant?

Do GRASP/Optimized, GRASP/HP and GRASP/Models use all the same cloud screening, quality flagging and goodness of fit metrics? I'm trying to understand the differences between the number of retrieved cases in, for example, Table 3.

I find Figure 18 (and most of the subsequent maps) to be too small to see properly. Making them larger, however, would be overwhelming. Could this be condensed to fewer maps, say by plotting sGCOS fraction alone?

---

## Referee Comment (RC3) · Gregory L. Schuster (Referee) · 17 Sep 2020

This paper applies several versions of a relatively new retrieval algorithm (GRASP/HP, GRASP/Optimized, GRASP/Models) to an existing satellite measurement archive (POLDER/PARASOL). The authors then compare the results from these algorithms to several legacy retrieval algorithms, including MODIS Dark Target (MODIS/DT), Deep Blue (MODIS/DB), Multi-Angle Implementation of Atmospheric Correction (MAIAC), the operational PARASOL product (PARASOL/Operational), and AERONET. The authors provide a large number of maps and statistics that not only inform the reader about

the performance of the GRASP products, but they also inform the reader about the performance of the legacy aerosol retrievals (e.g., Tables 9 & 10). The paper is clear and well written and I find it suitable for publication.

The paper is also quite long (48 pages of text, 23 tables, and 28 figures) and probably won't be carefully read in its entirety by anyone except the reviewers. One could paraphrase the paper as "here are some new data products, and here is how they compare to similar data products as well as the gold standard (AERONET)." Nobody will learn the machinery behind the retrievals from this paper, but there are other papers for that. One reviewer pointed out that the statistical parameters chosen for this paper are not ideal, but the authors use statistical parameters that are familiar to many readers (correlation coefficient, bias, RMSE, etc.) and common in many satellite/AERONET comparison papers. Unfortunately, the aerosol remote sensing community has not yet adopted a "skill score" for comparisons, as is sometimes used in the modeling community (Taylor, 2001).

It is important to have all of this material in one place, in my opinion, so that readers can quickly assess the relative performance of the different algorithms for the various parameters. However, hyperlinks to tables, figures, citations, and section headings would greatly improve the readability of the paper. A bookmarked Table of Contents in the sidebar would also be helpful. I had to keep two copies of the paper open on my screen – one for the text, and another for reading tables and figures. Otherwise, I would have spent as much time scrolling as reading for this paper! Hyperlinks would allow the reader to go directly to a table, and then return to the text with the "previous view" buttons. Hopefully this is something that can be accommodated in the typesetting process.

Taylor, K.: Summarizing multiple aspects of model performance in a single diagram, J. Geophys. Res., 106, 7183–7192, 2001.

SIGNIFICANT ISSUES

I noticed that the data volume for GRASP/Models is much different than the data volume for GRASP/HP and GRASP/Optimized. For instance, GRASP/Optimized shows 41,268 AOD comparisons with AERONET over land in Table 3, but GRASP/Models only shows 27,551 comparisons. However, GRASP/Model comparisons are greater than GRASP/Optimized over ocean (2064 vs 1495). These large discrepancies appear elsewhere in the paper as well. I found this quite odd, since all three retrievals use the same instrument (PARASOL). I imagine that the cloud screening procedure is identical for all three algorithms, so I suspect that GRASP/Model fails to provide a retrieval much more frequently over land than than the other two GRASP algorithms (and that GRASP/Optimized, GRASP/HP fail more frequently than GRASP/Models over ocean). This should be discussed, since GRASP/Models is lauded for its ability to retrieve AOD(550) (e.g., line 1180). The success rate of a retrieval is important to readers, too.

It is also curious that GRASP/Models did so well for AOD at nearly all wavelengths over both land and ocean (Table 3), but the AEs for GRASP/Models is significantly worse than the other GRASPs. Since AE is derived from AOD, I would have thought the retrieval that produced the best AOD at multiple wavelengths would also produce the best AE. A comment about this would be helpful.

Is there a reason for comparing AODf and AODc to the SDA extinction-based retrievals instead of using the sky scan retrievals? The sky scans are probably more accurate. Many readers (most?) won't know the methodology behind SDA and may incorrectly assume that it is derived from the sky scans. The SDA papers use the AERONET almucantar scans as a performance benchmark, so why not use the same benchmark? You're already using the AERONET sky-scans for SSA and AAOD.

Line 588:
The authors say that they are using AERONET L2 inversions, but which version of AERONET (i.e., Version 2 or Version 3)?
Lines 590-595:
This paragraph will probably confuse some people. Line 590 says that AERONET L2 inversion products require $AOD(440) > 0.4$, but the PARASOL/GRASP filtering includes much lower values, especially over ocean (the authors require PARASOL/GRASP $AOD(443) > 0.3$ over land and $AOD(443) > 0.02$ over ocean). However, since AERONET L2 requires $AOD(440) > 0.4$, many of the low PARASOL/GRASP AODs won't actually appear in the comparisons anyways. . . . Unless the authors using Level 1.5 AERONET inversions at low AOD, like some other authors? If so, what are the Level 1.5 constraints?

MINOR ISSUES
Line 35:
The links do not take me directly to the data products. www.icare.univ-lille.fr takes me to the main page, and the 2nd link on that line tries to take me to www.grasp-35, but that is a dead end.

Line 1125
Do you mean GRASP/Models instead of PARASOL/Models?

It would be interesting to repeat the AOD comparisons using AERONET's "coincident" AOD (that is, using only the AODs that are used during the sky scans). This would be interesting because the cloud screening for the sky-scan products is more comprehensive than for the direct AOD measurements, and it is possible that satellite (and model) AOD performance comparisons wrt AERONET will differ for these two datasets. If the coincident AODs comparisons are different than the "all AODs" comparisons, this could assist our thinking wrt the other sky-scan products. You are probably already set up to do this. I include this as a minor issue, though, because the paper is already too long and this should really be a topic for another day.

[Figure]

2020.

---

## Author Comment (AC1) · 7 Nov 2020

**Reviewer #1:**

The paper describes a thorough evaluation of aerosol products retrieved by the GRASP algorithm (in different configurations) from POLDER-3/PARASOL. First, a comparison to AERONET for the full data set is presented. Second, a comparison with MODIS aerosol products is performed. It is concluded that the GRASP/Models AOD product is at least as good as (and probably better than) the MODIS AOD products and that the GRASP/HP product is superior for retrieving SSA and AE.

Overall, the paper is well written and the conclusions are sound. The part on the comparison with MODIS is quite detailed and sometimes a bit hard to follow (because of the comparison of 3 GRASP products with 3 MODIS products). I think this part can be shortened by removing the part of fine- and coarse mode AOD as I believe the AOD+AE comparison already tells the story.

I recommend publication of this paper after addressing my comments I added to the pdf file of the manuscript, most of which are minor.

**Response:**

Dear Otto,

Thank you for the constructive and positive comments on our paper. Here are the point-by-point responses:

We agree that the Ångström exponent (AE) can qualitatively indicate the domination of fine or coarse model aerosol. At the same time, we would like to point out that quantitative analysis of fine/coarse mode contribution to total AOD is very challenging and unclear without fine mode AOD (AODF) and coarse mode AOD (AODC). Moreover, the AODF is a parameter of particular interest for various applied research targeting characterization of Air Quality and anthropogenic effects. For example, the Air Quality related studies of Wei et al. (2020) are focused on only AODF from POLDER and MODIS. In addition, the multi-angular polarimetry is known for high sensitivity to fine mode aerosol and the POLDER/Operational products provide AODF over land and ocean as the main "operational POLDER" products. Therefore, we consider beneficial to keep AODF and AODC analysis in our

paper. We validated the POLDER/GRASP AODF with AERONET AODF, and inter-compared with POLDER/Operational and MODIS over ocean. We believe that our analysis provide useful insights for the users of satellite AODF and AODC products.

Wei, Y., Z. Li, Y. Zhang, C. Chen, O. Dubovik, Y. Zhang, H. Xu, K. Li, J. Chen, H. Wang, B. Ge, C. Fan, "Validation of POLDER GRASP aerosol optical retrieval over China using SONET observations", J. Quant. Spectrosc. Radiait. Transfer, 246, 106931, 2020. doi:10.1016/j.jqsrt.2020.106931, 2020.

Main points:

Two comments I'd like to highlight here: - It seems that the GRASP/Models product has significantly less valid retrievals than the GRASP/HP product (~31000 vs ~44000). What is the reason? Is the filter for GRASP/Models stricter? This is not clear from the text (in fact the opposite is suggested). May this be the reason for the better performance? Some discussion is needed here.

**Response:**

Indeed, the different number of points and somewhat different approaches of quality filtering is one of the main shortcomings in our study. Reviewer #3 raised similar question. In fact, the post-processing flow (L1-L2-L3) was based on several attempts dictated by practical needs. These attempts provided us valuable inside but they could be fully evaluated only after full-scale validation. For example, the level 3 GRASP/HP and Optimized archives were generated and released much earlier than GRASP/Models. Also, we have done preprocessing of GRASP/Models over land at first and learned that screening was very conservative. Based on that we used less conservative screening for Models reprocessing over ocean. Once the products were released, they were used by many users, therefore, regenerating Level 3 products was not reasonable for this study. We are considering the harmonization of the all archives in future once time and resources allow that.

At the same time, we have looked at possible effect of applying tighter screening to HP and Optimized data. Our analysis showed that although stricter screening somewhat improves the correlations, it doesn't change conceptually the results of validations. For example, it does not improve the BIAS which is considered as a

main issue for these data sets. Some explanations of this aspect were added in the Sect 2.4 and Sect 3.1 as follow.

"*For GRASP/Models product we did not use any filter, because a stricter quality assurance filter has been applied in GRASP/Models products generation from L1 to L2 and L3 than for other GRASP datasets. In principle, the post-processing of all PARASOL/GRASP products was done in similar ways. At the same time, the L3 products were prepared and released not at the same time. For example, the L3 GRASP/HP and Optimized archives were generated and released much earlier than GRASP/Models. Therefore, the post-processing and quality screening approaches used for different data archives are not exactly the same. Unfortunately, most of the differences were identified after the release of the products, its extensive use and the full-scale validation. In these regards, the harmonization of the all archives is likely to be done in future, but it will lead in release of new products.*"

"*As described in section 2.4, the different post-processing scheme resulted in the difference for matched points between GRASP/Optimized, GRASP/HP and GRASP/Models. It can be noticed that applying a much stricter filter may improve the overall correlation against AERONET AOD for GRASP/HP and GRASP/Optimized products, but leads to significant loss of points and, most importantly, do not improve the BIAS, which is considered as a main issue for them.*"

The evaluation puts large focus on the correlation coefficient when comparing the performance of different products. This is not always a good metric because it is heavily influenced by the range, i.e. a limited number of points at the end of the range can have large effect on the correlation. I recommend to put more emphasis on other metric such as RMSE and MAE (Mean Absolute Error).

**Response:**

The idea of our analysis was to provide comprehensive characterization of the observed relationships, rather than to focus on one selected parameter. Therefore, we provided many different parameters describing the relationship and did not intend to focus on correlation coefficient. Thanks for the suggestion about evaluation metrics! We have revised the text to make more comprehensive discussion of other correlation parameters including RMSE and BIAS.

At the same time, we would like to note that RMSE and BIAS also have limitations in characterizing quality of agreement. For example, RMSE is always much lower for the dataset that include mainly small values of AOD. Therefore, the retrievals failing to report the retrieval at larger AODs have smaller RMSE than those that provide AODs with larger values.

Minor points and grammar:

Line 103: Please also cite Mishchenko and Travis, JGR, 1997 doi:10.1029/96JD02425 and Hasekamp and Landgraf, Appl. Opt., 2007, https://doi.org/10.1364/AO.46.003332)

**Response:**

Corrected.

Line 139: 910 nm

**Response:**

Corrected.

Line 159: Mention that also other MAP algorithms have been developed / are being devloped: SRON (Hasekamp et al., J. Geophys. Res., 116, D14204, https://doi.org/10.1029/2010JD015469, 2011; Fu and Hasekamp https://doi.org/10.5194/amt-11-6627-2018, 2018) JPL (Xu et al., https://doi.org/10.1002/2017jd026776, 2017; https://doi.org/10.3390/rs11070746, 2019), LaRC (Stamnes et al., https://doi.org/10.1364/AO.57.002394, 2018.) All these algorithms follow a similar principle of online RT calculations and no restriction to aerosol models.

**Response:**

We fully agree that the text of article didn't mention other advanced algorithms and therefore could be misleading. We added corresponding discussion in Sect. 2.1 as below.

[revised manuscript text omitted]

Line 210: There are also other algorithms that do that (see my comment above).
**Response:**
Corrected.

Line 307: In Introduction together with other MAP aerosol applications (see above).
**Response:**
Corrected.

Line 416: I would say for SSA/AAOD it is more comparison than validation because also AERONET has large uncertainties for these properties.

**Response:**

We agree that SSA and AAOD provided by AERONET have substantial uncertainties in some situations (such as the low-AOD case). At the same time, we consider that AERONET, so far, provides overall the most comprehensive and reliable SSA and AAOD data among all available data sources. Since AERONET retrievals rely on direct Sun observation for getting AOD, they have serious advantages over satellite data for constraining aerosol absorption retrieval.

At the same time, we do not consider our POLDER paper as a right place for the discussions, we rewrote the sentence as follows to avoid the discussion: *"AAOD and SSA products are chosen as references for satellite products comparison and evaluation."*

Line 430: How to go from 6km to 0.1 degree?

**Response:**

The gdalwarp regridding technique (https://gdal.org/programs/gdalwarp.html) is used to generate 0.1 degree products from original ~6 km retrieval. The algebraic mean is then calculated from all pixels in a grid box.

Line 440: How to interpret these numbers (residual)?

**Response:**

The relative residual is the root mean square of relative error in fitting the measurements by the algorithm.

Line 441: Not clear. Actually the GRASP/Models results seem most heavily filtered as there are much less points in the validation plots.

**Response:**

Rather elaborated filtering scheme was used to generate GRASP/Models L3 products. Hence, no additional filtering was applied to L3 GRASP/Models 0.1degree products in the validation considerations.

Line 450: So, this is 9 0.1 degree pixels?

**Response:**

Not exactly. For POLDER/GRASP and MODIS DT, DB and MAIAC_0.1, it is 9 0.1 degree pixels. While for MAIAC_0.01, 9 0.01 degree pixels are used, and 9 18.5 km pixels are for POLDER/Operational products. We now clarify this in the text.

Line 461: Why? Indications for clouds?

**Response:**

The purpose of this criterion is to remove some evident outliers, which stand out within 3x3 or 9x9 windows. It is proven to be helpful for validation. It is quite possible that the high variability in 3x3 or 9x9 windows can be related to clouds, but further investigations for supporting this idea are needed.

Line 480: I would say the AERONET uncertainty should be added quadratically ....

**Response:**

We agree that if we consider the statistical rule of adding standard deviations the quadratic addition should be done. However, GCOS criteria seem to be a criterion defined based on practical considerations rather than rigorous statistical consideration. In this regards, we simply followed the common practice and have adopted the GCOS requirements, $GCOS = max(\pm0.04, \pm0.1AOD)$, following the latest Aerosol_cci study (Popp et al., 2016).

Line 489: Why just 2008?

**Response:**

The year 2008 was chosen as an example year to comprehensive evaluation of the consistencies and differences between POLDER/GRASP and MODIS aerosol products. The observations during 2008 year contain generally good observational statistics and all types of aerosol are clearly present. The year has also been used as a reference in many evaluation studies, e.g., in Aerosol_cci study.

Line 505: But there are much less points.

**Response:**

Yes, this is correct, GRASP/Models product contains less point than GRASP/Optimized and GRASP/HP due to different filter scheme used for generation of L3 0.1degree products. This aspect has already been discussed above and a discussion is added in the Sect. 2.4 and Sect 3.1 of the revised manuscript.

Line 543: Do fine and coarse mode have the same definition on GRASP and AERONET?

**Response:**

The separation of the  fine and coarse modes in AERONET and GRASP are not exactly the same, but they provide very close results in case then both modes are separated by distinct minimum.

Line 582: So, is GRASP/Models really better? Or is it a 'lucky' compensation of errors?

**Response:**

In terms of spectral AOD products, GRASP/Models show better agreement with AERONET measurements in many senses. The biggest problem of GRASP/Optimized and GRASP/HP AOD products is a distinct BIAS. At the same time, the validation of AODF and AODC indicates that the BIAS for GRASP/Optimized and GRASP/HP comes mainly from the coarse mode, and the AODF provided by GRASP/Optimized and GRASP/HP has almost no BIAS and show statistic that even better than AODF from GRASP/Models. This observation suggests that the approach of GRASP/Optimized and GRASP/HP can likely be improved in the future.

By the way, this result shows also the importance of keeping the section of fine and coarse mode validation for helping to improve understanding of the overall retrieval performance.

Line 592: But you compare against AERONET L2 which only includes AOD > 0.4. So, I don't understand the low AOD filter over ocean. because by comparing to AERONET L2 only large AOD cases will be included in the end. Or am I missing something?

**Response:**

Yes, we are comparing against AERONET L2 inversion data, which includes only AERONET AOD>0.4. Here we use additional filter for satellite AOD (Land: AOD 443 nm>0.3; Ocean: AOD 443 nm>0.02).

Line 655: What do you mean by this?

**Response:**

We rewrote the sentence as follows. "MAIAC products cover some land-containing ocean tiles, however due to limited coverage of these retrievals we do not consider MAIAC ocean products here."

Line 717: It would be rather straightforward to compure GRASP AOD at 470 and 660 so that the exact same quantity can be compared.

**Response:**

In the paper, we intended to present the standard AE products validation for each dataset. Hence, we decided to use different wavelengths to compute AE for POLDER/GRASP and MODIS. Figure R1 shows the comparison of POLDER/GRASP AE (470/660) against with AERONET AE (470/660). This comparison is done by interpolating both POLDER and AERONET AOD to 470 and 660 nm based on the nearest available wavelengths. It presents similar performance with GRASP AE (440/870) in Figure 11.

[Figure]

Figure R1. Validation of POLDER/GRASP (a. GRASP/Optimized, b. GRASP/HP and c. GRASP/Models) AE over land in 2008.

Line 899: decrease?

**Response:**

Thanks. It should be 'decrease'. It was revised.

Line 948: I would say the other statistical parameters are more important to discuss than the correlation.

**Response:**

In the revised manuscript, we have adjusted the discussion with including more comprehensive discussion of other parameters, e.g. R, RMSE, BIAS, GCOS, etc.

Line 1055: This needs more discussion. The bias between GRASP/Models and GRASP/HP is larger for most of the globe than between GRASP/HP and DT, right? This is quite unexpected given the AERONET comparison.

**Response:**

In Figure 24, indeed the differences between GRASP/Models and GRASP/HP are larger than DT and GRASP/HP. Table 18 confirms the difference (GRASP/Models - GRASP/HP) is 0.12 over land, and DT - GRASP/HP is 0.11; over ocean, the difference (GRASP/Models - GRASP/HP) is 0.35, and DT - GRASP/HP is 0.25. This is due to the phenomenon that GRASP/Models tend to overestimate AE for coarse particles (AE<~1.0) (see Figure 3). While in terms of correlation, GRASP/Models and GRASP/HP (Table 18) agree well (R>0.7) over both land and ocean, which can be interpreted for agreement in qualitatively indicating fine or coarse model aerosol. Overall, GRASP/Models can be improved by adjusting adopted aerosol models specifically for coarse mode models.

Line 1124: It is important to highlight the better AOD performance of GRASP/Models than the other GRASP versions before going to the comparison with other data sets. Namely, the better performance holds for GRASP/Models but not always for the other versions.

**Response:**

We now clarify it in the text as follow.

"*the PARASOL spectral products including AOD for six wavelengths in the range 443 to 1020 nm agree well with AERONET AOD measurements, e.g. for PARASOL/Models AOD correlation coefficients R are ≥ 0.86 over land and ≥ 0.94 over ocean with BIAS not exceeding 0.01 over land and 0.02 over ocean for all wavelengths. PARSOL/Optimized and PARASOL/HP also show good agreement with AERONET for spectral AOD, however they have non-negligible bias ~0.05-0.07 spectrally.*"

---

## Author Comment (AC2) · 7 Nov 2020

**Reviewer #2:**

Dear Kirk,

We would like to thank you for the constructive and positive comments on our paper. Please find our responses below.

First and foremost, I'm not sure this manuscript is within the scope of ESSD. The "Aims and scope" portion of the ESSD website (https://www.earth-system-sciencedata.net/about/aims_and_scope.html) says: *"Earth System Science Data (ESSD) is an international, interdisciplinary journal for the publication of articles on original research data (sets), furthering the reuse of high quality data of benefit to Earth system sciences. The editors encourage submissions on original data or data collections which are of sufficient quality and have the potential to contribute to these aims. The journal maintains sections for regular length articles, brief communications (e.g. on additions to data sets) and commentaries, as well as review articles and special issues. Articles in the data section may pertain to the planning, instrumentation, and execution of experiments or collection of data. Any interpretation of data is outside the scope of regular articles. Articles on methods describe nontrivial statistical and other methods employed (e.g. to filter, normalize, or convert raw data to primary published data) as well as nontrivial instrumentation or operational methods. Any comparison to other methods is beyond the scope of regular articles."* I would think that an ESSD style manuscript on this topic would simply describe the various GRASP algorithms (briefly) and where they are archived. The majority of the paper is indeed comparisons to other data sets. I think it is far more appropriate for Atmospheric Measurement Techniques, for example (to pick another Copernicus Journal). I imagine this is something the editor needs to weigh in upon, but I wouldn't look to ESSD for this type of manuscript.

**Response:**

Multi-angular polarimetry (MAP) is always considered ideal for comprehensive retrieval of aerosol properties. GRASP algorithm was developed originally for operational processing of MAP measurements (Dubovik et al., 2011; 2014). The goal of this study is to announce the release of three archives of multi-angular polarimetry POLDER aerosol products processed by GRASP algorithm and provide comprehensive evaluation of these products against ground-based AERONET dataset,

and popular MODIS aerosol products from DT, DB and MAIAC algorithms. For example, we found out that the quality of AOD retrieval from MAP (e.g. POLDER) is at least comparable to those of MODIS like imagers. In addition, we show that the MAP observations provide more information on detailed aerosol properties, e.g. spectral fine/coarse AOD, AE, as well as aerosol absorption properties such as AAOD and SSA. In this way, we assessed the potential of MAP sensors for aerosol monitoring. These both aspects are not surprising and were already discussed intensively in aerosol community. At the same, the absence of actual product from MAP sensors has often used as an argument for suggesting some overstatement of MAP potential. In these regards, our paper is aimed to answer this pessimism.

Several of our colleagues and co-authors suggested publishing our paper in new ESSD journal. Additionally, we were also inspired by your paper (Knobelspiesse et al., 2020) published in Earth System Science Data describing the ACEPOL (Aerosol Characterization from Polarimeters and Lidar) field campaign, both of them show advances for aerosol characterization by utilizing the new era of MAP measurements from different perspectives. After additional consideration, we admit publishing our paper in other journals could be appropriate, but we remain convinced that this manuscript is rather appropriate for the Earth System Science Data. In addition, given the fact, that it was already exposed in open discussion and received several reviews, we prefer to continue with this journal.

Secondly, about the length. The manuscript review version is 108 pages long with 23 tables and 28 figures. You've chosen an extensive set of data to compare and contrast the various versions of POLDER data to the various versions of MODIS and then again to AERONET. What I'm missing is a concise set of objectives and how those are met. While I'm sure they are in the manuscript, they are lost among the noise. Because of the scale of the task, you need to be creative in finding ways to condense all of this analysis into something that easily and simply supports your work. Given the above two points, you may consider splitting this manuscript. For example, you could make an ESSD manuscript that describes basics of the dataset and its creation and archive location. It would point to one (or more) manuscripts that contain analysis. One could be on continuity with MODIS (and VIIRS?) and another on the

full set of PARASOL products and comparison to AERONET. This is just a suggestion.

**Response:**

We agree that the manuscript is a long and that for the reader may be not easy to follow all details of the manuscript. In order to address that, we have revised the manuscript by combining all comments from reviewers, and trying to make it more readable. Generally, there are 3 main parts of this manuscript, (1) validation three archives (Optimized, HP and Models) PARASOL/GRASP products (spectral AOD, AE, AODF, AODC, AAOD and SSA) against AERONET data for entire PARASOL 2005-2013; (2) comparison of results obtained from validation of PARASOL (GRASP and Operational) and MODIS (DT, DB and MAIAC) aerosol products against AERONET in year 2008; (3) Inter-comparison satellite products at global pixel-to-pixel scale. The first part was on the full set of PARASOL products and comparison to AERONT. The second part tried to compare PARASOL and MODIS aerosol products by validating with AERONET follow the same criteria. The third part was inter-comparing PARASOL and MODIS aerosol products over globe at pixel level. We agree that they can be split in different papers but the separation of the materials of the paper is very difficult, because these three parts are quite complimentary. We feel that having the 3 parts together make the story more complete and after considerations we prefer to not split the paper.

Since the core of this manuscript is comparisons of other datasets, the choice of statistical metrics is very important. Table 3, for example, shows the use of nine different metrics, although Pearson's linear correlation coefficient is most commonly employed in the text. Unfortunately, some of these metrics, and especially the linear correlation coefficient, are not suitable for use on non-gaussian distributed data. The correlation coefficient is an expression of association, not agreement (Altman and Bland, 1983 and Bland and Altman, 1986), and is subject to numerical distribution, outliers, and sample range. So, for example, the R values for SSA are lower, but is that because of the lower success of the PARASOL retrieval (what you want to know) or because of the truncated numerical distribution of SSA which makes it non gaussian distributed? Additionally, what threshold of R can be considered a success? To that end, I think your metric for percentage within the GCOS requirements is a much more appropriate measure, and should instead be emphasized, although you

need to take care to account for measurement uncertainty in both POLDER and MODIS or AERONET. Seegers et al. 2018 is a nice overview of these issues in ocean color data products, but equally appropriate here. They also identify regression slope and root mean square error as problematic, while noting that mean bias (which you use) and mean absolute error as appropriate. Bland and Altman suggest something similar, also recommending the pairwise mean bias and the "limits of agreement" which is similar to the mean absolute error. Variable measurement uncertainty can also be incorporated into these techniques (Knobelspiesse et al, 2019) which addresses the salient question "Do measurements agree to within stated uncertainties?" Ultimately, you should revise (and perhaps simplify) the metrics you use to assess your results.

Altman, D. G. and Bland, J. M.: Measurement in medicine: the analysis of method comparison studies, The statistician, 307–317 , 1983.

Bland, J. M. and Altman, D.: Statistical methods for assessing agreement between two methods of clinical measurement, The lancet, 327(8476), 307–310 , 1986.

Knobelspiesse, K., Tan, Q., Bruegge, C., Cairns, B., Chowdhary, J., van Diedenhoven, B., Diner, D., Ferrare, R., van Harten, G., Jovanovic, V., Ottaviani, M., Redemann, J., Seidel, F., and Sinclair, K.: Intercomparison of airborne multi-angle polarimeter observations from the Polarimeter Definition Experiment, Appl. Optics, 58(3), 650–669 , 2019.

Seegers, B. N., Stumpf, R. P., Schaeffer, B. A., Loftin, K. A., and Werdell, P. J.: Performance metrics for the assessment of satellite data products: an ocean color case study, Optics express, 26(6), 7404–7422 , 2018.

**Response:**

We fully agree that to choosing the adequate statistic metrics is very challenging. Therefore, for addressing this challenge we have presented many parameters at the same. In our understanding this approach allows us to have more comprehensive evaluation of the comparison results. Indeed, each single criterion has some limitations. For example, we agree that GCOS requirement is a good measure for AOD comparison, however, the total GCOS value tends to bias to small AOD, since >70% cases are coming from AOD<0.2. In Figures 7 and 8, GRASP/Models, MODIS DT, DB and MAIAC are all showing GCOS>45% for all AOD cases; while for AOD>0.2, most of them having GCOS<30%. The root mean square error (RMSE) also tends to be smaller for the products dominated by the results at lower AOD. The

BIAS may be misleading in cases when many deviations with opposite sign are added together. For example, it is often a case for AE. In Figure 3, GRASP/Models tend to overestimate of small AE and underestimate high AE, and total BIAS is close to ~zero, which is smaller than GRASP/Optimized and GRASP/HP. Another example, it the Table 10, where for very bright surfaces, many retrievals have positive bias that is compensated by the negative bias at higher AODs. Thus, in the revised version of the paper, we have tired not to focus the discussion on a single parameter throughout the manuscript. Also, we wanted to make sure to provide all parameters, such as correlation coefficients, that are traditionally used in satellite comparisons. This helps us compare our results with published ones. In addition, following your recommendations, we have revised the expression to emphasize on all evaluation metrics, e.g. R, RMSE, BIAS, Slope, Offset and GCOS.

Co-author Sayer has a publication about the numerical distribution of AOD and its impact on averages of data which is relevant to your matchup methodology and your use of Level 3 products. Were you calculating arithmetic or geometric means? I assume they are arithmetic since you don't mention otherwise, however this can in some cases cause an artificial bias in the results.

Sayer, A. M. and Knobelspiesse, K. D.: How should we aggregate data? Methods accounting for the numerical distributions, with an assessment of aerosol optical depth, Atmos. Chem. Phys., 19(23), 15023–15048, https://doi.org/10.5194/acp-19-15023-2019, 2019.

**Response:**

Yes, the L3 0.1 degree products were calculating using arithmetic mean and gdalwarp regridding technique (https://gdal.org/programs/gdalwarp.html). We have included this information in the text. As the spatial resolution is fine (compared to typical satellite composites at 1 degree), the arithmetic vs. geometric differences (as discussed in Sayer & Knobelspiesse, 2019) is likely significantly smaller for the present case.

I'm a little surprised that you make no mention of the retrieval algorithms for PARASOL (Hasekamp et al. 2011). In the beginning of section 2.1 you mention "A unique aspect of GRASP is that it can perform radiative transfer (RT) computations fully accounting for multiple interactions of the scattered solar light in the

atmosphere, and that it can perform it online without the use of traditional LUTs." You show that GRASP is an extremely power retrieval algorithm, but it is certainly not unique in its use of iterative RT computations, and I find it problematic that you make this claim and do not mention similar algorithms. Without acknowledging that there are other algorithms, even for POLDER, the manuscript sounds more like a sales pitch for GRASP and less an dispassionate piece of peer reviewed literature. In addition to Hasekamp et al, 2011, which is applied to POLDER/PARASOL, many others come to mind including those listed below.

Di Noia, A., Hasekamp, O. P., Wu, L., van Diedenhoven, B., Cairns, B., and Yorks, J. E.: Combined neural network/Phillips–Tikhonov approach to aerosol retrievals over land from the NASA Research Scanning Polarimeter, Atmos. Meas. Tech., 10(11), 4235–4252 , https://doi.org/10.5194/amt-10-4235-2017, 2017.

[revised manuscript text omitted]

---

## Author Comment (AC3) · 7 Nov 2020

**Reviewer #3:**

Dear Greg,

We would like to thank you for the constructive and positive comments on our paper. Please find our responses below.

This paper applies several versions of a relatively new retrieval algorithm (GRASP/HP, GRASP/Optimized, GRASP/Models) to an existing satellite measurement archive (POLDER/PARASOL). The authors then compare the results from these algorithms to several legacy retrieval algorithms, including MODIS Dark Target (MODIS/DT), Deep Blue (MODIS/DB), Multi-Angle Implementation of Atmospheric Correction (MAIAC), the operational PARASOL product (PARASOL/Operational), and AERONET. The authors provide a large number of maps and statistics that not only inform the reader about the performance of the GRASP products, but they also inform the reader about the performance of the legacy aerosol retrievals (e.g., Tables 9 & 10). The paper is clear and well written and I find it suitable for publication.

**Response:**

Thank you for the positive comments on our manuscript. We provide point-by-point responses as follows.

The paper is also quite long (48 pages of text, 23 tables, and 28 figures) and probably won't be carefully read in its entirety by anyone except the reviewers. One could paraphrase the paper as "here are some new data products, and here is how they compare to similar data products as well as the gold standard (AERONET)." Nobody will learn the machinery behind the retrievals from this paper, but there are other papers for that. One reviewer pointed out that the statistical parameters chosen for this paper are not ideal, but the authors use statistical parameters that are familiar to many readers (correlation coefficient, bias, RMSE, etc.) and common in many satellite/AERONET comparison papers. Unfortunately, the aerosol remote sensing community has not yet adopted a "skill score" for comparisons, as is sometimes used in the modeling community (Taylor, 2001).

It is important to have all of this material in one place, in my opinion, so that readers can quickly assess the relative performance of the different algorithms for the various

parameters. However, hyperlinks to tables, figures, citations, and section headings would greatly improve the readability of the paper. A bookmarked Table of Contents in the sidebar would also be helpful. I had to keep two copies of the paper open on my screen – one for the text, and another for reading tables and figures. Otherwise, I would have spent as much time scrolling as reading for this paper! Hyperlinks would allow the reader to go directly to a table, and then return to the text with the "previous view" buttons. Hopefully this is something that can be accommodated in the typesetting process.

Taylor, K.: Summarizing multiple aspects of model performance in a single diagram, J. Geophys. Res., 106, 7183–7192, 2001.

**Response:**

These are very constructive suggestions. We have revised the manuscript and included a Table of Content and hyper link of all tables and figure.

I noticed that the data volume for GRASP/Models is much different than the data volume for GRASP/HP and GRASP/Optimized. For instance, GRASP/Optimized shows 41,268 AOD comparisons with AERONET over land in Table 3, but GRASP/Models only shows 27,551 comparisons. However, GRASP/Model comparisons are greater than GRASP/Optimized over ocean (2064 vs 1495). These large discrepancies appear elsewhere in the paper as well. I found this quite odd, since all three retrievals use the same instrument (PARASOL). I imagine that the cloud screening procedure is identical for all three algorithms, so I suspect that GRASP/Model fails to provide a retrieval much more frequently over land than than the other two GRASP algorithms (and that GRASP/Optimized, GRASP/HP fail more frequently than GRASP/Models over ocean). This should be discussed, since GRASP/Models is lauded for its ability to retrieve AOD (550) (e.g., line 1180). The success rate of a retrieval is important to readers, too.

**Response:**

Indeed, the different number of points and somewhat different approaches of quality filtering is one of the main shortcomings in our study. Reviewer #1 raised similar question. In fact, the post-processing flow (L1-L2-L3) was based on several attempts dictated by practical needs. These attempts provided us valuable inside but the could be fully evaluated only after full-scale validation. For example, the level 3 GRASP/HP and Optimized archives were generated and released much earlier than

GRASP/Models. Also, we have done preprocessing of GRASP/Models over land at first and learned that screening was very conservative. Based on that we used less conservative screening for Models reprocessing over ocean. Once the products were released, they were used by many users, therefore, regenerating Level 3 products was not reasonable for this study. We are considering the harmonization of the all archives in future once time and resources allow that.

At the same time, we have looked at possible effect of applying tighter screening to HP and Optimized data. Our analysis showed that although stricter screening somewhat improves the correlations, it doesn't change conceptually the results of validations. For example, it does not improve the BIAS, which is considered as a main issue for these data sets. Some explanations of this aspect were added in the Sect 2.4 and Sect 3.1 as follow.

It is also curious that GRASP/Models did so well for AOD at nearly all wavelengths over both land and ocean (Table 3), but the AEs for GRASP/Models is significantly worse than the other GRASPs. Since AE is derived from AOD, I would have thought the retrieval that produced the best AOD at multiple wavelengths would also produce the best AE. A comment about this would be helpful.

**Response:**

This is a good point that should be mentioned in the text. Yes, according to the analysis, we found out that the good agreement of spectral AOD is not equivalent to the good agreement for AE, even though AE is derived spectral AOD at two wavelengths. Apparently, obtaining good agreement for spectral AOD seems easier than for AE. The potential reason is that the spectral contrast is crucial to calculate AE, and small error in AOD at each wavelength can result in large AE uncertainty. The level of uncertainty (e.g. ±0.1, RMSE=0.1~0.15) of satellite-derived spectral AOD may makes it challenging accurate estimation of AE. At the same time, the same uncertainty is sufficient for good agreement of AOD at different wavelengths. Thus, the AOD in each channel may correlate well in time, while for each singe retrieval spectral deviations can be significant. Also, relatively small spectral deviations may perturb AE strongly while not to be as notable for AOD at each wavelength.

Is there a reason for comparing AODf and AODc to the SDA extinction-based retrievals instead of using the sky scan retrievals? The sky scans are probably more accurate. Many readers (most?) won't know the methodology behind SDA and may incorrectly assume that it is derived from the sky scans. The SDA papers use the AERONET almucantar scans as a performance benchmark, so why not use the same benchmark? You're already using the AERONET sky-scans for SSA and AAOD.

**Response:**

We have done AODF and AODC comparison with AERONET ALM retrievals (see in Figures R2 and R3). In general, they show quite similar performance with Figures 4 and 5 that evaluation against AERONET SDA AODF and AODC. Please note, in order to find more matched pairs, the AERONET ALM retrievals are collocated within ±180mins for satellite overpass, which is much bigger time window that that for SDA products ±30mins. To ensure the retrieval quality, AERONET ALM L2 retrieval products are available for AOD440>0.4, which roughly filter 80% low AOD cases (in future we could use AERONET L1.5 inversion products). By using SDA extinction-based products, we get almost the same amount of points for AOD, AODF, and AODC, which help to understand the overall performance for low, medium and high AOD cases.

[Figure]

Figure R2. Evaluation of all archive PARASOL/GRASP AODF (2005-2013) at 440 nm with AERONET INV AODF, (a) GRASP/Optimized; (b) GRASP/HP; (c) GRASP/Models.

[Figure]

Figure R3. The same with Figure R2, but for AODC at 440 nm

Line 588:

The authors say that they are using AERONET L2 inversions, but which version of AERONET (i.e., Version 2 or Version 3)?

**Response:**

We are using Version 3 AERONET L2 inversion. We now clarify this in the text.

Lines 590-595:

This paragraph will probably confuse some people. Line 590 says that AERONET L2 inversion products require AOD(440) > 0:4, but the PARASOL/GRASP filtering includes much lower values, especially over ocean (the authors require PARASOL/GRASP AOD(443) > 0:3 over land and AOD(443) > 0:02 over ocean). However, since AERONET L2 requires AOD(440) > 0:4, many of the low PARASOL/GRASP AODs won't actually appear in the comparisons anyways. . . . Unless the authors using Level 1.5 AERONET inversions at low AOD, like some other authors? If so, what are the Level 1.5 constraints?

**Response:**

We are using Version 3 AERONET L2 inversion products, which includes only AERONET AOD>0.4. Here we use additional filter for satellite AOD (Land: AOD 443 nm>0.3; Ocean: AOD 443 nm>0.02).

MINOR ISSUES

Line 35:

The links do not take me directly to the data products. www.icare.univ-lille.fr takes me to the main page, and the 2nd link on that line tries to take me to www.grasp-35, but that is a dead end.

**Response:**

Yes, the provided link is to ICARE main page, the PARASOL products are published at this path: https://www.icare.univ-lille.fr/data-access/data-archive-access/?dir=PARASOL/, ICARE account, that is free registration, is required to login. For the second link, it should work as https://www.grasp-open.com/products/.

Line 1125

Do you mean GRASP/Models instead of PARASOL/Models?

**Response:**

Yes, we use 'PARASOL/Models' to represent PARASOL products generated by GRASP/Models approach.

It would be interesting to repeat the AOD comparisons using AERONET's "coincident" AOD (that is, using only the AODs that are used during the sky scans). This would be interesting because the cloud screening for the sky-scan products is more comprehensive than for the direct AOD measurements, and it is possible that satellite (and model) AOD performance comparisons wrt AERONET will differ for these two datasets. If the coincident AODs comparisons are different than the "all AODs" comparisons, this could assist our thinking wrt the other sky-scan products. You are probably already set up to do this. I include this as a minor issue, though, because the paper is already too long and this should really be a topic for another day.

**Response:**

This is a good suggestion, which brings us additional thoughts! We agree that AERONET direct sun cloud screening is not as comprehensive as for sky-scan products. However, the evaluation with collocated satellite retrievals also introduces the satellite cloud screen to ensure the quality. On the other hand, when utilizing AERONET sky-scan products, in order to find more matched pairs we normally adopt a wider time window, e.g. ±180 mins, which may increase the issue of aerosol temporal variability.